# Tighter Convergence Bounds for Shuffled SGD via Primal-Dual Perspective

**Xufeng Cai**[*]
Department of Computer Sciences
University of Wisconsin-Madison
`xcai74@wisc.edu`

**Cheuk Yin Lin**[*]
Department of Computer Sciences
University of Wisconsin-Madison
`cylin@cs.wisc.edu`

**Jelena Diakonikolas**
Department of Computer Sciences
University of Wisconsin-Madison
`jelena@cs.wisc.edu`

## Abstract

Stochastic gradient descent (SGD) is perhaps the most prevalent optimization method in modern machine learning. Contrary to the empirical practice of sampling from the datasets *without replacement* and with (possible) reshuffling at each epoch, the theoretical counterpart of SGD usually relies on the assumption of *sampling with replacement*. It is only very recently that SGD using sampling without replacement – shuffled SGD – has been analyzed with matching upper and lower bounds. However, we observe that those bounds are too pessimistic to explain often superior empirical performance of data permutations (sampling without replacement) over vanilla counterparts (sampling with replacement) on machine learning problems. Through fine-grained analysis in the lens of primal-dual cyclic coordinate methods and the introduction of novel smoothness parameters, we present several results for shuffled SGD on smooth and non-smooth convex losses, where our novel analysis framework provides tighter convergence bounds over all popular shuffling schemes (IG, SO, and RR). Notably, our new bounds predict faster convergence than existing bounds in the literature – by up to a factor of $O(\sqrt{n})$, mirroring benefits from tighter convergence bounds using component smoothness parameters in randomized coordinate methods. Lastly, we numerically demonstrate on common machine learning datasets that our bounds are indeed much tighter, thus offering a bridge between theory and practice.

## 1 Introduction

Originally proposed in [38], SGD has been broadly studied in the machine learning literature due to its effectiveness in large-scale settings, where full gradient computations are often computationally prohibitive. When applied to unconstrained finite-sum problems

$$\min_{\boldsymbol{x} \in \mathbb{R}^d} f(\boldsymbol{x}), \quad \text{where} \quad f(\boldsymbol{x}) := \frac{1}{n} \sum_{i=1}^{n} f_i(\boldsymbol{x}), \tag{P}$$

SGD performs the update $\boldsymbol{x}_t = \boldsymbol{x}_{t-1} - \eta \nabla f_{i_t}(\boldsymbol{x}_{t-1})$ for $i_t \in [n]$ ($[n] := \{1, \ldots, n\}$), in each iteration $t$. Traditional theoretical analysis for SGD builds upon the assumption of sampling $i_t \in [n]$ with replacement according to a fixed distribution $\mathbf{p} = (p_1, \ldots, p_n)^\top$ over $[n]$, which leads to

---

[*]Equal contribution

38th Conference on Neural Information Processing Systems (NeurIPS 2024).

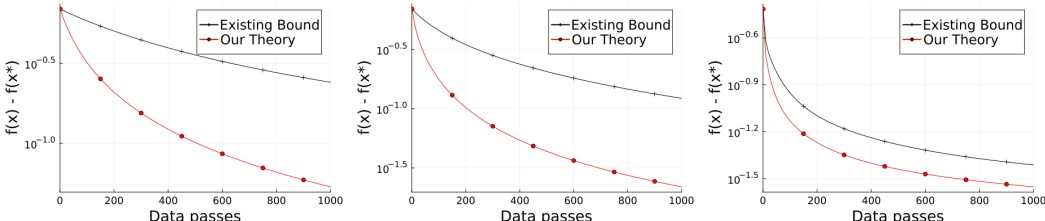

Figure 1: An illustration of the convergence behaviour of shuffled SGD for logistic regression problems on LIBSVM datasets `luke`, `leu` and `a9a`, where we use step sizes from existing bounds and our work. Due to randomness, we average over 20 runs for each plot and include a ribbon around each line to show its variance. However, as suggested by the concentration of $\hat{L}$ (see Section 4.1 and Appendix E), the variance across multiple runs is negligible, hence the ribbons are not observable.

$\mathbb{E}_{i_t}[\nabla f_{i_t}(\boldsymbol{x}_{t-1})/(np_{i_t})] = \nabla f(\boldsymbol{x}_{t-1})$, and thus much of the (deterministic) gradient descent-style analysis can be transferred to this setting. By contrast, no such connection between the component and the full gradient can be established for shuffled SGD — which employs sampling *without replacement* — making its analysis much more challenging. As a result, despite its fundamental nature, there were no non-asymptotic convergence results for shuffled SGD until a very recent line of work [2, 12, 20, 21, 30, 31, 34, 35, 42]. All existing results consider general finite sum problems, with the same regularity condition constant (Lipschitz constant of $f_i$ or its gradient) assumed for all the component functions. As a result, the obtained convergence bounds are typically no better than for (full) gradient descent, and are only better than the bounds for SGD with replacement sampling if the algorithm is run for many full passes over the data [30, 34].

Furthermore, there is a large gap between the empirical performance of shuffled SGD and the predicted convergence rates from prior work [20, 30]. One cause for this discrepancy are overly pessimistic bounds on the step size in prior work, which are of order $1/(nL_{\max})$, where $L_{\max}$ is the maximum smoothness constant over components $f_i$ in (P). In practice, the step sizes are tuned to achieve better convergence bounds than predicted by the current theory. We illustrate how restrictions on the step size affect convergence of shuffled SGD (with random permutations in each epoch) in Fig. 1, where we plot the resulting optimality gap over full data passes when shuffled SGD is applied to logistic regression problems on standard datasets. To compare the effect of the step size $\eta$ from prior work and our work, we choose take $\eta = 1/(\sqrt{2}nL_{\max})$ based on [30], and $\eta = 1/(n\sqrt{\hat{L}\tilde{L}})$ from our work, where $\hat{L}, \tilde{L}$ are our novel fine-grained, data-dependent smoothness parameters defined in Section 3 for smooth convex finite-sum problems with linear predictors. As can be observed from Fig. 1, larger step sizes resulting from our theory lead to faster convergence of shuffled SGD and, as a result, our convergence bounds better predict the performance of shuffled SGD.

Building on these insights, we introduce a fined-grained theoretical analysis to transparently show how the structure of the data and the possibly different Lipschitz constants of the component functions or their gradients affect the performance of shuffled SGD, thus providing a better explanation of the heuristic success of shuffled SGD in modern machine learning.

### 1.1 Background and related work

SGD (with replacement) has been extensively studied in many settings (see e.g., [1, 9, 10, 38] for convex optimization). Compared to SGD, shuffled SGD usually exhibits faster convergence in practice [8, 37], and is easier and more efficient to implement [5]. For each epoch $k$, shuffled SGD-style algorithms perform incremental gradient updates based on the sample ordering (permutation of the data points) denoted by $\pi^{(k)}$. There are three main choices of data permutations: (i) $\pi^{(k)} \equiv \pi$ for some fixed permutation of $[n]$ for all epochs, where shuffled SGD reduces to the incremental gradient (IG) method; (ii) $\pi^{(k)} \equiv \tilde{\pi}$ where $\tilde{\pi}$ is randomly chosen only once, at the beginning of the first epoch, referred to as the shuffle-once (SO) scheme; (iii) $\pi^{(k)}$ randomly generated at the beginning of each epoch, referred to as random reshuffling (RR).

For general smooth convex settings, the convergence of shuffled SGD has been established only recently. For the number of epochs $K$ sufficiently large, [31] proved a convergence rate

Table 1: Comparison of our results with state of the art, in terms of individual gradient oracle complexity required to output $\boldsymbol{x}_{\text{out}}$ with $\mathbb{E}[f(\boldsymbol{x}_{\text{out}}) - f(\boldsymbol{x}_*)] \leq \epsilon$, where $\epsilon > 0$ is the target error and $\boldsymbol{x}_*$ is the optimal solution. Here, $\sigma_*^2 = \frac{1}{n}\sum_{i=1}^n \|\nabla f_i(\boldsymbol{x}_*)\|_2^2$, $D = \|\boldsymbol{x}_0 - \boldsymbol{x}_*\|_2$, and generalized linear model refers to objectives of the form $f(\boldsymbol{x}) = \frac{1}{n}\sum_{i=1}^n \ell_i(\boldsymbol{a}_i^\top \boldsymbol{x})$ as defined in Section 3. Parameters $\hat{L}^g, \tilde{L}^g$ are defined in Section 2 and satisfy $\hat{L}^g \leq \frac{1}{n}\sum_{i=1}^n L_i$ and $\tilde{L}^g \leq L_{\max}$. Parameters $\hat{L}, \tilde{L}$, and $\bar{G}$ are defined in Section 3, and are discussed in the text of this section.

| PAPER | | COMPLEXITY | ASSUMPTIONS | STEP SIZE |
|---|---|---|---|---|
| NGUYEN ET AL. [34] CHA ET AL. [12] | (RR) | $\mathcal{O}\left(\frac{nL_{\max}D^2}{\epsilon} + \frac{\sqrt{nL_{\max}}\sigma_* D^2}{\epsilon^{3/2}}\right)$ | $f_i$: $L_{\max}$-SMOOTH, CONVEX | $\mathcal{O}\left(\frac{1}{nL_{\max}}\right)$ |
| MISHCHENKO ET AL. [30] | (RR/SO) | $\mathcal{O}\left(\frac{nL_{\max}D^2}{\epsilon} + \frac{\sqrt{nL_{\max}}\sigma_* D^2}{\epsilon^{3/2}}\right)$ | $f_i$: $L_{\max}$-SMOOTH, CONVEX | $\mathcal{O}\left(\frac{1}{nL_{\max}}\right)$ |
| **[Ours, Theorem 1]** | (RR/SO) | $\mathcal{O}\left(\frac{n\sqrt{\hat{L}^g \tilde{L}^g}D^2}{\epsilon} + \frac{\sqrt{n\tilde{L}^g}\sigma_* D^2}{\epsilon^{3/2}}\right)$ | $f_i$: $L_i$-SMOOTH, CONVEX | $\mathcal{O}\left(\frac{1}{n\sqrt{\hat{L}^g \tilde{L}^g}}\right)$ |
| **[Ours, Theorem 2]** | (RR/SO) | $\mathcal{O}\left(\frac{n\sqrt{\hat{L}\tilde{L}}D^2}{\epsilon} + \frac{\sqrt{n\tilde{L}}\sigma_* D^2}{\epsilon^{3/2}}\right)$ | $\ell_i$: $L_i$-SMOOTH, CONVEX GENERALIZED LINEAR MODEL | $\mathcal{O}\left(\frac{1}{n\sqrt{\hat{L}\tilde{L}}}\right)$ |
| CHA ET AL. [12] LOWER BOUND | (RR) | $\Omega\left(\frac{\sqrt{nL_{\max}}\sigma_* D^2}{\epsilon^{3/2}}\right)$ | $f_i$: $L_{\max}$-SMOOTH, CONVEX, LARGE $K$ | $\mathcal{O}\left(\frac{1}{nL_{\max}}\right)$ |
| SHAMIR [42] | (RR/SO) | $\mathcal{O}\left(\frac{\bar{B}^2 G_{\max}^2}{\epsilon^2}\right)$ $(K = 1, n = \Omega(1/\epsilon^2))$ | $\ell_i$: $G_{\max}$-LIPSCHITZ, CONVEX $\bar{B}$-BOUNDED ITERATES, $\|\boldsymbol{a}_i\| \leq 1$ GENERALIZED LINEAR MODEL | $\mathcal{O}\left(\frac{1}{\sqrt{n}}\right)$ |
| **[Ours, Theorem 3]** | (RR/SO) | $\mathcal{O}\left(\frac{n\bar{G}D^2}{\epsilon^2}\right)$ | $\ell_i$: $G_i$-LIPSCHITZ, CONVEX GENERALIZED LINEAR MODEL | $\mathcal{O}\left(\frac{1}{n\sqrt{\bar{G}K}}\right)$ |

$\mathcal{O}(1/\sqrt{nK})$ for RR, which leads to the complexity matching SGD. This result was later improved to $\mathcal{O}(1/(n^{1/3}K^{2/3}))$ by [12, 30, 34] for $K$ sufficiently large and with bounded variance assumed at the minimizer, while the same rate holds for SO [30]. These results were complemented by matching lower bounds in [12], under sufficiently small step sizes as utilized in prior work. The results in [30, 34] require restricted $\mathcal{O}(1/(nL))$ step sizes and reduce to $\mathcal{O}(1/K)$ for small $K$, acquiring the same iteration complexity as full-gradient methods. Unlike in strongly convex settings, we are not aware of any follow-up work with improvements under small $K$ for smooth convex settings.

The major difficulty in analyzing shuffled SGD comes from characterizing the difference between the intermediate iterate and the iterate after one full data pass, for which current analysis (see e.g., [30] in smooth convex settings) uses the global smoothness constant with a triangle inequality. Such a bound may be too pessimistic and fail capturing the nuances of intermediate progress of shuffled SGD, which leads to a small step size and large $K$ restrictions. To provide a more fine-grained analysis that narrows the theory-practice gap for shuffled SGD, we notice that such a proof difficulty is reminiscent of the analysis of cyclic block coordinate methods relating the partial gradients to the full one. This natural connection was further emphasized in studies of cyclic methods with random permutations [24, 47]; however, these results were limited to convex quadratics. More generally, it is possible to interpret shuffled SGD as a primal-dual method performing cyclic updates on the dual side (see (PD) in Section 2.1 and (PL-PD) in Section 3). We note here that prior work on dual coordinate methods [41] provided theoretical guarantees only for the algorithms that choose the dual coordinate to optimize uniformly at random, while the cyclic variant (related to shuffled SGD) had only been studied numerically up until this work. Further discussion of related work appears in Appendix A.

## 1.2 Contributions

In this work, we study the convergence rates of shuffled SGD in various settings through a unified primal-dual perspective, making intriguing connections to cyclic coordinate methods. This analysis framework is novel and allows us to leverage cyclic bias accumulation techniques on the dual side to obtain fine-grained convergence bounds. The obtained bounds mirror the improvements in randomized coordinate methods, which come from different coordinate smoothness parameters. While coordinate methods are no better than full-gradient methods in the worst case, on typical

problem instances, they are much faster and the improvements come precisely from a more fine-grained view of smoothness. We see a similar phenomenon in our analysis, which highlights the usefulness of the fine-grained smoothness characterizations introduced in our work.

We provide improved bounds for all three popular data permutation strategies RR, SO and IG, in smooth convex settings. When the problem objective narrows to empirical risk minimization with linear predictors, we are able to exploit the data-dependent structure and uncouple the linear and nonlinear parts of the objective function, allowing us to provide tighter data-dependent bounds, up to a factor of $O(\sqrt{n})$. Moreover, we show that our techniques extend to non-smooth convex settings, providing improved bounds over existing work.

We summarize our results and compare them to the state of the art in Table 1. As is standard, all complexity results in Table 1 are expressed in terms of individual (component) gradient evaluations. They represent the number of gradient evaluations required to construct a solution with (expected) optimality gap $\epsilon$, given a target error $\epsilon > 0$.

**Extensions to mini-batching and IG.** When presenting our results for general finite-sum problems (in Section 2), we consider simple updates without mini-batching for ease of presentation and to avoid introducing excessive notation. However, we emphasize that all our results can be extended to shuffled SGD with mini-batching. Our results are also the first to provide convergence bounds that demonstrate benefits of mini-batching in shuffled SGD. For completeness and generality, the proofs in the appendix are carried out for mini-batch settings with arbitrary batch sizes $b \in \{1, \ldots, n\}$. Thus, all the results stated in Section 2 can be recovered by setting $b = 1$. Moreover, our framework can provide similar fine-grained convergence bounds for IG. However, as IG is not as commonly used in practice compared to RR and SO and due to space constraints, we only present our results for RR and SO in the main body and include the results for IG in the appendix.

## 1.3 Notation

We consider a real $d$-dimensional Euclidean space $(\mathbb{R}^d, \|\cdot\|)$ where $d$ is finite and $\|\cdot\|$ is the $\ell_2$-norm. For a vector $\boldsymbol{x}$, we let $\boldsymbol{x}^j$ denote its $j$-th coordinate. For any positive integer $m$, we use $[m]$ to denote the set $\{1, 2, \ldots, m\}$. Given a matrix $\boldsymbol{A}$, $\|\boldsymbol{A}\| := \sup_{\boldsymbol{x} \in \mathbb{R}^d, \|\boldsymbol{x}\| \leq 1} \|\boldsymbol{A}\boldsymbol{x}\|$ denotes its operator norm. For a positive definite matrix $\boldsymbol{\Lambda}$, $\|\cdot\|_{\boldsymbol{\Lambda}}$ denotes the Mahalanobis norm, $\|\boldsymbol{x}\|_{\boldsymbol{\Lambda}} := \sqrt{\langle \boldsymbol{\Lambda}\boldsymbol{x}, \boldsymbol{x}\rangle}$. We use $\boldsymbol{I}$ to denote the identity matrix, and $\mathrm{diag}(\boldsymbol{v})$ to denote the diagonal matrix with vector $\boldsymbol{v}$ on the main diagonal. For any $j \in [n]$, we define $\boldsymbol{I}_{j\uparrow}$ as the matrix obtained from the identity matrix $\boldsymbol{I}$ by setting the first $j$ diagonal elements to zero, and let $\boldsymbol{I}_j$ be the matrix with only the $j$-th diagonal element nonzero and equal to $1$. To handle the cases with random data permutations, we use the following definitions corresponding to the data permutation $\pi = \{\pi^1, \pi^2, \ldots, \pi^n\}$ of $[n]$: $\boldsymbol{A}_\pi := \left[\boldsymbol{a}_{\pi_1}, \boldsymbol{a}_{\pi_2}, \ldots, \boldsymbol{a}_{\pi_n}\right]^\top$ permuting the rows based on $\pi$ given a matrix $\boldsymbol{A} = \left[\boldsymbol{a}_1, \boldsymbol{a}_2, \ldots, \boldsymbol{a}_n\right]^\top$, and $\boldsymbol{v}_\pi := \left(\boldsymbol{v}^{\pi_1}, \boldsymbol{v}^{\pi_2}, \ldots, \boldsymbol{v}^{\pi_n}\right)^\top$ permuting the coordinates/subvectors based on $\pi$ given a vector $\boldsymbol{v} = (\boldsymbol{v}^1, \boldsymbol{v}^2, \ldots, \boldsymbol{v}^n)^\top$.

## 2 Primal-Dual Framework for Smooth Convex Finite-Sum Problems

Throughout this section, we make the following standard assumptions.

**Assumption 1.** *Each $f_i$ is convex and $L_i$-smooth, and there exists a minimizer $\boldsymbol{x}_* \in \mathbb{R}^d$ for $f(\boldsymbol{x})$.*

Assumption 1 implies that $f$ and all component functions $f_i$ are $L$-smooth, where $L_{\max} := \max_{i \in [n]} L_i$. It also implies that each convex conjugate $f_i^*$ is $\frac{1}{L_i}$-strongly convex [3]. In this section, we define $\boldsymbol{\Lambda} = \mathrm{diag}(\underbrace{L_1, \ldots, L_1}_{d}, \ldots, \underbrace{L_n, \ldots, L_n}_{d}) \in \mathbb{R}^{nd \times nd}$, and slightly abuse the notation to use $\boldsymbol{\Lambda}_\pi = \mathrm{diag}\left(\underbrace{L_{\pi^1}, \ldots, L_{\pi^1}}_{d}, \ldots, \underbrace{L_{\pi^n}, \ldots, L_{\pi^n}}_{d}\right)$ given a permutation $\pi$ of $[n]$. For the permutation $\pi_k$ at the $k$-th epoch, we denote $\boldsymbol{\Lambda}_k = \boldsymbol{\Lambda}_{\pi_k}$, for brevity.

We further assume that the variance at $\boldsymbol{x}_*$ is bounded, same as prior work [30, 34].

**Assumption 2.** *The quantity $\sigma_*^2 = \frac{1}{n}\sum_{i=1}^n \|\nabla f_i(\boldsymbol{x}_*)\|^2$ is bounded.*

**Algorithm 1** Shuffled SGD (Primal-Dual View, General Convex Smooth)

---

1: **Input:** Initial point $\boldsymbol{x}_0 \in \mathbb{R}^d$, step size $\{\eta_k\} > 0$, number of epochs $K > 0$
2: **for** $k = 1$ to $K$ **do**
3:     Generate some permutation $\pi_k$ of $[n]$ (either deterministic or random)
4:     $\boldsymbol{x}_{k-1,1} = \boldsymbol{x}_{k-1}$
5:     **for** $i = 1$ to $n$ in the ordering of $\pi_k$ **do**
6:         $\boldsymbol{y}_k^i = \arg\max_{\boldsymbol{y}^i \in \mathbb{R}^d} \left\{ \langle \boldsymbol{y}^i, \boldsymbol{x}_{k-1,i} \rangle - f_i^*(\boldsymbol{y}^i) \right\}$
7:         $\boldsymbol{x}_{k-1,i+1} = \arg\min_{\boldsymbol{x} \in \mathbb{R}^d} \left\{ \langle \boldsymbol{y}_k^i, \boldsymbol{x} \rangle + \frac{1}{2\eta_k} \|\boldsymbol{x} - \boldsymbol{x}_{k-1,i}\|^2 \right\} = \boldsymbol{x}_{k-1,i} - \eta_k \nabla f(\boldsymbol{x}_{k-1,i})$
8:     **end for**
9:     $\boldsymbol{x}_k = \boldsymbol{x}_{k-1,n+1}$, $\boldsymbol{y}_k = \left( \boldsymbol{y}_k^1, \boldsymbol{y}_k^2, \ldots, \boldsymbol{y}_k^n \right)^\top$
10: **end for**
11: **Return:** $\hat{\boldsymbol{x}}_K = \sum_{k=1}^K \eta_k \boldsymbol{x}_k / \sum_{k=1}^K \eta_k$

---

## 2.1 Primal-dual view of shuffled SGD

Problem (P) can be reformulated into a primal-dual form using the standard Fenchel conjugacy argument (see, e.g., [13, 14]),

$$\min_{\boldsymbol{x} \in \mathbb{R}^d} \max_{\boldsymbol{y} \in \mathbb{R}^{nd}} \left\{ \mathcal{L}(\boldsymbol{x}, \boldsymbol{y}) := \frac{1}{n} \sum_{i=1}^n \left( \langle \boldsymbol{y}^i, \boldsymbol{x} \rangle - f_i^*(\boldsymbol{y}^i) \right) \right\}, \tag{PD}$$

where we slightly abuse the notation to denote $\boldsymbol{y} = (\boldsymbol{y}^1, \ldots, \boldsymbol{y}^n)^\top \in \mathbb{R}^{nd}$ and $f_i^*$ is the convex conjugate of $f_i$ defined by $f_i^*(\boldsymbol{y}) = \sup_{\boldsymbol{x} \in \mathbb{R}^d} \left\{ \langle \boldsymbol{y}, \boldsymbol{x} \rangle - f_i(\boldsymbol{x}) \right\}$. We let $\boldsymbol{y}_{\boldsymbol{x}} = (\boldsymbol{y}_{\boldsymbol{x}}^1, \ldots, \boldsymbol{y}_{\boldsymbol{x}}^n)^\top \in \mathbb{R}^{nd}$ be the conjugate pair of $\boldsymbol{x} \in \mathbb{R}^d$, i.e., $\boldsymbol{y}_{\boldsymbol{x}}^i = \arg\max_{\boldsymbol{y} \in \mathbb{R}^d} \{ \langle \boldsymbol{y}, \boldsymbol{x} \rangle - f_i^*(\boldsymbol{y}) \}$, and we denote $\boldsymbol{y}_* = \boldsymbol{y}_{\boldsymbol{x}_*}$.

Given a primal-dual pair $(\boldsymbol{x}, \boldsymbol{y})$, the primal-dual gap of (PD) is defined by $\text{Gap}(\boldsymbol{x}, \boldsymbol{y}) = \max_{(\boldsymbol{u}, \boldsymbol{v})} \{ \mathcal{L}(\boldsymbol{x}, \boldsymbol{v}) - \mathcal{L}(\boldsymbol{u}, \boldsymbol{y}) \}$. In particular, we consider the pair $(\boldsymbol{x}, \boldsymbol{y}_*)$ for $\boldsymbol{x} \in \mathbb{R}^d$, and bound $\text{Gap}^{\boldsymbol{v}}(\boldsymbol{x}, \boldsymbol{y}_*) := \mathcal{L}(\boldsymbol{x}, \boldsymbol{v}) - \mathcal{L}(\boldsymbol{x}_*, \boldsymbol{y}_*)$ for an arbitrary but fixed $\boldsymbol{v}$. To finally obtain the function value gap $f(\boldsymbol{x}) - f(\boldsymbol{x}_*)$ for (P), we only need to choose $\boldsymbol{v} = \arg\max_{\boldsymbol{w}} \mathcal{L}(\boldsymbol{x}, \boldsymbol{w}) = \boldsymbol{y}_{\boldsymbol{x}}$.

Using this primal-dual formulation and standard convex conjugacy arguments, we can *equivalently* write the standard shuffled SGD algorithm in a primal-dual form as summarized in Algorithm 1.

**Improved bounds with new smoothness constants.** To prove a convergence bound for shuffled SGD in this general setting, we first construct an upper estimate of $\text{Gap}^{\boldsymbol{v}}(\boldsymbol{x}_k, \boldsymbol{y}_*)$ for some fixed $\boldsymbol{v}$ to be set later, as summarized in the following lemma.

**Lemma 1.** *Under Assumption 1, for any $k \in [K]$, the iterates $\{\boldsymbol{y}_k^i\}_{i=1}^n$ and $\{\boldsymbol{x}_{k-1,i}\}_{i=1}^{n+1}$ generated by Algorithm 1 satisfy*

$$\mathcal{E}_k \leq \frac{\eta_k}{n} \sum_{i=1}^n \langle \boldsymbol{y}_k^i, \boldsymbol{x}_k - \boldsymbol{x}_{k-1,i+1} \rangle + \frac{\eta_k}{n} \sum_{i=1}^n \langle \boldsymbol{v}_k^i - \boldsymbol{y}_k^i, \boldsymbol{x}_k - \boldsymbol{x}_{k-1,i} \rangle$$
$$- \frac{\eta_k}{2n} \|\boldsymbol{y}_k - \boldsymbol{v}_k\|_{\boldsymbol{\Lambda}_k^{-1}}^2 - \frac{\eta_k}{2n} \|\boldsymbol{y}_k - \boldsymbol{y}_{*,k}\|_{\boldsymbol{\Lambda}_k^{-1}}^2 - \frac{1}{2n} \sum_{i=1}^n \|\boldsymbol{x}_{k-1,i+1} - \boldsymbol{x}_{k-1,i}\|^2, \tag{1}$$

*where $\mathcal{E}_k := \eta_k \text{Gap}^{\boldsymbol{v}}(\boldsymbol{x}_k, \boldsymbol{y}_*) + \frac{1}{2n} \|\boldsymbol{x}_* - \boldsymbol{x}_k\|_2^2 - \frac{1}{2n} \|\boldsymbol{x}_* - \boldsymbol{x}_{k-1}\|_2^2$, $\boldsymbol{v}_k = \boldsymbol{v}_{\pi(k)}$, and $\boldsymbol{y}_{*,k} = \boldsymbol{y}_{*,\pi(k)}$ are the (block-wise) permuted vectors based on the permutation $\pi_k$ at the $k$-th epoch.*

We note that the first term $\mathcal{T}_1 := \frac{\eta_k}{n} \sum_{i=1}^n \langle \boldsymbol{y}_k^i, \boldsymbol{x}_k - \boldsymbol{x}_{k-1,i+1} \rangle$ from Lemma 1 can be aggregated into the terms capturing the primal progress within one epoch and cancelled by the last term in Eq. (1). The precise bound on $\mathcal{T}_1$ and its proof are provided in Lemma 10 in Appendix C.1. The second term $\mathcal{T}_2 := \frac{\eta_k}{n} \sum_{i=1}^n \langle \boldsymbol{v}_k^i - \boldsymbol{y}_k^i, \boldsymbol{x}_k - \boldsymbol{x}_{k-1,i} \rangle$ requires us to relate the intermediate iterate $\boldsymbol{x}_{k-1,i}$ to the iterate $\boldsymbol{x}_k$ after one full data pass, which corresponds to a partial sum of the component gradients, each at different iterates $\{\boldsymbol{x}_{k-1,j}\}_{j=i}^n$. In contrast to prior analyses (e.g., Mishchenko et al. [30]) using the global smoothness and triangle inequality to bound this partial sum, we provide a tighter bound on $\mathcal{T}_2$ that tracks the progress of the cyclic update on the dual side, in the aggregate.

To simplify the notation in the following lemmas and to clearly compare our results, we introduce the following novel definitions of smoothness constants for shuffled SGD:

$$\hat{L}_\pi^g := \frac{1}{n^2} \big\| \mathbf{\Lambda}_\pi^{1/2} \big( \sum_{i=1}^n \mathbf{I}_{d(i-1)\uparrow} \mathbf{E}\mathbf{E}^\top \mathbf{I}_{d(i-1)\uparrow} \big) \mathbf{\Lambda}_\pi^{1/2} \big\|_2, \qquad \hat{L}^g = \max_\pi \hat{L}_\pi^g,$$

$$\tilde{L}_\pi^g := \big\| \mathbf{\Lambda}_\pi^{1/2} \big( \sum_{i=1}^n \mathbf{I}_{(di)} \mathbf{E}\mathbf{E}^\top \mathbf{I}_{(di)} \big) \mathbf{\Lambda}_\pi^{1/2} \big\|_2, \qquad\qquad \tilde{L}^g = \max_\pi \tilde{L}_\pi^g, \tag{2}$$

where $\mathbf{I}_{(di)} = \sum_{j=d(i-1)+1}^{di} \mathbf{I}_j$ and $\mathbf{E} = [\underbrace{\mathbf{I}_d, \ldots, \mathbf{I}_d}_{n}]^\top \in \mathbb{R}^{nd \times d}$. Permutation-dependent quantities

$\hat{L}_\pi^g$ and $\tilde{L}_\pi^g$ defined in (2) are obtained directly from our analysis. We remark that $\hat{L}^g$ is bounded by the average smoothness of $f$ and $\tilde{L}^g$ is bounded by the max of individual smoothness constants of $f_i$; see more details in Appendix B. However, as we argue in later sections, these upper bounds on $\hat{L}_\pi^g$ and $\tilde{L}_\pi^g$ are loose in general, and so the convergence bounds based on $\hat{L}_\pi^g$ and $\tilde{L}_\pi^g$ that we obtain align better with the empirical performance of shuffled SGD.

Assuming that a uniformly random data shuffling strategy is used (SO or RR), the resulting bound on $\mathcal{T}_2$ is summarized in Lemma 2, while its proof is deferred to Appendix B.

**Lemma 2.** *Under Assumptions 1 and 2, for any $k \in [K]$, the iterates $\{\boldsymbol{y}_k^i\}_{i=1}^n$ and $\{\boldsymbol{x}_{k-1,i}\}_{i=1}^{n+1}$ generated by Algorithm 1 with uniformly random shuffling (RR/SO) satisfy*

$$\mathbb{E}[\mathcal{T}_2] \le \mathbb{E}\Big[ \eta_k^3 n \hat{L}_{\pi(k)}^g \tilde{L}_{\pi(k)}^g \|\boldsymbol{y}_k - \boldsymbol{y}_{*,k}\|_{\mathbf{\Lambda}_k^{-1}}^2 + \frac{\eta_k}{2n} \|\boldsymbol{v}_k - \boldsymbol{y}_k\|_{\mathbf{\Lambda}_k^{-1}}^2 \Big] + \frac{\eta_k^3 (n+1) \tilde{L}^g}{6} \sigma_*^2,$$

*where $\mathcal{T}_2 := \frac{\eta_k}{n} \sum_{i=1}^n \langle \boldsymbol{v}_k^i - \boldsymbol{y}_k^i, \boldsymbol{x}_k - \boldsymbol{x}_{k-1,i} \rangle$, $\boldsymbol{v}_k = \boldsymbol{v}_{\pi(k)}$ and $\boldsymbol{y}_{*,k} = \boldsymbol{y}_{*,\pi(k)}$.*

With Lemmas 1 and 2 in tow, we are ready to present the main result of this section.

**Theorem 1.** *Under Assumptions 1 and 2, if $\eta_k \le \frac{1}{n\sqrt{2\hat{L}_{\pi(k)}^g \tilde{L}_{\pi(k)}^g}}$ and $H_K = \sum_{k=1}^K \eta_k$, the output $\hat{\boldsymbol{x}}_K$ of Algorithm 1 with uniformly random (RR/SO) shuffling satisfies*

$$\mathbb{E}[H_K(f(\hat{\boldsymbol{x}}_K) - f(\boldsymbol{x}_*))] \le \frac{1}{2n} \|\boldsymbol{x}_0 - \boldsymbol{x}_*\|_2^2 + \sum_{k=1}^K \frac{\eta_k^3 (n+1) \tilde{L}^g}{6} \sigma_*^2.$$

*As a consequence, for any $\epsilon > 0$, there exists a choice of a constant step size $\eta_k = \eta$ for which $\mathbb{E}[f(\hat{\boldsymbol{x}}_K) - f(\boldsymbol{x}_*)] \le \epsilon$ after $\mathcal{O}\big( \frac{n\sqrt{\hat{L}^g \tilde{L}^g} \|\boldsymbol{x}_0 - \boldsymbol{x}_*\|_2^2}{\epsilon} + \frac{\sqrt{n\tilde{L}^g} \sigma_* \|\boldsymbol{x}_0 - \boldsymbol{x}_*\|_2^2}{\epsilon^{3/2}} \big)$ individual gradient queries.*

## 3 Tighter Bounds for Convex Finite-Sum Problems with Linear Predictors

To study the effect of the structure of the data on the convergence of shuffled SGD, we sharpen the focus from general finite-sum problems to convex finite-sum with linear predictors:

$$\min_{\boldsymbol{x} \in \mathbb{R}^d} \Big\{ f(\boldsymbol{x}) := \frac{1}{n} \sum_{i=1}^n \ell_i(\boldsymbol{a}_i^\top \boldsymbol{x}) \Big\}, \tag{PL}$$

where $\boldsymbol{a}_i \in \mathbb{R}^d$ ($i \in [n]$) are data vectors and $\ell_i : \mathbb{R} \to \mathbb{R}$ are convex and either smooth or Lipschitz nonsmooth functions associated with the linear predictors $\langle \boldsymbol{a}_i, \boldsymbol{x} \rangle$ for $i \in [n]$. In addition to their explicit dependence on the data, it is worth noting that problems of the form (PL) cover most of the standard convex ERM problems where shuffled SGD is commonly applied, such as support vector machines, least absolute deviation, least squares, and logistic regression.

Problem (PL) admits an explicit primal-dual formulation using the standard Fenchel conjugacy argument (see, e.g., [13, 14]),

$$\min_{\boldsymbol{x} \in \mathbb{R}^d} \max_{\boldsymbol{y} \in \mathbb{R}^n} \Big\{ \mathcal{L}(\boldsymbol{x}, \boldsymbol{y}) := \frac{1}{n} \langle \boldsymbol{A}\boldsymbol{x}, \boldsymbol{y} \rangle - \frac{1}{n} \sum_{i=1}^n \ell_i^*(\boldsymbol{y}^i) = \frac{1}{n} \sum_{i=1}^n \big( \boldsymbol{a}_i^\top \boldsymbol{x}\boldsymbol{y}^i - \ell_i^*(\boldsymbol{y}^i) \big) \Big\}, \tag{PL-PD}$$

where $\boldsymbol{A} = [\boldsymbol{a}_1, \boldsymbol{a}_2, \ldots, \boldsymbol{a}_n]^\top \in \mathbb{R}^{n \times d}$ is the data matrix and $\ell_i^* : \mathbb{R} \to \mathbb{R}$ is the convex conjugate of $\ell_i$. This observation allows us to again interpret without-replacement SGD updates as cyclic

coordinate updates on the dual side. Note that due to the objective structure in (PL), the primal-dual formulation (PL-PD) can decouple the linear ($\boldsymbol{a}_i^\top \boldsymbol{x}$) and the non-linear ($\ell_i$) parts within individual loss functions $f_i$. We redefine the conjugate pair of $\boldsymbol{x} \in \mathbb{R}^d$ to be $\boldsymbol{y_x} = (y_{\boldsymbol{x}}^1, \ldots, y_{\boldsymbol{x}}^n)^\top \in \mathbb{R}^n$, with $\boldsymbol{y}_{\boldsymbol{x}}^i = \arg\max_{\boldsymbol{y}^i \in \mathbb{R}} \{\boldsymbol{y}^i \boldsymbol{a}_i^\top \boldsymbol{x} - \ell_i^*(\boldsymbol{y}^i)\}$.

In this section, we consider shuffled SGD with *mini-batch* estimators of size $b$ and assume without loss of generality that $n = bm$ for some positive integer $m$. The detailed primal-dual view of shuffled SGD adapted to (PL-PD) and mini-batch estimators is provided in Alg. 2 in Appendix C.

## 3.1 Smooth and convex objectives

Throughout this subsection, we make the following (standard) assumptions, corresponding to Assumptions 1 and 2 from Section 2.

**Assumption 3.** *Each $\ell_i$ is convex and $L_i$-smooth ($i \in [n]$), i.e., $|\ell_i'(x) - \ell_i'(y)| \leq L_i |x - y|$ for any $x, y \in \mathbb{R}$. There exists a minimizer $\boldsymbol{x}_* \in \arg\min_{\boldsymbol{x} \in \mathbb{R}^d} f(\boldsymbol{x})$.*

We remark that Assumption 3 implies that both $f$ and each component function $f_i(\boldsymbol{x}) = \ell_i(\boldsymbol{a}_i^\top \boldsymbol{x})$ are $L_{\max}$-smooth, where $L_{\max} = \max_{i \in [n]} L_i \|\boldsymbol{a}_i\|_2^2$. Assumption 3 also implies that each convex conjugate $\ell_i^*$ is $\frac{1}{L_i}$-strongly convex [3]. In the following, we let $\boldsymbol{\Lambda} = \text{diag}(L_1, L_2, \ldots, L_n)$, and $\boldsymbol{\Lambda}_\pi = \text{diag}(L_{\pi^1}, L_{\pi^2}, \ldots, L_{\pi^n})$, given a permutation $\pi$ of $[n]$.

We further assume bounded variance at $\boldsymbol{x}_*$, same as prior work [30, 34, 45, 46].

**Assumption 4.** $\sigma_*^2 := \frac{1}{n} \sum_{i=1}^n \|\nabla f_i(\boldsymbol{x}_*)\|^2 = \frac{1}{n} \sum_{i=1}^n (\ell_i'(\boldsymbol{a}_i^\top \boldsymbol{x}_*))^2 \|\boldsymbol{a}_i\|_2^2$ *is bounded.*

**Improved bounds with new smoothness constants.** Our convergence bounds depend on the smoothness parameters defined in Eq. (3) below. We provide a detailed discussion on how these parameters relate to traditional smoothness parameters both in the worst case and on typical datasets, in Section 4.1, with additional numerical results provided in Appendix E.

$$
\begin{aligned}
\hat{L}_\pi &:= \frac{1}{mn} \big\| \boldsymbol{\Lambda}_\pi^{1/2} \big( \textstyle\sum_{j=1}^m \boldsymbol{I}_{b(j-1)\uparrow} \boldsymbol{A}_\pi \boldsymbol{A}_\pi^\top \boldsymbol{I}_{b(j-1)\uparrow} \big) \boldsymbol{\Lambda}_\pi^{1/2} \big\|_2, \qquad &\hat{L} = \max_\pi \hat{L}_\pi, \\
\tilde{L}_\pi &:= \frac{1}{b} \big\| \boldsymbol{\Lambda}_\pi^{1/2} \big( \textstyle\sum_{j=1}^m \boldsymbol{I}_{(j)} \boldsymbol{A}_\pi \boldsymbol{A}_\pi^\top \boldsymbol{I}_{(j)} \big) \boldsymbol{\Lambda}_\pi^{1/2} \big\|_2, \qquad &\tilde{L} = \max_\pi \tilde{L}_\pi,
\end{aligned}
\tag{3}
$$

where $\boldsymbol{I}_{(j)} := \sum_{i=b(j-1)+1}^{bj} \boldsymbol{I}_i$. In comparison to the smoothness constants defined in Eq. (2) for general finite-sum problems, we note that the constants in Eq. (3) applying to generalized linear models are tighter and more informative estimates, as the data matrix $\boldsymbol{A}$ and the smoothness constants from the nonlinear part $\boldsymbol{\Lambda}$ are separated in Eq. (3). Thus, the constants $\hat{L}_\pi$ and $\tilde{L}_\pi$ directly depend on the data matrix, which explicitly demonstrates how the structure of the data affects the convergence of shuffled SGD. The following theorem states the convergence of Algorithm 2 with these new refined smoothness constants, while its proof is provided in Appendix C.

**Theorem 2.** *Under Assumptions 3 and 4, if $\eta_k \leq \frac{b}{n\sqrt{2\hat{L}_{\pi(k)}\tilde{L}_{\pi(k)}}}$ and $H_K = \sum_{k=1}^K \eta_k$, then the output $\hat{\boldsymbol{x}}_K$ of Alg. 1 with uniformly random (RR/SO) shuffling satisfies*

$$
\mathbb{E}[H_K(f(\hat{\boldsymbol{x}}_K) - f(\boldsymbol{x}_*))] \leq \frac{b}{2n} \|\boldsymbol{x}_0 - \boldsymbol{x}_*\|_2^2 + \sum_{k=1}^K \frac{\eta_k^3 \tilde{L}(n-b)(n+b)}{6b^2(n-1)} \sigma_*^2.
$$

*As a result, given $\epsilon > 0$, there exists a constant step size $\eta_k = \eta$ such that $\mathbb{E}[f(\hat{\boldsymbol{x}}_K) - f(\boldsymbol{x}_*)] \leq \epsilon$ after $\mathcal{O}\big(\frac{n\sqrt{\hat{L}\tilde{L}}\|\boldsymbol{x}_0 - \boldsymbol{x}_*\|_2^2}{\epsilon} + \sqrt{\frac{(n-b)(n+b)}{n(n-1)}} \frac{\sqrt{n\tilde{L}}\sigma_*\|\boldsymbol{x}_0 - \boldsymbol{x}_*\|_2^2}{\epsilon^{3/2}}\big)$ individual gradient queries.*

A few remarks are in order here. When $b = n$, we recover the standard guarantee of gradient descent, which serves as a sanity check as in this case the algorithm reduces to standard gradient descent. When $\epsilon = \Omega\big(\frac{(n-b)(n+b)\sigma_*^2}{n^2(n-1)\hat{L}}\big)$, the resulting complexity is $\mathcal{O}\big(\frac{n\sqrt{\hat{L}\tilde{L}}\|\boldsymbol{x}_0 - \boldsymbol{x}_*\|_2^2}{\epsilon}\big)$. Observe that this case can happen when either $\epsilon$ is large (compared to, say, $1/n$) or when $\sigma_*$ is small (it is, in fact, possible for $\sigma_*$ to be zero, which happens, for example, when the data rows are linearly independent). Unlike

in bounds from previous work, we observe from our bounds the benefit of using shuffled SGD compared to full gradient descent, where the difference is by a factor that can be as large as $\sqrt{n}$, as we have discussed in the introduction (see also Section 4). When $\epsilon = \mathcal{O}(\frac{(n-b)(n+b)\sigma_*^2}{n^2(n-1)\hat{L}})$, the second term in our complexity bound dominates. In this case, when $b = 1$, we recover the state of the art results from [12, 30, 34], while for $b > 1$ our bound provides the $\Omega\big(\sqrt{\frac{n(n-1)}{(n-b)(n+b)} \cdot \frac{L}{\hat{L}}}\big)$-factor improvement, providing insights into benefits from the mini-batching strategy commonly used in practice.

## 3.2 Extension to non-smooth convex objectives

In non-smooth settings, we make the following standard assumption.

**Assumption 5.** *Each $\ell_i$ is convex and $G_i$-Lipschitz $(i \in [n])$, i.e., $|\ell_i(x) - \ell_i(y)| \leq G_i|x - y|$ for any $x, y \in \mathbb{R}$; thus $|g_i(x)| \leq G_i$ where $g_i(x) \in \partial\ell_i(x)$. There exists a minimizer $\boldsymbol{x}_* \in \arg\min_{\boldsymbol{x}\in\mathbb{R}^d} f(\boldsymbol{x})$.*

If Assumption 5 holds, each $\ell_i(\boldsymbol{a}_i^\top \boldsymbol{x})$ is also $G_{\max}$-Lipschitz with respect to $\boldsymbol{x}$, where $G_{\max} = \max_{i\in[n]} G_i\|\boldsymbol{a}_i\|_2$. To state our results, we define $\boldsymbol{\Gamma} := \mathrm{diag}(G_1^2, G_2^2, \ldots, G_n^2)$ and $\boldsymbol{\Gamma}_\pi = \mathrm{diag}\big(G_{\pi_1}^2, G_{\pi_2}^2, \ldots, G_{\pi_n}^2\big)$, given a data permutation $\pi$ of $[n]$.

We now extend our analysis of Algorithm 1 to convex nonsmooth Lipschitz settings, where the conjugate functions $\ell_i^*(y^i)$ are only convex. Proceeding as in Lemma 1, we obtain a bound on the primal-dual gap similar to (1), but lose two retraction terms induced by smoothness. Instead of cancelling the corresponding error terms like in the smooth case, we rely on the boundedness of the subgradients to bound these terms under a sufficiently small step size, which is common in nonsmooth Lipschitz settings. Similar to Section 2, we introduce the following quantities to obtain a tighter guarantee with respect to the data matrix and Lipschitz constants

$$\hat{G}_\pi := \frac{1}{mn}\big\|\boldsymbol{\Gamma}_\pi^{1/2}\big(\textstyle\sum_{j=1}^m \boldsymbol{I}_{b(j-1)\uparrow}\boldsymbol{A}_\pi\boldsymbol{A}_\pi^\top\boldsymbol{I}_{b(j-1)\uparrow}\big)\boldsymbol{\Gamma}_\pi^{1/2}\big\|_2,$$

$$\tilde{G}_\pi := \frac{1}{b}\big\|\boldsymbol{\Gamma}_\pi^{1/2}\big(\textstyle\sum_{j=1}^m \boldsymbol{I}_{(j)}\boldsymbol{A}_\pi\boldsymbol{A}_\pi^\top\boldsymbol{I}_{(j)}\big)\boldsymbol{\Gamma}_\pi^{1/2}\big\|_2.$$

We discuss the improvements in convergence from $\hat{G}_\pi$ and $\tilde{G}_\pi$ in Section 4, while the convergence of Algorithm 2 is described in Theorem 3, with its proof deferred to Appendix D.

**Theorem 3.** *Under Assumption 5, if $H_K = \sum_{k=1}^K \eta_k$ and $\bar{G} = \mathbb{E}_\pi[\sqrt{\hat{G}_\pi\tilde{G}_\pi}]$, the output $\hat{\boldsymbol{x}}_K$ of Alg. 1 with possible uniformly random shuffling satisfies*

$$\mathbb{E}[H_K(f(\hat{\boldsymbol{x}}_K) - f(\boldsymbol{x}_*))] \leq \frac{1}{2n}\|\boldsymbol{x}_0 - \boldsymbol{x}_*\|_2^2 + \sum_{k=1}^K 2\eta_k^2 n\bar{G},$$

*As a result, for any $\epsilon > 0$, there exists a step size $\eta_k = \eta$ such that $\mathbb{E}[f(\hat{\boldsymbol{x}}_K) - f(\boldsymbol{x}_*)] \leq \epsilon$ after $\mathcal{O}\big(\frac{n\bar{G}\|\boldsymbol{x}_0-\boldsymbol{x}_*\|_2^2}{\epsilon^2}\big)$ individual gradient queries.*

## 4 Discussion of Our New Smoothness Constants and Numerical Results

To succinctly explain where our improvements come from, we now consider (PL) where $\ell_i$ is 1-smooth and $b = 1$, ignoring the gains from the mini-batch estimators (for large $K$) and our softer guarantee that handles individual smoothness constants. For this specific case, $\hat{L} = L_{\max} = \max_{1\leq i\leq n}\|\boldsymbol{a}_i\|^2$, and thus our results for the smooth case and the RR and SO variants match state of the art in the second term, which dominates when there are many ($K = \Omega(\frac{L_{\max}^2 D^2 n}{\sigma_*^2})$) epochs. When there are $K = O(\frac{L_{\max}^2 D^2 n}{\sigma_*^2})$ epochs in the SO and RR variants or for all regimes of $K$ in the IG variant, the difference between our and state of the art bounds comes from the constant $\hat{L}$ that replaces $L_{\max}$, and our improvement is by a factor $\sqrt{L_{\max}/\hat{L}}$. Note that $\mathcal{O}(\frac{nL_{\max}}{\epsilon})$ from prior bounds, which is the dominating term in the small $K$ regime, is even worse than the complexity of full gradient descent, as the full gradient Lipschitz constant of $f$ in this case is $\frac{1}{n}\|\boldsymbol{A}\boldsymbol{A}^\top\|_2 \leq L_{\max}$.

Given a worst-case permutation $\bar{\pi}$, and denoting by $\boldsymbol{A}_{\bar{\pi}}$ the data matrix $\boldsymbol{A}$ with its rows permuted according to $\bar{\pi}$, our constant $\hat{L}$ can be bounded above by $L_{\max}$ using the following sequence of inequalities:

$$
\begin{aligned}
\hat{L} = \frac{1}{n^2} \| \sum_{j=1}^{n} \boldsymbol{I}_{(j-1)\uparrow} \boldsymbol{A}_{\bar{\pi}} \boldsymbol{A}_{\bar{\pi}}^{\top} \boldsymbol{I}_{(j-1)\uparrow} \|_2 & \overset{(i)}{\leq} \frac{1}{n^2} \sum_{j=1}^{n} \| \boldsymbol{I}_{(j-1)\uparrow} \boldsymbol{A}_{\bar{\pi}} \boldsymbol{A}_{\bar{\pi}}^{\top} \boldsymbol{I}_{(j-1)\uparrow} \|_2 \\
& \overset{(ii)}{\leq} \frac{1}{n^2} \sum_{j=1}^{n} \| \boldsymbol{A}_{\bar{\pi}} \boldsymbol{A}_{\bar{\pi}}^{\top} \|_2 \\
& \overset{(iii)}{\leq} \frac{1}{n} \sum_{i=1}^{n} \| \boldsymbol{a}_i \|_2^2 \leq \max_{1 \leq i \leq n} \| \boldsymbol{a}_i \|_2^2 = L_{\max},
\end{aligned}
\tag{4}
$$

where $(i)$ holds by the triangle inequality, $(ii)$ holds because the operator norm of the matrix $\boldsymbol{I}_{(j-1)\uparrow} \boldsymbol{A}_{\pi} \boldsymbol{A}_{\pi}^{\top} \boldsymbol{I}_{(j-1)\uparrow}$ (equal to the operator norm of the bottom right $(n-j+1) \times (n-j+1)$ submatrix of $\boldsymbol{A}_{\pi} \boldsymbol{A}_{\pi}^{\top}$) is always at most $\| \boldsymbol{A}_{\pi} \boldsymbol{A}_{\pi}^{\top} \| = \| \boldsymbol{A} \boldsymbol{A}^{\top} \|$, for any permutation $\pi$, and $(iii)$ holds by bounding above the operator norm of a symmetric matrix by its trace. Hence $\hat{L}$ is never larger than $L_{\max}$, but can generally be much smaller, due to the sequence of inequality relaxations in (4). While each of these inequalities can be loose, we emphasize that $(iii)$ is almost always loose, by a factor that can be as large as $n$.

As a specific example where $\hat{L}$ is smaller than $L_{\max}$ by a factor of $n$, consider the example of Gaussian data, where we draw $n$ i.i.d. standard Gaussian vectors from $\mathcal{N}(\boldsymbol{0}, \mathbf{I}_d)$ and take $d = n$. By standard concentration results, with high probability, all columns/rows of $\mathbf{A}_{\bar{\pi}}$ in this case are near-orthogonal (see, e.g., [7, Chapter]) and $\| \boldsymbol{a}_i \|_2^2 \approx d = n$ for all $i$. As a result, the operator norm to trace inequality $(iii)$ is loose by a factor $d = n$, with high probability. Note that in this example all individual smoothness parameters of components $f_i$ are essentially the same (w.h.p.) and equal $\| \boldsymbol{a}_i \|_2^2$, thus the improvement of our bound on the smoothness parameter does not come from averaging but from the structure of the data. This observation is important for contrasting the results from Section 2 and Section 3. In particular, focusing solely on the finite sum structure and ignoring the structure of the data matrix would provide no improvements in the resulting convergence bounds.

As further evidence, we empirically evaluate $L_{\max}/\hat{L}$ on 15 large-scale machine learning datasets and demonstrate that on those datasets $L_{\max}/\hat{L}$ is of the order $n^{\alpha}$, for $\alpha \in [0.15, 0.96]$ (see Sec. 4.1 for more details), providing strong evidence of a tighter guarantee as a function of $n$.

For the nonsmooth settings, by a similar sequence of inequalities, we can show that $\bar{G} \leq G_{\max}^2$, which can be loose by a factor $1/n$ due to the operator norm to trace inequality. Thus, our bound is never worse than what would be obtained from the full subgradient method, but can match the bound of standard SGD, or even improve[1] upon it for at least some data matrices $\boldsymbol{A}$.

## 4.1 Numerical results and discussion

In this section, we provide empirical evidence to support our claim about usefulness of the new convergence bounds obtained in our work. In particular, we conduct numerical evaluations to compare $\hat{L}$ to the classical smoothness constant $L$ on synthetic datasets and on popular machine learning benchmark datasets.

For a more streamlined comparison and to focus on the dependence on the data matrix, we assume that the loss functions $\ell_i$ all have the same smoothness constant, which leads to $L_{\max}/\hat{L} = (\max_{1 \leq i \leq n} \{ \| \boldsymbol{a}_i \|^2 \}) / (\frac{1}{n^2} \| \sum_{j=1}^{n} \boldsymbol{I}_{(j-1)\uparrow} \boldsymbol{A}_{\bar{\pi}} \boldsymbol{A}_{\bar{\pi}}^{\top} \boldsymbol{I}_{(j-1)\uparrow} \|_2)$. Since the scale of the smoothness constant of the loss functions is irrelevant for the ratio $L_{\max}/\hat{L}$ in this case, for simplicity, we take it to equal one. Note that assuming different smoothness constants over component loss functions would only make our bound better compared to related work (see Eq. (3) and the discussion following it).

We also compare $\hat{L}$ and $L_{\max}$ on a number of benchmarking datasets from LIBSVM [15], MNIST [17], CIFAR10 [22], and Broad Bioimage Benchmark Collection [28]. For each dataset, we generate a uniformly random permutation $\pi$ for the data matrix $\boldsymbol{A}$ and compute $\hat{L}_{\pi}$. We repeat this procedure 1000 times for all datasets and display the average $L_{\max}/\hat{L}_{\pi}$ in Table 2, except for e2006train, CIFAR10, MNIST, and BBBC005 where we do 20 repetitions due to limitations of

---

[1]This is because it is possible for inequalities $(i)$ and $(ii)$ to be loose, in addition to $(iii)$.

Table 2: The following table shows the computed values of $L_{\max}/\hat{L}$ where $\hat{L}$ is the empirical mean of $\hat{L}_\pi$ over random permutations. We note that the quantity $\sqrt{L_{\max}/\hat{L}}$ represents the improvement provided by the bound via our novel primal-dual perspective, compared to previous work.

| DATASET | #FEATURES ($d$) | #DATAPOINTS ($n$) | $L_{\max}/\hat{L}$ | $\log_n L_{\max}/\hat{L}$ | $\log_{\min(d,n)} L_{\max}/\hat{L}$ |
|---|---|---|---|---|---|
| A1A | 123 | 1605 | 5.50 | 0.231 | 0.354 |
| A9A | 123 | 32561 | 5.49 | 0.164 | 0.354 |
| BBBC005 | 361920 | 19201 | 18.3 | 0.295 | 0.295 |
| BBBC010 | 361920 | 201 | 7.04 | 0.368 | 0.368 |
| CIFAR10 | 3072 | 50000 | 10.0 | 0.213 | 0.287 |
| DUKE | 7129 | 44 | 38.0 | 0.962 | 0.962 |
| E2006TRAIN | 150360 | 16087 | 5.35 | 0.173 | 0.173 |
| GISETTE | 5000 | 6000 | 3.52 | 0.145 | 0.148 |
| LEU | 7129 | 38 | 32.8 | 0.960 | 0.960 |
| MNIST | 780 | 60000 | 19.1 | 0.268 | 0.443 |
| NEWS20 | 1355191 | 19996 | 42.1 | 0.378 | 0.378 |
| RCV1 | 47236 | 20242 | 111 | 0.475 | 0.475 |
| REAL-SIM | 20958 | 72309 | 194 | 0.471 | 0.529 |
| SONAR | 60 | 208 | 6.26 | 0.344 | 0.448 |
| TMC2007 | 30438 | 21519 | 10.9 | 0.239 | 0.239 |

computation resources required for each calculation. We observe that among the datasets that we consider, which contain all three data matrix "shapes" $d >> n$, $d << n$, and $d \approx n$, our novel bound dependent on $\hat{L}$ is much tighter. For instance, for `rcv1` and `real-sim` datasets, where $d$ and $n$ are of the same order, we observe that $L_{\max}/\hat{L}$ are approximately 111 and 194, respectively. For `news20` dataset where $d >> n$, $L_{\max}/\hat{L} \approx 42.1$. For `MNIST`, where $d << n$, $L_{\max}/\hat{L} \approx 19.1$. Further results are provided in Appendix E.

Finally, as a justification for using the empirical mean of $\hat{L}_\pi$ over random permutations $\pi$ in the results displayed in Table 2, we observe in our evaluations that the values of $L_{\max}/\hat{L}_\pi$ are fairly concentrated around their empirical mean values. Histogram plots showing the empirical distributions of $L_{\max}/\hat{L}_\pi$ for each of the datasets are provided in Appendix E.

We conclude with a few additional remarks. Our results indicate that the structure of the data is important for predicting behavior of popular machine learning methods such as variants of shuffled SGD considered in our work, and thus should be incorporated in their study: as demonstrated in the Gaussian data example, considering simple finite sum structure and ignoring the dependence on the data can lead to overly pessimistic bounds. Thus it would be interesting to provide a further theoretical study of shuffled SGD that incorporates distributional assumptions for the data. Additionally, as mentioned in the previous paragraph, we empirically observed that permutation-dependent parameter $\hat{L}_\pi$ concentrates around its mean for permutations generated uniformly at random. Thus, it would be interesting to consider whether our theoretical results can be strengthened to depend on the mean value of $\hat{L}_\pi$ (as opposed to maximum). We leave such considerations for future work.

## Acknowledgements

This research was supported in part by the U.S. Office of Naval Research under contract number N00014-22-1-2348.

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

# Supplementary Material

**Outline.** The supplementary material of the paper is organized as follows:

- Section A provides a brief survey on shuffled SGD and its related work.
- Section B presents the proofs related to the smooth convex setting from Section 2, where we only assume each component function $f_i$ to be convex and $L_i$-smooth.
- Section C presents the proofs related to the smooth convex setting with linear predictors from Section 3.
- Section D presents the proofs related to the non-smooth convex setting with linear predictors from Section 3.
- Section E presents the full details of the computational experiments performed in the paper.

## A  Further Related Work

In this section, we continue with the discussion on the background of shuffled SGD from Section 1. We would like to briefly recall that shuffled SGD usually performs better in practice when compared to SGD, and is also easier and more efficient to implement. However, in terms of the theoretical analysis, sampling without replacement introduces the sampling bias at each iteration, making it difficult to approximate shuffled SGD by full gradient descent. Using empirical observations, shuffled SGD was conjectured to converge much faster than SGD with replacement, based on the *noncommutative arithmetic-geometric mean inequality* conjecture [36], which was later proved to be false [16, 23]. As a consequence, whether or not shuffled SGD can be faster than SGD at least in some regimes remained open [8] until a breakthrough result in [20], where it was shown that for the class of smooth strongly convex optimization problems, the convergence of the RR variant of shuffled SGD is essentially of the order-$(1/K^2)$ for $K$ full passes of the data (also called epochs), which is faster than order-$(1/nK)$ convergence of SGD for sufficiently large $K$. This bound for the smooth strongly convex case was later improved under various regimes and additional assumptions [2, 21, 30, 31, 34, 42], while the tightest of those bounds were matched by lower bounds in [12, 35, 39, 51].

Since our results are for the general (non-strongly) convex regimes, in this section we focus on the results that apply to those (convex, smooth or nonsmooth Lipschitz) regimes. For convex nonsmooth Lipschitz problems, we are only aware of the results in [42]. These results are only useful when the number of data passes $K$ is small and the number of component functions $n$ is large, as they contain an irreducible order-$\frac{1}{\sqrt{n}}$ error, and are not directly comparable to our results.

For the IG variant of SGD without replacement (deterministic order), asymptotic convergence was established in [6, 29], with further convergence results for both smooth and nonsmooth settings provided in [18, 25, 30, 32, 34, 50]. As IG does not benefit from randomization, it is known to have a worse convergence bound than RR under the Lipschitz Hessian assumption [18, 21], which was also shown in more general settings [30].

In this paper, we viewed shuffled SGD as a primal-dual method where the updates are performed on the dual side in a cyclic manner, thus we can leverage techniques from general cyclic methods. However, in contrast to randomized methods (corresponding to standard SGD), cyclic methods are usually more challenging to analyze [33], basic variants exhibit much worse *worst-case* complexity than even full gradient methods [4, 19, 26, 26, 40, 44, 48, 49], with more refined results being established only recently [11, 27, 43]. While the inspiration for our work came from these recent results [11, 27, 43], they are completely technically disjoint. First, all these results rely on non-standard block Lipschitz assumptions, which are not present in our work. Second, all of them leverage proximal gradient-style cyclic updates to carry out the analysis, which is inapplicable in our case for the cyclic updates on the dual side, as otherwise the method would not correspond to (shuffled) SGD. Finally, [27, 43] utilize extrapolation steps, which would break the connection to shuffled SGD in our setting, while [11] relies on a gradient descent-type descent lemma, which is impossible to establish in our setting.

## B Omitted Proofs From Section 2

In this section, we consider the general finite-sum setting where we assume that each component function $f_i$ is convex and smooth, and derive the refined analysis under this setting. Here we focus on the smooth convex problems as prior work did [30, 34], since smoothness is essential to showing the advantage of shuffled SGD [31] over SGD, otherwise the rate of SGD is optimal. In particular, we study the general smooth convex finite-sum problem (P)

$$\min_{\boldsymbol{x}\in\mathbb{R}^d}\Big\{f(\boldsymbol{x}) := \frac{1}{n}\sum_{i=1}^{n} f_i(\boldsymbol{x})\Big\}, \tag{P}$$

where each $f_i$ is convex and smooth. (P) is equivalent to

$$\min_{\boldsymbol{x}\in\mathbb{R}^d}\max_{\boldsymbol{y}\in\mathbb{R}^{nd}}\Big\{\mathcal{L}(\boldsymbol{x},\boldsymbol{y}) := \frac{1}{n}\sum_{i=1}^{n}\Big(\langle\boldsymbol{y}^i,\boldsymbol{x}\rangle - f_i^*(\boldsymbol{y}^i)\Big) = \frac{1}{n}\langle\boldsymbol{E}\boldsymbol{x},\boldsymbol{y}\rangle - \frac{1}{n}\sum_{i=1}^{n}f_i^*(\boldsymbol{y}^i)\Big\}, \tag{PD}$$

where we slightly abuse the notation in this section and use $\boldsymbol{y}^i \in \mathbb{R}^d$ to be the $i$-th $d$ elements of the vector $\boldsymbol{y}$ such that $\boldsymbol{y} = (\boldsymbol{y}^1,\dots,\boldsymbol{y}^n)^\top \in \mathbb{R}^{nd}$, $\boldsymbol{E} = [\underbrace{\boldsymbol{I}_d,\dots,\boldsymbol{I}_d}_{n}]^\top \in \mathbb{R}^{nd\times d}$ is the vertical concatenation of $n$ identity matrices $\boldsymbol{I}_d \in \mathbb{R}^{d\times d}$ and $f_i^*$ is the convex conjugate of $f_i$ defined by $f_i(\boldsymbol{x}) = \sup_{\boldsymbol{y}^i\in\mathbb{R}^d}\langle\boldsymbol{y}^i,\boldsymbol{x}\rangle - f_i^*(\boldsymbol{y}^i)$. In the following, we consider the mini-batch estimator of batch size $b$, and let $\boldsymbol{y}^{(i)} \in \mathbb{R}^{bd}$ denote the vector comprised of the $i^{\text{th}}$ $bd$ elements of $\boldsymbol{y}$. For simplicity and without loss of generality, we assume that $n = bm$ for some positive integer $m$, so that $\boldsymbol{y} = (\boldsymbol{y}^{(1)},\dots,\boldsymbol{y}^{(m)})^\top$. Note that if choosing $b = 1$, our setting is the same as the ones in [30, 34]. Then we have the primal-dual view of shuffled SGD scheme for general smooth convex minimization as in Alg. 1, where $\boldsymbol{E}_b^\top = [\underbrace{\boldsymbol{I}_d,\dots,\boldsymbol{I}_d}_{b}]^\top \in \mathbb{R}^{bd\times d}$ is the vertical concatenation of $b$ identity matrices $\boldsymbol{I}_d \in \mathbb{R}^{d\times d}$. Given the data permutation $\pi^{(k)} = \{\pi_1^{(k)},\pi_2^{(k)},\dots,\pi_n^{(k)}\}$ of $[n]$ at the $k$-th epoch, we use the same notation of $\boldsymbol{v}_k = (\boldsymbol{v}^{\pi_1^{(k)}},\dots,\boldsymbol{v}^{\pi_n^{(k)}})^\top \in \mathbb{R}^{nd}$, $\boldsymbol{y}_{*,k} = (\boldsymbol{y}_*^{\pi_1^{(k)}},\dots,\boldsymbol{y}_*^{\pi_n^{(k)}})^\top \in \mathbb{R}^{nd}$ as in previous sections except now each $\boldsymbol{v}^{\pi_i^{(k)}}, \boldsymbol{y}_*^{\pi_i^{(k)}}$ are $d$-dimensional subvectors. Further, we denote the permuted smoothness constant matrices by $\boldsymbol{\Lambda}_k = \mathrm{diag}(\underbrace{L_{\pi_1^{(k)}},\dots,L_{\pi_1^{(k)}}}_{d},\dots,\underbrace{L_{\pi_n^{(k)}},\dots,L_{\pi_n^{(k)}}}_{d}) \in \mathbb{R}^{nd\times nd}$, and we use $\boldsymbol{I}$ for $\boldsymbol{I}_{nd} \in \mathbb{R}^{nd\times nd}$ throughout this section.

**New smoothness constants and comparisons.** We first recall the new smoothness constants for any permutation $\pi$ of $[n]$, defined in Eq. (2):

$$\hat{L}_\pi^g := \frac{1}{mn}\big\|\boldsymbol{\Lambda}_\pi^{1/2}\big(\textstyle\sum_{i=1}^{m}\boldsymbol{I}_{bd(i-1)\uparrow}\boldsymbol{E}\boldsymbol{E}^\top\boldsymbol{I}_{bd(i-1)\uparrow}\big)\boldsymbol{\Lambda}_\pi^{1/2}\big\|_2, \qquad \hat{L}^g = \max_\pi \hat{L}_\pi^g,$$

$$\tilde{L}_\pi^g := \frac{1}{b}\big\|\boldsymbol{\Lambda}_\pi^{1/2}\big(\textstyle\sum_{i=1}^{m}\boldsymbol{I}_{(di)}\boldsymbol{E}\boldsymbol{E}^\top\boldsymbol{I}_{(di)}\big)\boldsymbol{\Lambda}_\pi^{1/2}\big\|_2, \qquad \tilde{L}^g = \max_\pi \tilde{L}_\pi^g,$$

where $\boldsymbol{I}_{(di)} = \sum_{j=bd(i-1)+1}^{bdi}\boldsymbol{I}_j$.

To compare $\hat{L}_\pi^g$ and $L := \max_{i\in[n]} L_i$, we make use of the Kronecker product with notation $\otimes$ defined by

$$\boldsymbol{A}\otimes\boldsymbol{B} = \begin{bmatrix} A_{11}\boldsymbol{B} & \cdots & A_{1n}\boldsymbol{B} \\ \vdots & \ddots & \vdots \\ A_{m1}\boldsymbol{B} & \cdots & A_{nn}\boldsymbol{B} \end{bmatrix}$$

for two matrices $\boldsymbol{A} \in \mathbb{R}^{m\times n}$ and $\boldsymbol{B} \in \mathbb{R}^{p\times q}$. The following lemma states a useful fact for the Kronecker product.

**Lemma 3.** *For square matrices $\boldsymbol{A}$ and $\boldsymbol{B}$ of sizes $p$ and $q$ and with eigenvalues $\lambda_i$ ($i \in [p]$) and $\mu_j$ ($j \in [q]$) respectively, the eigenvalues of $\boldsymbol{A}\otimes\boldsymbol{B}$ are $\lambda_i\mu_j$ for $i \in [p], j \in [q]$.*

We now use the following chain of inequalities to compare $\hat{L}_\pi^g$ and $L$ for any permutation $\pi$ of $[n]$:

$$\hat{L}_\pi^g = \frac{1}{mn}\left\|\boldsymbol{\Lambda}_\pi^{1/2}\Big(\sum_{i=1}^m \boldsymbol{I}_{bd(i-1)\uparrow}\boldsymbol{E}\boldsymbol{E}^\top \boldsymbol{I}_{bd(i-1)\uparrow}\Big)\boldsymbol{\Lambda}_\pi^{1/2}\right\|_2$$

$$\le \frac{1}{n}\left\|\boldsymbol{\Lambda}^{1/2}\boldsymbol{E}\boldsymbol{E}^\top\boldsymbol{\Lambda}^{1/2}\right\|_2$$

$$= \frac{1}{n}\left\|(\boldsymbol{l}_\pi \boldsymbol{l}_\pi^\top)\otimes \boldsymbol{I}_d\right\|_2$$

$$\overset{(i)}{=} \frac{1}{n}\sum_{i=1}^n L_i \le L,$$

where we define $\boldsymbol{l}_\pi = \left(\sqrt{L_{\pi_1}}, \sqrt{L_{\pi_2}}, \ldots, \sqrt{L_{\pi_n}}\right)^\top$. For $(i)$, we use Lemma 3 and notice that the eigenvalues of $\boldsymbol{I}_d$ all equal 1, while the largest eigenvalue of $\boldsymbol{l}_\pi \boldsymbol{l}_\pi^\top = \|\boldsymbol{l}\|_2^2 = \sum_{i=1}^n L_i$, so the operator norm of $(\boldsymbol{l}_k \boldsymbol{l}_k^\top)\otimes \boldsymbol{I}_d$ is $\sum_{i=1}^n L_i$.

To compare $\tilde{L}_\pi^g$ and $L$, we notice that

$$\boldsymbol{\Lambda}_\pi^{1/2}\big(\sum_{i=1}^m \boldsymbol{I}_{(di)}\boldsymbol{E}\boldsymbol{E}^\top\boldsymbol{I}_{(di)}\big)\boldsymbol{\Lambda}_\pi^{1/2} = \sum_{i=1}^m \boldsymbol{I}_{(di)}\boldsymbol{\Lambda}_\pi^{1/2}\boldsymbol{E}\boldsymbol{E}^\top\boldsymbol{\Lambda}_\pi^{1/2}\boldsymbol{I}_{(di)}$$

is a block diagonal matrix whose operator norm is the maximum of the operator norms over its diagonal block submatrices, so we have

$$\tilde{L}_\pi^g = \frac{1}{b}\max_{i\in[m]}\left\|\boldsymbol{I}_{(di)}\boldsymbol{\Lambda}_\pi^{1/2}\boldsymbol{E}\boldsymbol{E}^\top\boldsymbol{\Lambda}_\pi^{1/2}\boldsymbol{I}_{(di)}\right\|$$

$$= \frac{1}{b}\max_{i\in[m]}\left\|\boldsymbol{I}_{(di)}\big((\boldsymbol{l}_\pi \boldsymbol{l}_\pi^\top)\otimes \boldsymbol{I}_d\big)\boldsymbol{I}_{(di)}\right\|$$

$$\overset{(i)}{=} \max_{i\in[m]}\frac{1}{b}\sum_{j=1}^b L_{\pi_{b(i-1)+j}} \le L,$$

where for $(i)$ we use Lemma 3 for each submatrix $(\boldsymbol{l}_\pi^{(i)}\boldsymbol{l}_\pi^{(i)\top})\otimes \boldsymbol{I}_d$ and $\boldsymbol{l}_\pi^{(i)} = (0,\ldots,0,\sqrt{L_{\pi_{b(i-1)+1}}},\ldots,\sqrt{L_{\pi_{bi}}},0,\ldots,0)^\top$. Similar to the case of generalized linear models, the inequality is tight when $b = 1$ but can be loose for other values of $b$.

Before proceeding to the omitted proofs, we first state the following standard definitions and first-order characterization of strong convexity, for completeness.

**Definition 1.** *A function* $f : \mathbb{R}^d \to \mathbb{R}$ *is said to be* $\mu$-*strongly convex with parameter* $\mu > 0$, *if for any* $\boldsymbol{x}, \boldsymbol{y} \in \mathbb{R}^d$ *and any* $\lambda \in (0,1)$:

$$f(\lambda\boldsymbol{x} + (1-\lambda)\boldsymbol{y}) \le \lambda f(\boldsymbol{x}) + (1-\lambda)f(\boldsymbol{y}) - \frac{\mu}{2}\lambda(1-\lambda)\|\boldsymbol{x}-\boldsymbol{y}\|_2^2.$$

**Lemma 4.** *Let* $f : \mathbb{R}^d \to \mathbb{R}$ *be a continuous* $\mu$-*strongly convex function with* $\mu > 0$. *Then, for any* $\boldsymbol{x}, \boldsymbol{y} \in \mathbb{R}^d$:

$$f(\boldsymbol{y}) \ge f(\boldsymbol{x}) + \langle \boldsymbol{g}_{\boldsymbol{x}}, \boldsymbol{y} - \boldsymbol{x}\rangle + \frac{\mu}{2}\|\boldsymbol{x}-\boldsymbol{y}\|_2^2,$$

*where* $\boldsymbol{g}_{\boldsymbol{x}} \in \partial f(\boldsymbol{x})$, *and* $\partial f(\boldsymbol{x})$ *is the subdifferential of* $f$ *at* $\boldsymbol{x}$.

We also include the following lemma on the variance bound under without-replacement sampling, which is useful for our proof.

**Lemma 5.** *Let* $\mathcal{B}$ *be the set of* $|\mathcal{B}| = b$ *samples from* $[n]$, *drawn without replacement and uniformly at random. Then,* $\forall \boldsymbol{x} \in \mathbb{R}^d$,

$$\mathbb{E}_{\mathcal{B}}\Big[\big\|\frac{1}{b}\sum_{i\in\mathcal{B}}\nabla f_i(\boldsymbol{x}) - \nabla f(\boldsymbol{x})\big\|_2^2\Big] = \frac{n-b}{b(n-1)}\mathbb{E}_i\big[\|\nabla f_i(\boldsymbol{x}) - \nabla f(\boldsymbol{x})\|_2^2\big].$$

*Proof.* We first expand the square on the left-hand side, as follows

$$\mathbb{E}_{\mathcal{B}}\left[\left\|\frac{1}{b}\sum_{i\in\mathcal{B}}\nabla f_i(\boldsymbol{x}) - \nabla f(\boldsymbol{x})\right\|_2^2\right]$$

$$= \frac{1}{b^2}\mathbb{E}_{\mathcal{B}}\left[\sum_{i,i'\in\mathcal{B}}\langle\nabla f_i(\boldsymbol{x}) - \nabla f(\boldsymbol{x}), \nabla f_{i'}(\boldsymbol{x}) - \nabla f(\boldsymbol{x})\rangle\right]$$

$$= \frac{1}{b^2}\mathbb{E}_{\mathcal{B}}\left[\sum_{i,i'\in\mathcal{B},i\neq i'}\langle\nabla f_i(\boldsymbol{x}) - \nabla f(\boldsymbol{x}), \nabla f_{i'}(\boldsymbol{x}) - \nabla f(\boldsymbol{x})\rangle\right] + \frac{1}{b}\mathbb{E}_i\left[\|\nabla f_i(\boldsymbol{x}) - \nabla f(\boldsymbol{x})\|_2^2\right].$$

Since the batch $\mathcal{B}$ is sampled uniformly and without replacement from $[n]$, the probability that any pair $(i, i')$ from $[n]$ with $i \neq i'$ is in $\mathcal{B}$ is $\frac{b(b-1)}{n(n-1)}$. By the linearity of expectation, we have

$$\mathbb{E}_{\mathcal{B}}\left[\sum_{i,i'\in\mathcal{B},i\neq i'}\langle\nabla f_i(\boldsymbol{x}) - \nabla f(\boldsymbol{x}), \nabla f_{i'}(\boldsymbol{x}) - \nabla f(\boldsymbol{x})\rangle\right]$$

$$= \mathbb{E}_{\mathcal{B}}\left[\sum_{i,i'\in[n],i\neq i'}\mathbb{1}_{i,i'\in\mathcal{B}}\langle\nabla f_i(\boldsymbol{x}) - \nabla f(\boldsymbol{x}), \nabla f_{i'}(\boldsymbol{x}) - \nabla f(\boldsymbol{x})\rangle\right]$$

$$= \sum_{i,i'\in[n],i\neq i'}\mathbb{E}_{\mathcal{B}}\left[\mathbb{1}_{i,i'\in\mathcal{B}}\langle\nabla f_i(\boldsymbol{x}) - \nabla f(\boldsymbol{x}), \nabla f_{i'}(\boldsymbol{x}) - \nabla f(\boldsymbol{x})\rangle\right]$$

$$= \frac{b(b-1)}{n(n-1)}\sum_{i,i'\in[n],i\neq i'}\langle\nabla f_i(\boldsymbol{x}) - \nabla f(\boldsymbol{x}), \nabla f_{i'}(\boldsymbol{x}) - \nabla f(\boldsymbol{x})\rangle,$$

where $\mathbb{1}$ is the indicator function such that $\mathbb{1}_{i,i'\in\mathcal{B}} = 1$ if both $i, i' \in \mathcal{B}$ and is equal to zero otherwise. Hence, we obtain

$$\mathbb{E}_{\mathcal{B}}\left[\left\|\frac{1}{b}\sum_{i\in\mathcal{B}}\nabla f_i(\boldsymbol{x}) - \nabla f(\boldsymbol{x})\right\|^2\right]$$

$$= \frac{b-1}{bn(n-1)}\sum_{i,i'\in[n],i\neq i'}\langle\nabla f_i(\boldsymbol{x}) - \nabla f(\boldsymbol{x}), \nabla f_{i'}(\boldsymbol{x}) - \nabla f(\boldsymbol{x})\rangle + \frac{1}{b}\mathbb{E}_i\left[\|\nabla f_i(\boldsymbol{x}) - \nabla f(\boldsymbol{x})\|^2\right]$$

$$= \frac{b-1}{bn(n-1)}\sum_{i,i'\in[n]}\langle\nabla f_i(\boldsymbol{x}) - \nabla f(\boldsymbol{x}), \nabla f_{i'}(\boldsymbol{x}) - \nabla f(\boldsymbol{x})\rangle + \frac{n-b}{b(n-1)}\mathbb{E}_i\left[\|\nabla f_i(\boldsymbol{x}) - \nabla f(\boldsymbol{x})\|^2\right]$$

$$\stackrel{(i)}{=} \frac{n-b}{b(n-1)}\mathbb{E}_i\left[\|\nabla^j f_i(\boldsymbol{x}) - \nabla^j f(\boldsymbol{x})\|^2\right],$$

where $(i)$ is due to $f = \frac{1}{n}\sum_{i=1}^n f_i$ having the finite sum structure. □

Now we provide the omitted proofs from Section 2.

**Lemma 6.** *Under Assumption 1, for any $k \in [K]$, the iterates $\{\boldsymbol{y}_k^{(i)}\}_{i=1}^m$ and $\{\boldsymbol{x}_{k-1,i}\}_{i=1}^{m+1}$ generated by Algorithm 1 satisfy*

$$\mathcal{E}_k \leq \frac{\eta_k}{n}\sum_{i=1}^m\left\langle\boldsymbol{E}_b^\top\boldsymbol{y}_k^{(i)}, \boldsymbol{x}_k - \boldsymbol{x}_{k-1,i+1}\right\rangle + \frac{\eta_k}{n}\sum_{i=1}^m\left\langle\boldsymbol{E}_b^\top(\boldsymbol{v}_k^{(i)} - \boldsymbol{y}_k^{(i)}), \boldsymbol{x}_k - \boldsymbol{x}_{k-1,i}\right\rangle$$

$$- \frac{\eta_k}{2n}\|\boldsymbol{y}_k - \boldsymbol{v}_k\|_{\boldsymbol{\Lambda}_k^{-1}}^2 - \frac{\eta_k}{2n}\|\boldsymbol{y}_k - \boldsymbol{y}_{*,k}\|_{\boldsymbol{\Lambda}_k^{-1}}^2 - \frac{b}{2n}\sum_{i=1}^m\|\boldsymbol{x}_{k-1,i+1} - \boldsymbol{x}_{k-1,i}\|^2, \tag{5}$$

*where $\mathcal{E}_k := \eta_k\mathrm{Gap}^{\boldsymbol{v}}(\boldsymbol{x}_k, \boldsymbol{y}_*) + \frac{b}{2n}\|\boldsymbol{x}_* - \boldsymbol{x}_k\|_2^2 - \frac{b}{2n}\|\boldsymbol{x}_* - \boldsymbol{x}_{k-1}\|_2^2.$*

*Proof.* We first note that based on Line 6 of Alg. 1, we have

$$\left\langle\boldsymbol{E}_b^\top\boldsymbol{y}^{(i)}, \boldsymbol{x}_{k-1,i}\right\rangle - \sum_{j=1}^b f_{\pi_{b(i-1)+j}^{(k)}}^*(\boldsymbol{y}^j) = \sum_{j=1}^b\left(\langle\boldsymbol{y}^j, \boldsymbol{x}_{k-1,i}\rangle - f_{\pi_{b(i-1)+j}^{(k)}}^*(\boldsymbol{y}^j)\right).$$

Since the max problem defining $\boldsymbol{y}_k$ is separable, we have for $b(i-1)+1 \le j \le bi$ and $i \in [m]$

$$\boldsymbol{y}_k^j = \operatorname*{arg\,max}_{\boldsymbol{y}^j \in \mathbb{R}^d} \left\{ \left\langle \boldsymbol{y}^j, \boldsymbol{x}_{k-1,i} \right\rangle - f_{\pi_j^{(k)}}^*(\boldsymbol{y}^j) \right\},$$

which leads to $\boldsymbol{x}_{k-1,i} \in \partial f_{\pi_j^{(k)}}^*(\boldsymbol{y}_k^j)$. Further, since each component function $f_j^*$ is $\frac{1}{L_j}$-strongly convex thus for $b(i-1)+1 \le j \le bi$, we also have

$$f_{\pi_j^{(k)}}^*(\boldsymbol{v}_k^j) \ge f_{\pi_j^{(k)}}^*(\boldsymbol{y}_k^j) + \left\langle \boldsymbol{x}_{k-1,i}, \boldsymbol{v}_k^j - \boldsymbol{y}_k^j \right\rangle + \frac{1}{2L_{\pi_j^{(k)}}} \|\boldsymbol{v}_k^j - \boldsymbol{y}_k^j\|^2,$$

which leads to

$$
\begin{aligned}
&\mathcal{L}(\boldsymbol{x}_k, \boldsymbol{v}) \\
&= \frac{1}{n} \sum_{i=1}^{m} \left( \left\langle \boldsymbol{E}_b^\top \boldsymbol{v}_k^{(i)}, \boldsymbol{x}_{k-1,i} \right\rangle - \sum_{j=b(i-1)+1}^{bi} f_{\pi_j^{(k)}}^*(\boldsymbol{v}_k^j) \right) + \frac{1}{n} \sum_{i=1}^{m} \left\langle \boldsymbol{E}_b^\top \boldsymbol{v}_k^{(i)}, \boldsymbol{x}_k - \boldsymbol{x}_{k-1,i} \right\rangle \\
&\le \frac{1}{n} \sum_{i=1}^{m} \left( \left\langle \boldsymbol{E}_b^\top \boldsymbol{y}_k^{(i)}, \boldsymbol{x}_{k-1,i} \right\rangle - \sum_{j=b(i-1)+1}^{bi} f_{\pi_j^{(k)}}^*(\boldsymbol{y}_k^j) \right) + \frac{1}{n} \sum_{i=1}^{m} \left\langle \boldsymbol{E}_b^\top \boldsymbol{v}_k^{(i)}, \boldsymbol{x}_k - \boldsymbol{x}_{k-1,i} \right\rangle \\
&\quad - \frac{1}{2n} \|\boldsymbol{y}_k - \boldsymbol{v}_k\|_{\boldsymbol{\Lambda}_k^{-1}}^2.
\end{aligned}
$$

Using the same argument, as $\boldsymbol{x}_* \in \partial f_i^*(\boldsymbol{y}_*^i)$ for $i \in [n]$, we have

$$f_{\pi_i^{(k)}}^*(\boldsymbol{y}_k^i) \ge f_{\pi_i^{(k)}}^*(\boldsymbol{y}_{*,k}^i) + \left\langle \boldsymbol{x}_*, \boldsymbol{y}_k^i - \boldsymbol{y}_{*,k}^i \right\rangle + \frac{1}{2L_{\pi_i^{(k)}}} \|\boldsymbol{y}_k^i - \boldsymbol{y}_{*,k}^i\|^2.$$

Thus,

$$
\begin{aligned}
&\mathcal{L}(\boldsymbol{x}_*, \boldsymbol{y}_*) \\
&= \frac{1}{n} \sum_{i=1}^{m} \left( \left\langle \boldsymbol{E}_b^\top \boldsymbol{y}_{*,k}^{(i)}, \boldsymbol{x}_* \right\rangle - \sum_{j=b(i-1)+1}^{bi} f_{\pi_j^{(k)}}^*(\boldsymbol{y}_{*,k}^j) \right) \\
&\ge \frac{1}{n} \sum_{i=1}^{m} \left( \left\langle \boldsymbol{E}_b^\top \boldsymbol{y}_k^{(i)}, \boldsymbol{x}_* \right\rangle - \sum_{j=b(i-1)+1}^{bi} f_{\pi_j^{(k)}}^*(\boldsymbol{y}_k^j) \right) + \frac{1}{2n} \|\boldsymbol{y}_k - \boldsymbol{y}_{*,k}\|_{\boldsymbol{\Lambda}_k^{-1}}^2 \\
&= \frac{1}{n} \sum_{i=1}^{m} \left( \left\langle \boldsymbol{E}_b^\top \boldsymbol{y}_k^{(i)}, \boldsymbol{x}_* \right\rangle + \frac{b}{2\eta_k} \|\boldsymbol{x}_* - \boldsymbol{x}_{k-1,i}\|^2 - \frac{b}{2\eta_k} \|\boldsymbol{x}_* - \boldsymbol{x}_{k-1,i}\|^2 - \sum_{j=b(i-1)+1}^{bi} f_{\pi_j^{(k)}}^*(\boldsymbol{y}_k^j) \right) \\
&\quad + \frac{1}{2n} \|\boldsymbol{y}_k - \boldsymbol{y}_{*,k}\|_{\boldsymbol{\Lambda}_k^{-1}}^2.
\end{aligned}
$$

Using the updating scheme of $\boldsymbol{x}_{k-1,i+1}$ and noticing that $\phi_k^i(\boldsymbol{x}) = \left\langle \boldsymbol{E}_b^\top \boldsymbol{y}_k^{(i)}, \boldsymbol{x} \right\rangle + \frac{b}{2\eta_k} \|\boldsymbol{x} - \boldsymbol{x}_{k-1,i}\|^2$ is $\frac{b}{\eta_k}$-strongly convex and minimized at $\boldsymbol{x}_{k-1,i+1}$, we have

$$
\begin{aligned}
&\left\langle \boldsymbol{E}_b^\top \boldsymbol{y}_k^{(i)}, \boldsymbol{x}_* \right\rangle + \frac{b}{2\eta_k} \|\boldsymbol{x}_* - \boldsymbol{x}_{k-1,i}\|^2 \\
&\ge \left\langle \boldsymbol{E}_b^\top \boldsymbol{y}_k^{(i)}, \boldsymbol{x}_{k-1,i+1} \right\rangle + \frac{b}{2\eta_k} \|\boldsymbol{x}_{k-1,i+1} - \boldsymbol{x}_{k-1,i}\|^2 + \frac{b}{2\eta_k} \|\boldsymbol{x}_{k-1,i+1} - \boldsymbol{x}_*\|^2,
\end{aligned}
$$

which leads to

$$\mathcal{L}(\boldsymbol{x}_*, \boldsymbol{y}_*) \geq \frac{1}{n} \sum_{i=1}^{m} \Big( \Big\langle \boldsymbol{E}_b^\top \boldsymbol{y}_k^{(i)}, \boldsymbol{x}_{k-1,i+1} \Big\rangle + \frac{b}{2\eta_k} \| \boldsymbol{x}_{k-1,i+1} - \boldsymbol{x}_{k-1,i} \|^2 - \sum_{j=b(i-1)+1}^{bi} f_{\pi_j^{(k)}}^*(\boldsymbol{y}_k^i) \Big)$$

$$+ \frac{b}{2n\eta_k} \sum_{i=1}^{m} \Big( \| \boldsymbol{x}_{k-1,i+1} - \boldsymbol{x}_* \|^2 - \| \boldsymbol{x}_{k-1,i} - \boldsymbol{x}_* \|^2 \Big) + \frac{1}{2n} \| \boldsymbol{y}_k - \boldsymbol{y}_{*,k} \|_{\boldsymbol{\Lambda}_k^{-1}}^2$$

$$= \frac{1}{n} \sum_{i=1}^{m} \Big( \Big\langle \boldsymbol{E}_b^\top \boldsymbol{y}_k^{(i)}, \boldsymbol{x}_{k-1,i+1} \Big\rangle + \frac{b}{2\eta_k} \| \boldsymbol{x}_{k-1,i+1} - \boldsymbol{x}_{k-1,i} \|^2 - \sum_{j=b(i-1)+1}^{bi} f_{\pi_j^{(k)}}^*(\boldsymbol{y}_k^j) \Big)$$

$$+ \frac{b}{2n\eta_k} \Big( \| \boldsymbol{x}_k - \boldsymbol{x}_* \|^2 - \| \boldsymbol{x}_{k-1} - \boldsymbol{x}_* \|^2 \Big) + \frac{1}{2n} \| \boldsymbol{y}_k - \boldsymbol{y}_{*,k} \|_{\boldsymbol{\Lambda}_k^{-1}}^2 .$$

Hence, combining the bounds on $\mathcal{L}(\boldsymbol{x}_k, \boldsymbol{v})$ and $\mathcal{L}(\boldsymbol{x}_*, \boldsymbol{y}_*)$ and letting

$$\mathcal{E}_k := \eta_k \big( \mathcal{L}(\boldsymbol{x}_k, \boldsymbol{v}) - \mathcal{L}(\boldsymbol{x}_*, \boldsymbol{y}_*) \big) + \frac{b}{2n} \| \boldsymbol{x}_k - \boldsymbol{x}_* \|^2 - \frac{b}{2n} \| \boldsymbol{x}_{k-1} - \boldsymbol{x}_* \|^2 ,$$

we obtain

$$\mathcal{E}_k \leq \frac{\eta_k}{n} \sum_{i=1}^{m} \Big\langle \boldsymbol{E}_b^\top \boldsymbol{y}_k^{(i)}, \boldsymbol{x}_{k-1,i} - \boldsymbol{x}_{k-1,i+1} \Big\rangle + \frac{\eta_k}{n} \sum_{i=1}^{m} \Big\langle \boldsymbol{E}_b^\top \boldsymbol{v}_k^{(i)}, \boldsymbol{x}_k - \boldsymbol{x}_{k-1,i} \Big\rangle$$

$$- \frac{\eta_k}{2n} \| \boldsymbol{y}_k - \boldsymbol{v}_k \|_{\boldsymbol{\Lambda}_k^{-1}}^2 - \frac{\eta_k}{2n} \| \boldsymbol{y}_k - \boldsymbol{y}_{*,k} \|_{\boldsymbol{\Lambda}_k^{-1}}^2 - \frac{b}{2n} \sum_{i=1}^{m} \| \boldsymbol{x}_{k-1,i+1} - \boldsymbol{x}_{k-1,i} \|^2$$

$$= \frac{\eta_k}{n} \sum_{i=1}^{m} \Big\langle \boldsymbol{E}_b^\top \boldsymbol{y}_k^{(i)}, \boldsymbol{x}_k - \boldsymbol{x}_{k-1,i+1} \Big\rangle + \frac{\eta_k}{n} \sum_{i=1}^{m} \Big\langle \boldsymbol{E}_b^\top \big( \boldsymbol{v}_k^{(i)} - \boldsymbol{y}_k^{(i)} \big), \boldsymbol{x}_k - \boldsymbol{x}_{k-1,i} \Big\rangle$$

$$- \frac{\eta_k}{2n} \| \boldsymbol{y}_k - \boldsymbol{v}_k \|_{\boldsymbol{\Lambda}_k^{-1}}^2 - \frac{\eta_k}{2n} \| \boldsymbol{y}_k - \boldsymbol{y}_{*,k} \|_{\boldsymbol{\Lambda}_k^{-1}}^2 - \frac{b}{2n} \sum_{i=1}^{m} \| \boldsymbol{x}_{k-1,i+1} - \boldsymbol{x}_{k-1,i} \|^2 ,$$

thus completing the proof. $\qquad \square$

We note that the first inner product term $\mathcal{T}_1 := \frac{\eta_k}{n} \sum_{i=1}^{m} \Big\langle \boldsymbol{E}_b^\top \boldsymbol{y}_k^{(i)}, \boldsymbol{x}_k - \boldsymbol{x}_{k-1,i+1} \Big\rangle$ in Eq. (5) can be cancelled by the last negative term $-\frac{b}{2n} \sum_{i=1}^{m} \| \boldsymbol{x}_{k-1,i+1} - \boldsymbol{x}_{k-1,i} \|^2$ therein, as precisely proved in Lemma 10 of Appendix C. In the following subsections, we continue our analysis and handle the remaining terms in Eq. (5) according to different shuffling and derive the final complexity.

## B.1 Random reshuffling/shuffle-once schemes

We introduce the following lemma to bound the second inner product term $\mathcal{T}_2 := \frac{\eta_k}{n} \sum_{i=1}^{m} \Big\langle \boldsymbol{E}_b^\top \big( \boldsymbol{v}_k^{(i)} - \boldsymbol{y}_k^{(i)} \big), \boldsymbol{x}_k - \boldsymbol{x}_{k-1,i} \Big\rangle$ in Lemma 6 when there are random permutations.

**Lemma 7.** *Under Assumptions 1 and 2, for any $k \in [K]$, the iterates $\{ \boldsymbol{y}_k^{(i)} \}_{i=1}^{m}$ and $\{ \boldsymbol{x}_{k-1,i} \}_{i=1}^{m+1}$ generated by Algorithm 1 with uniformly random shuffling (RR/SO) satisfy*

$$\mathbb{E}[\mathcal{T}_2] \leq \mathbb{E} \Big[ \frac{\eta_k^3 n \hat{L}_{\pi^{(k)}}^g \tilde{L}_{\pi^{(k)}}^g}{b^2} \| \boldsymbol{y}_k - \boldsymbol{y}_{*,k} \|_{\boldsymbol{\Lambda}_k^{-1}}^2 + \frac{\eta_k}{2n} \| \boldsymbol{v}_k - \boldsymbol{y}_k \|_{\boldsymbol{\Lambda}_k^{-1}}^2 \Big] + \frac{\eta_k^3 \tilde{L}^g (n-b)(n+b)}{6b^2(n-1)} \sigma_*^2 ,$$

*where $\mathcal{T}_2 := \frac{\eta_k}{n} \sum_{i=1}^{m} \Big\langle \boldsymbol{E}_b^\top \big( \boldsymbol{v}_k^{(i)} - \boldsymbol{y}_k^{(i)} \big), \boldsymbol{x}_k - \boldsymbol{x}_{k-1,i} \Big\rangle.$*

*Proof.* First note that $\boldsymbol{x}_k - \boldsymbol{x}_{k-1,i} = \sum_{j=i}^{m}(\boldsymbol{x}_{k-1,j+1} - \boldsymbol{x}_{k-1,j}) = -\frac{\eta_k}{b}\sum_{j=i}^{m}\boldsymbol{E}_b^\top \boldsymbol{y}_k^{(j)} = -\frac{\eta_k}{b}\boldsymbol{E}^\top \boldsymbol{I}_{bd(i-1)\uparrow}\boldsymbol{y}_k$, so we have

$$\frac{\eta_k}{n}\sum_{i=1}^{m}\left\langle \boldsymbol{E}_b^\top (\boldsymbol{v}_k^{(i)} - \boldsymbol{y}_k^{(i)}), \boldsymbol{x}_k - \boldsymbol{x}_{k-1,i}\right\rangle$$

$$= \frac{\eta_k}{n}\sum_{i=1}^{m}\left\langle \boldsymbol{E}_b^\top (\boldsymbol{v}_k^{(i)} - \boldsymbol{y}_k^{(i)}), \sum_{j=i}^{m}(\boldsymbol{x}_{k-1,j+1} - \boldsymbol{x}_{k-1,j})\right\rangle$$

$$= -\frac{\eta_k^2}{bn}\sum_{i=1}^{m}\left\langle \boldsymbol{E}^\top \boldsymbol{I}_{(di)}(\boldsymbol{v}_k - \boldsymbol{y}_k), \boldsymbol{E}^\top \boldsymbol{I}_{bd(i-1)\uparrow}\boldsymbol{y}_k\right\rangle$$

$$= \underbrace{-\frac{\eta_k^2}{bn}\sum_{i=1}^{m}\left\langle \boldsymbol{E}^\top \boldsymbol{I}_{(di)}(\boldsymbol{v}_k - \boldsymbol{y}_k), \boldsymbol{E}^\top \boldsymbol{I}_{bd(i-1)\uparrow}(\boldsymbol{y}_k - \boldsymbol{y}_{*,k})\right\rangle}_{\mathcal{I}_1}$$

$$\underbrace{-\frac{\eta_k^2}{bn}\sum_{i=1}^{m}\left\langle \boldsymbol{E}^\top \boldsymbol{I}_{(di)}(\boldsymbol{v}_k - \boldsymbol{y}_k), \boldsymbol{E}^\top \boldsymbol{I}_{bd(i-1)\uparrow}\boldsymbol{y}_{*,k}\right\rangle}_{\mathcal{I}_2}.$$

For the term $\mathcal{I}_1$, we use Young's inequality with $\alpha > 0$ to be set later and obtain

$$\mathcal{I}_1 = -\frac{\eta_k^2}{bn}\sum_{i=1}^{m}\left\langle \boldsymbol{E}^\top \boldsymbol{I}_{(di)}(\boldsymbol{v}_k - \boldsymbol{y}_k), \boldsymbol{E}^\top \boldsymbol{I}_{bd(i-1)\uparrow}(\boldsymbol{y}_k - \boldsymbol{y}_{*,k})\right\rangle$$

$$\leq \frac{\eta_k^2}{2bn\alpha}\sum_{i=1}^{m}\|\boldsymbol{E}^\top \boldsymbol{I}_{(di)}(\boldsymbol{v}_k - \boldsymbol{y}_k)\|^2 + \frac{\eta_k^2\alpha}{2bn}\sum_{i=1}^{m}\|\boldsymbol{E}^\top \boldsymbol{I}_{bd(i-1)\uparrow}(\boldsymbol{y}_k - \boldsymbol{y}_{*,k})\|^2. \qquad (6)$$

Further, notice that

$$\frac{\eta_k^2\alpha}{2bn}\sum_{i=1}^{m}\|\boldsymbol{E}^\top \boldsymbol{I}_{bd(i-1)\uparrow}(\boldsymbol{y}_k - \boldsymbol{y}_{*,k})\|^2$$

$$= \frac{\eta_k^2\alpha}{2bn}\sum_{i=1}^{m}(\boldsymbol{y}_k - \boldsymbol{y}_{*,k})^\top \boldsymbol{I}_{bd(i-1)\uparrow}\boldsymbol{E}\boldsymbol{E}^\top \boldsymbol{I}_{bd(i-1)\uparrow}(\boldsymbol{y}_k - \boldsymbol{y}_{*,k})$$

$$= \frac{\eta_k^2\alpha}{2bn}(\boldsymbol{y}_k - \boldsymbol{y}_{*,k})^\top \Big(\sum_{i=1}^{m}\boldsymbol{I}_{bd(i-1)\uparrow}\boldsymbol{E}\boldsymbol{E}^\top \boldsymbol{I}_{bd(i-1)\uparrow}\Big)(\boldsymbol{y}_k - \boldsymbol{y}_{*,k})$$

$$= \frac{\eta_k^2\alpha}{2bn}(\boldsymbol{y}_k - \boldsymbol{y}_{*,k})^\top \boldsymbol{\Lambda}_k^{-1/2}\boldsymbol{\Lambda}_k^{1/2}\Big(\sum_{i=1}^{m}\boldsymbol{I}_{bd(i-1)\uparrow}\boldsymbol{E}\boldsymbol{E}^\top \boldsymbol{I}_{bd(i-1)\uparrow}\Big)\boldsymbol{\Lambda}_k^{1/2}\boldsymbol{\Lambda}_k^{-1/2}(\boldsymbol{y}_k - \boldsymbol{y}_{*,k})$$

$$\leq \frac{\eta_k^2\alpha}{2bn}\Big\|\boldsymbol{\Lambda}_k^{1/2}\Big(\sum_{i=1}^{m}\boldsymbol{I}_{bd(i-1)\uparrow}\boldsymbol{E}\boldsymbol{E}^\top \boldsymbol{I}_{bd(i-1)\uparrow}\Big)\boldsymbol{\Lambda}_k^{1/2}\Big\|_2\|\boldsymbol{y}_k - \boldsymbol{y}_{*,k}\|_{\boldsymbol{\Lambda}_k^{-1}}^2$$

$$= \frac{\eta_k^2 m\alpha}{2b}\hat{L}_{\pi(k)}^g\|\boldsymbol{y}_k - \boldsymbol{y}_{*,k}\|_{\boldsymbol{\Lambda}_k^{-1}}^2, \qquad (7)$$

where for the last inequality we use Cauchy-Schwarz inequality. Using the same argument, we can bound

$$\frac{\eta_k^2}{2bn\alpha}\sum_{i=1}^{m}\|\boldsymbol{E}^\top \boldsymbol{I}_{(di)}(\boldsymbol{v}_k - \boldsymbol{y}_k)\|^2 \leq \frac{\eta_k^2}{2bn\alpha}\Big\|\boldsymbol{\Lambda}_k^{1/2}\Big(\sum_{i=1}^{m}\boldsymbol{I}_{(di)}\boldsymbol{E}\boldsymbol{E}^\top \boldsymbol{I}_{(di)}\Big)\boldsymbol{\Lambda}_k^{1/2}\Big\|_2\|\boldsymbol{v}_k - \boldsymbol{y}_k\|_{\boldsymbol{\Lambda}_k^{-1}}^2$$

$$= \frac{\eta_k^2}{2n\alpha}\tilde{L}_{\pi(k)}^g\|\boldsymbol{v}_k - \boldsymbol{y}_k\|_{\boldsymbol{\Lambda}_k^{-1}}^2. \qquad (8)$$

Thus, combining (6)–(8) and choosing $\alpha = 2\eta_k\tilde{L}_{\pi(k)}^g$, we obtain

$$\mathcal{I}_1 \leq \frac{\eta_k^3 m\hat{L}_{\pi(k)}^g\tilde{L}_{\pi(k)}^g}{b}\|\boldsymbol{y}_k - \boldsymbol{y}_{*,k}\|_{\boldsymbol{\Lambda}_k^{-1}}^2 + \frac{\eta_k}{4n}\|\boldsymbol{v}_k - \boldsymbol{y}_k\|_{\boldsymbol{\Lambda}_k^{-1}}^2.$$

For the term $\mathcal{I}_2$, we again apply Young's inequality with $\beta > 0$ to be set later and obtain

$$
\begin{aligned}
\mathcal{I}_2 = {} & -\frac{\eta_k^2}{bn} \sum_{i=1}^{m} \big\langle \boldsymbol{E}^\top \boldsymbol{I}_{(di)}(\boldsymbol{v}_k - \boldsymbol{y}_k),\, \boldsymbol{E}^\top \boldsymbol{I}_{bd(i-1)\uparrow} \boldsymbol{y}_{*,k} \big\rangle \\
\leq {} & \frac{\eta_k^2 \beta}{2bn} \sum_{i=1}^{m} \| \boldsymbol{E}^\top \boldsymbol{I}_{bd(i-1)\uparrow} \boldsymbol{y}_{*,k} \|^2 + \frac{\eta_k^2}{2bn\beta} \sum_{i=1}^{m} \| \boldsymbol{E}^\top \boldsymbol{I}_{(di)}(\boldsymbol{v}_k - \boldsymbol{y}_k) \|^2 \\
\leq {} & \frac{\eta_k^2 \beta}{2bn} \sum_{i=1}^{m} \| \boldsymbol{E}^\top \boldsymbol{I}_{bd(i-1)\uparrow} \boldsymbol{y}_{*,k} \|^2 + \frac{\eta_k^2 \tilde{L}^g_{\pi^{(k)}}}{2n\beta} \| \boldsymbol{v}_k - \boldsymbol{y}_k \|^2_{\boldsymbol{\Lambda}_k^{-1}}.
\end{aligned}
$$

Choosing $\beta = 2\eta_k \tilde{L}^g$ and using the fact that $\tilde{L}^g_{\pi^{(k)}} \leq \tilde{L}^g$, we have

$$
\mathcal{I}_2 \leq \frac{\eta_k^3 \tilde{L}^g}{bn} \sum_{i=1}^{m} \| \boldsymbol{E}^\top \boldsymbol{I}_{bd(i-1)\uparrow} \boldsymbol{y}_{*,k} \|^2 + \frac{\eta_k}{4n} \| \boldsymbol{v}_k - \boldsymbol{y}_k \|^2_{\boldsymbol{\Lambda}_k^{-1}}.
$$

Hence, combining the above two estimates with $m = n/b$, we have

$$
\mathcal{T}_2 \leq \frac{\eta_k^3 \tilde{L}^g}{bn} \sum_{i=1}^{m} \| \boldsymbol{E}^\top \boldsymbol{I}_{bd(i-1)\uparrow} \boldsymbol{y}_{*,k} \|^2 + \frac{\eta_k^3 n \hat{L}^g_{\pi^{(k)}} \tilde{L}^g_{\pi^{(k)}}}{b^2} \| \boldsymbol{y}_k - \boldsymbol{y}_{*,k} \|^2_{\boldsymbol{\Lambda}_k^{-1}} + \frac{\eta_k}{2n} \| \boldsymbol{v}_k - \boldsymbol{y}_k \|^2_{\boldsymbol{\Lambda}_k^{-1}}.
$$

First, consider the RR scheme. Taking conditional expectation on both sides w.r.t. the randomness up to but not including $k$-th epoch, we have

$$
\begin{aligned}
\mathbb{E}_k[\mathcal{T}_2] \leq {} & \frac{\eta_k^3 \tilde{L}^g}{bn} \mathbb{E}_k \Big[ \sum_{i=1}^{m} \| \boldsymbol{E}^\top \boldsymbol{I}_{bd(i-1)\uparrow} \boldsymbol{y}_{*,k} \|^2 \Big] \\
& + \mathbb{E}_k \Big[ \frac{\eta_k^3 n \hat{L}^g_{\pi^{(k)}} \tilde{L}^g_{\pi^{(k)}}}{b^2} \| \boldsymbol{y}_k - \boldsymbol{y}_{*,k} \|^2_{\boldsymbol{\Lambda}_k^{-1}} + \frac{\eta_k}{2n} \| \boldsymbol{v}_k - \boldsymbol{y}_k \|^2_{\boldsymbol{\Lambda}_k^{-1}} \Big].
\end{aligned}
$$

For the first term, since the only randomness comes from the permutation $\pi^{(k)}$, we can proceed as in the proof of Lemma 11 and obtain

$$
\begin{aligned}
\frac{\eta_k^3 \tilde{L}^g}{bn} \mathbb{E}_k \Big[ \sum_{i=1}^{m} \| \boldsymbol{E}^\top \boldsymbol{I}_{bd(i-1)\uparrow} \boldsymbol{y}_{*,k} \|^2 \Big] \overset{(i)}{=} {} & \frac{\eta_k^3 \tilde{L}^g}{bn} \sum_{i=1}^{m} \mathbb{E}_{\pi^{(k)}} \Big[ \| \boldsymbol{E}^\top \boldsymbol{I}_{bd(i-1)\uparrow} \boldsymbol{y}_{*,k} \|^2 \Big] \\
= {} & \frac{\eta_k^3 \tilde{L}^g}{bn} \sum_{i=1}^{m} (n - b(i-1))^2 \mathbb{E}_{\pi^{(k)}} \Big[ \Big\| \frac{\boldsymbol{E}^\top \boldsymbol{I}_{bd(i-1)\uparrow} \boldsymbol{y}_{*,k}}{n - b(i-1)} \Big\|^2 \Big] \\
\overset{(ii)}{\leq} {} & \frac{\eta_k^3 \tilde{L}^g}{bn} \sum_{i=1}^{m} (n - b(i-1))^2 \frac{b(i-1)}{(n - b(i-1))(n-1)} \sigma_*^2 \\
= {} & \frac{\eta_k^3 \tilde{L}^g (n-b)(n+b)}{6b^2(n-1)} \sigma_*^2,
\end{aligned}
$$

where we use the linearity of expectation for $(i)$, and $(ii)$ is due to Lemma 5 and the definition $\sigma_*^2 = \frac{1}{n}\sum_{i=1}^{n} \| \nabla f_i(\boldsymbol{x}_*) \|^2 = \frac{1}{n}\sum_{i=1}^{n} \| \boldsymbol{y}_*^i \|^2$. Then taking expectation w.r.t. all randomness on both sides, we obtain

$$
\mathbb{E}[\mathcal{T}_2] \leq \mathbb{E}\Big[ \frac{\eta_k^3 n \hat{L}^g_{\pi^{(k)}} \tilde{L}^g_{\pi^{(k)}}}{b^2} \| \boldsymbol{y}_k - \boldsymbol{y}_{*,k} \|^2_{\boldsymbol{\Lambda}_k^{-1}} + \frac{\eta_k}{2n} \| \boldsymbol{v}_k - \boldsymbol{y}_k \|^2_{\boldsymbol{\Lambda}_k^{-1}} \Big] + \frac{\eta_k^3 \tilde{L}^g (n-b)(n+b)}{6b^2(n-1)} \sigma_*^2.
$$

Finally, we remark that the above argument for bounding the term $\frac{\eta_k^3 \tilde{L}^g}{bn} \mathbb{E}_k\Big[ \sum_{i=1}^{m} \| \boldsymbol{E}^\top \boldsymbol{I}_{bd(i-1)\uparrow} \boldsymbol{y}_{*,k} \|^2 \Big]$ also applies to the SO scheme, in which case there is only one random permutation at the very beginning that induces the randomness. $\qquad\square$

We state the final convergence rate and complexity in the following theorem and provide the proof for completeness.

**Theorem 4.** *Under Assumptions 1 and 2, if $\eta_k \leq \frac{b}{n\sqrt{2\hat{L}^g_{\pi(k)}\tilde{L}^g_{\pi(k)}}}$ and $H_K = \sum_{k=1}^K \eta_k$, the output $\hat{x}_K$ of Algorithm 1 with uniformly random (RR/SO) shuffling satisfies*

$$\mathbb{E}[H_K(f(\hat{x}_K) - f(x_*))] \leq \frac{b}{2n}\|x_0 - x_*\|_2^2 + \sum_{k=1}^K \frac{\eta_k^3 \tilde{L}^g(n-b)(n+b)}{6b^2(n-1)}\sigma_*^2.$$

*As a consequence, for any $\epsilon > 0$, there exists a choice of a constant step size $\eta_k = \eta$ for which $\mathbb{E}[f(\hat{x}_K) - f(x_*)] \leq \epsilon$ after $\mathcal{O}\Big(\frac{n\sqrt{\hat{L}^g\tilde{L}^g}\|x_0 - x_*\|_2^2}{\epsilon} + \sqrt{\frac{(n-b)(n+b)}{n(n-1)}}\frac{\sqrt{n\tilde{L}^g}\sigma_*\|x_0 - x_*\|_2^2}{\epsilon^{3/2}}\Big)$ gradient queries.*

*Proof.* Combining the bounds in Lemma 10 and 2 and plugging them into Eq. (5), we obtain

$$\mathbb{E}[\mathcal{E}_k] \leq \mathbb{E}\Big[\Big(\frac{\eta_k^3 n\hat{L}^g_{\pi(k)}\tilde{L}^g_{\pi(k)}}{b^2} - \frac{\eta_k}{2n}\Big)\|y_k - y_{*,k}\|_{\Lambda_k^{-1}}^2\Big] + \frac{\eta_k^3 \tilde{L}^g(n-b)(n+b)}{6b^2(n-1)}\sigma_*^2.$$

For the stepsize $\eta_k$ such that $\eta_k \leq \frac{b}{n\sqrt{2\hat{L}^g_{\pi(k)}\tilde{L}^g_{\pi(k)}}}$, we have $\frac{\eta_k^3 n\hat{L}^g_{\pi(k)}\tilde{L}^g_{\pi(k)}}{b^2} - \frac{\eta_k}{2n} \leq 0$, thus

$$\mathbb{E}[\mathcal{E}_k] \leq \frac{\eta_k^3 \tilde{L}^g(n-b)(n+b)}{6b^2(n-1)}\sigma_*^2.$$

Using our definition of $\mathcal{E}_k$ and telescoping from $k = 1$ to $K$, we have

$$\mathbb{E}\Big[\sum_{k=1}^K \eta_k \text{Gap}^v(x_k, y_*)\Big] \leq \frac{b}{2n}\|x_* - x_0\|_2^2 - \frac{b}{2n}\mathbb{E}[\|x_* - x_K\|_2^2] + \sum_{k=1}^K \frac{\eta_k^3 \tilde{L}^g(n-b)(n+b)}{6b^2(n-1)}\sigma_*^2.$$

Noticing that $\mathcal{L}(x, v)$ is convex in $x$ for a fixed $v$, we have $\text{Gap}^v(\hat{x}_K, y_*) \leq \sum_{k=1}^K \eta_k \text{Gap}^v(x_k, y_*)/H_K$, where $\hat{x}_K = \sum_{k=1}^K \eta_k x_k/H_K$ and $H_K = \sum_{k=1}^K \eta_k$, which leads to

$$\mathbb{E}\Big[H_K \text{Gap}^v(\hat{x}_K, y_*)\Big] \leq \frac{b}{2n}\|x_0 - x_*\|_2^2 + \sum_{k=1}^K \frac{\eta_k^3 \tilde{L}^g(n-b)(n+b)}{6b^2(n-1)}\sigma_*^2.$$

Further choosing $v = y_{\hat{x}_K}$, we obtain

$$\mathbb{E}[H_K\big(f(\hat{x}_K) - f(x_*)\big)] \leq \frac{b}{2n}\|x_0 - x_*\|_2^2 + \sum_{k=1}^K \frac{\eta_k^3 \tilde{L}^g(n-b)(n+b)}{6b^2(n-1)}\sigma_*^2. \qquad (9)$$

To analyze the individual gradient oracle complexity, we choose constant stepsizes $\eta \leq \frac{b}{n\sqrt{2\hat{L}^g\tilde{L}^g}}$, then Eq. (9) will become

$$\mathbb{E}[f(\hat{x}_K) - f(x_*)] \leq \frac{b}{2n\eta K}\|x_0 - x_*\|_2^2 + \frac{\eta^2 \tilde{L}^g(n-b)(n+b)}{6b^2(n-1)}\sigma_*^2.$$

Without loss of generality, we assume that $b \neq n$, otherwise the method and its analysis reduce to (full) gradient descent. We consider the following two cases:

- "Small $K$" case: if $\eta = \frac{b}{n\sqrt{2\hat{L}^g\tilde{L}^g}} \leq \Big(\frac{3b^3(n-1)\|x_0 - x_*\|_2^2}{n(n-b)(n+b)\tilde{L}^g K\sigma_*^2}\Big)^{1/3}$, we have

$$\mathbb{E}[f(\hat{x}_K) - f(x_*)]$$

$$\leq \frac{b}{2n\eta K}\|x_0 - x_*\|_2^2 + \frac{\eta^2 \tilde{L}^g(n-b)(n+b)}{6b^2(n-1)}\sigma_*^2$$

$$\leq \frac{\sqrt{\hat{L}^g\tilde{L}^g}}{\sqrt{2}K}\|x_0 - x_*\|_2^2 + \frac{1}{2}\Big(\frac{(n-b)(n+b)}{n^2(n-1)}\Big)^{1/3}\frac{(\tilde{L}^g)^{1/3}\sigma_*^{2/3}\|x_0 - x_*\|_2^{4/3}}{3^{1/3}K^{2/3}}.$$

- "Large $K$" case: if $\eta = \left( \frac{3b^3(n-1)\|\boldsymbol{x}_0 - \boldsymbol{x}_*\|_2^2}{n(n-b)(n+b)\tilde{L}^g K \sigma_*^2} \right)^{1/3} \leq \frac{b}{n\sqrt{2\hat{L}^g \tilde{L}^g}}$, we have

$$\mathbb{E}[f(\hat{\boldsymbol{x}}_K) - f(\boldsymbol{x}_*)] \leq \frac{b}{2n\eta K}\|\boldsymbol{x}_0 - \boldsymbol{x}_*\|_2^2 + \frac{\eta^2 \tilde{L}^g (n-b)(n+b)}{6b^2(n-1)}\sigma_*^2$$

$$\leq \left( \frac{(n-b)(n+b)}{n^2(n-1)} \right)^{1/3} \frac{(\tilde{L}^g)^{1/3}\sigma_*^{2/3}\|\boldsymbol{x}_0 - \boldsymbol{x}_*\|_2^{4/3}}{3^{1/3}K^{2/3}}.$$

Combining these two cases by setting $\eta = \min \left\{ \frac{b}{n\sqrt{2\hat{L}^g \tilde{L}^g}}, \left( \frac{3b^3(n-1)\|\boldsymbol{x}_0 - \boldsymbol{x}_*\|_2^2}{n(n-b)(n+b)\tilde{L}^g K \sigma_*^2} \right)^{1/3} \right\}$, we obtain

$$\mathbb{E}[f(\hat{\boldsymbol{x}}_K) - f(\boldsymbol{x}_*)] \leq \frac{\sqrt{\hat{L}^g \tilde{L}^g}}{\sqrt{2}K}\|\boldsymbol{x}_0 - \boldsymbol{x}_*\|_2^2 + \left( \frac{(n-b)(n+b)}{n^2(n-1)} \right)^{1/3} \frac{(\tilde{L}^g)^{1/3}\sigma_*^{2/3}\|\boldsymbol{x}_0 - \boldsymbol{x}_*\|_2^{4/3}}{3^{1/3}K^{2/3}}.$$

Hence, to guarantee $\mathbb{E}[f(\hat{\boldsymbol{x}}_K) - f(\boldsymbol{x}_*)] \leq \epsilon$ for $\epsilon > 0$, the total number of individual gradient evaluations will be

$$nK \geq \max \left\{ \frac{n\sqrt{2\hat{L}^g \tilde{L}^g}\|\boldsymbol{x}_0 - \boldsymbol{x}_*\|_2^2}{\epsilon}, \left( \frac{(n-b)(n+b)}{n-1} \right)^{1/2} \frac{2^{3/2}(\tilde{L}^g)^{1/2}\sigma_*\|\boldsymbol{x}_0 - \boldsymbol{x}_*\|_2^2}{3^{1/2}\epsilon^{3/2}} \right\},$$

as claimed. $\qquad\square$

## B.2 Incremental gradient descent (IG)

In this subsection, we provide the convergence results for incremental gradient descent which does not involve random permutations. We first prove the technical lemma below to bound the term $\mathcal{T}_2 := \frac{\eta_k}{n} \sum_{i=1}^m \left\langle \boldsymbol{E}_b^\top \left( \boldsymbol{v}_k^{(i)} - \boldsymbol{y}_k^{(i)} \right), \boldsymbol{x}_k - \boldsymbol{x}_{k-1,i} \right\rangle$ in Eq. (5) of Lemma 6.

**Lemma 8.** *For any $k \in [K]$, the iterates $\{\boldsymbol{y}_k^{(i)}\}_{i=1}^m$ and $\{\boldsymbol{x}_{k-1,i}\}_{i=1}^{m+1}$ generated by Algorithm 1 with fixed data ordering satisfy*

$$\mathcal{T}_2 \leq \frac{\eta_k^3 n}{b^2}\hat{L}_0^g \tilde{L}_0^g \|\boldsymbol{y}_k - \boldsymbol{y}_*\|_{\boldsymbol{\Lambda}^{-1}}^2 + \frac{\eta_k}{2n}\|\boldsymbol{v} - \boldsymbol{y}_k\|_{\boldsymbol{\Lambda}^{-1}}^2$$
$$+ \min \left\{ \frac{\eta_k^3 n}{b^2}\hat{L}_0^g \tilde{L}_0^g \|\boldsymbol{y}_*\|_{\boldsymbol{\Lambda}^{-1}}^2, \frac{\eta_k^3(n-b)^2}{b^2}\tilde{L}_0^g \sigma_*^2 \right\}. \tag{10}$$

*Proof.* Proceeding as in the proof of Lemma 7, we have

$$\frac{\eta_k}{n} \sum_{i=1}^m \left\langle \boldsymbol{E}_b^\top \left( \boldsymbol{v}^{(i)} - \boldsymbol{y}_k^{(i)} \right), \boldsymbol{x}_k - \boldsymbol{x}_{k-1,i} \right\rangle$$

$$= \frac{\eta_k}{n} \sum_{i=1}^m \left\langle \boldsymbol{E}_b^\top \left( \boldsymbol{v}^{(i)} - \boldsymbol{y}_k^{(i)} \right), \sum_{j=i}^m (\boldsymbol{x}_{k-1,j+1} - \boldsymbol{x}_{k-1,j}) \right\rangle$$

$$= -\frac{\eta_k^2}{bn} \sum_{i=1}^m \left\langle \boldsymbol{E}^\top \boldsymbol{I}_{(di)}(\boldsymbol{v} - \boldsymbol{y}_k), \boldsymbol{E}^\top \boldsymbol{I}_{bd(i-1)\uparrow}\boldsymbol{y}_k \right\rangle$$

$$= \underbrace{-\frac{\eta_k^2}{bn} \sum_{i=1}^m \left\langle \boldsymbol{E}^\top \boldsymbol{I}_{(di)}(\boldsymbol{v} - \boldsymbol{y}_k), \boldsymbol{E}^\top \boldsymbol{I}_{bd(i-1)\uparrow}(\boldsymbol{y}_k - \boldsymbol{y}_*) \right\rangle}_{\mathcal{I}_1}$$

$$\underbrace{-\frac{\eta_k^2}{bn} \sum_{i=1}^m \left\langle \boldsymbol{E}^\top \boldsymbol{I}_{(di)}(\boldsymbol{v} - \boldsymbol{y}_k), \boldsymbol{E}^\top \boldsymbol{I}_{bd(i-1)\uparrow}\boldsymbol{y}_* \right\rangle}_{\mathcal{I}_2}.$$

For both terms $\mathcal{I}_1$ and $\mathcal{I}_2$, we apply Young's inequality with $\alpha = 2\eta_k \tilde{L}_0^g$ and obtain

$$
\begin{aligned}
\mathcal{I}_1 &\leq \frac{\eta_k^2 \alpha}{2bn} \sum_{i=1}^{m} \|\boldsymbol{E}^\top \boldsymbol{I}_{bd(i-1)\uparrow}(\boldsymbol{y}_k - \boldsymbol{y}_*)\|_2^2 + \frac{\eta_k^2}{2bn\alpha} \sum_{i=1}^{m} \|\boldsymbol{E}^\top \boldsymbol{I}_{(di)}(\boldsymbol{v} - \boldsymbol{y}_k)\|_2^2 \\
&\leq \frac{\eta_k^2 n\alpha}{2b^2} \hat{L}_0^g \|\boldsymbol{y}_k - \boldsymbol{y}_*\|_{\boldsymbol{\Lambda}^{-1}}^2 + \frac{\eta_k^2}{2n\alpha} \tilde{L}_0^g \|\boldsymbol{v} - \boldsymbol{y}_k\|_{\boldsymbol{\Lambda}^{-1}}^2 \\
&= \frac{\eta_k^3 n}{b^2} \hat{L}_0^g \tilde{L}_0^g \|\boldsymbol{y}_k - \boldsymbol{y}_*\|_{\boldsymbol{\Lambda}^{-1}}^2 + \frac{\eta_k}{4n} \|\boldsymbol{v} - \boldsymbol{y}_k\|_{\boldsymbol{\Lambda}^{-1}}^2,
\end{aligned} \tag{11}
$$

and

$$
\begin{aligned}
\mathcal{I}_2 &\leq \frac{\eta_k^2 \alpha}{2bn} \sum_{i=1}^{m} \|\boldsymbol{E}^\top \boldsymbol{I}_{bd(i-1)\uparrow}\boldsymbol{y}_*\|_2^2 + \frac{\eta_k^2}{2bn\alpha} \sum_{i=1}^{m} \|\boldsymbol{E}^\top \boldsymbol{I}_{(di)}(\boldsymbol{v} - \boldsymbol{y}_k)\|_2^2 \\
&\leq \frac{\eta_k^2 \alpha}{2bn} \sum_{i=1}^{m} \|\boldsymbol{E}^\top \boldsymbol{I}_{bd(i-1)\uparrow}\boldsymbol{y}_*\|_2^2 + \frac{\eta_k^2}{2n\alpha} \tilde{L}_0^g \|\boldsymbol{v} - \boldsymbol{y}_k\|_{\boldsymbol{\Lambda}^{-1}}^2 \\
&= \frac{\eta_k^3 \tilde{L}_0^g}{nb} \sum_{i=1}^{m} \|\boldsymbol{E}^\top \boldsymbol{I}_{bd(i-1)\uparrow}\boldsymbol{y}_*\|_2^2 + \frac{\eta_k}{4n} \|\boldsymbol{v} - \boldsymbol{y}_k\|_{\boldsymbol{\Lambda}^{-1}}^2.
\end{aligned} \tag{12}
$$

We now show that the term $\frac{\eta_k^3 \tilde{L}_0^g}{nb} \sum_{i=1}^{m} \|\boldsymbol{E}^\top \boldsymbol{I}_{bd(i-1)\uparrow}\boldsymbol{y}_*\|_2^2$ is no larger than either $\frac{\eta_k^3 n}{b^2} \hat{L}_0^g \tilde{L}_0^g \|\boldsymbol{y}_*\|_{\boldsymbol{\Lambda}^{-1}}^2$ or $\frac{\eta_k^3 (n-b)^2}{b^2} \tilde{L}_0^g \sigma_*^2$. This is trivial when $b = n$ as $\boldsymbol{E}^\top \boldsymbol{I}_{0\uparrow}\boldsymbol{y}_* = \sum_{i=1}^{n} \boldsymbol{y}_*^i = \boldsymbol{0}$. When $b < n$, to show the former one, we have

$$
\sum_{i=1}^{m} \|\boldsymbol{E}^\top \boldsymbol{I}_{bd(i-1)\uparrow}\boldsymbol{y}_*\|_2^2 \leq \left\| \boldsymbol{\Lambda}^{1/2} \left( \sum_{i=1}^{m} \boldsymbol{I}_{bd(i-1)\uparrow} \boldsymbol{E}\boldsymbol{E}^\top \boldsymbol{I}_{bd(i-1)\uparrow} \right) \boldsymbol{\Lambda}^{1/2} \right\|_2 \|\boldsymbol{y}_*\|_{\boldsymbol{\Lambda}^{-1}}^2
$$

$$
= mn\hat{L}_0^g \|\boldsymbol{y}_*\|_{\boldsymbol{\Lambda}^{-1}}^2 = \frac{n^2}{b} \hat{L}_0^g \|\boldsymbol{y}_*\|_{\boldsymbol{\Lambda}^{-1}}^2.
$$

To prove the latter one, we notice that

$$
\begin{aligned}
\sum_{i=1}^{m} \|\boldsymbol{E}^\top \boldsymbol{I}_{bd(i-1)\uparrow}\boldsymbol{y}_*\|_2^2 &= \sum_{i=1}^{m} \left\| \sum_{j=b(i-1)+1}^{n} \boldsymbol{y}_*^j \right\|_2^2 = \sum_{i=0}^{m-1} \left\| \sum_{j=bi+1}^{n} \boldsymbol{y}_*^j \right\|_2^2 = \sum_{i=1}^{m-1} \left\| \sum_{j=bi+1}^{n} \boldsymbol{y}_*^j \right\|_2^2 \\
&= \sum_{i=1}^{m-1} \left\| \sum_{j=1}^{bi} \boldsymbol{y}_*^j \right\|_2^2,
\end{aligned}
$$

using the fact that $\sum_{i=1}^{n} \boldsymbol{y}_*^i = \boldsymbol{0}$. Then using Young's inequality we obtain

$$
\begin{aligned}
\sum_{i=1}^{m-1} \left\| \sum_{j=1}^{bi} \boldsymbol{y}_*^j \right\|_2^2 &\leq \sum_{i=1}^{m-1} bi \sum_{j=1}^{bi} \|\boldsymbol{y}_*^j\|_2^2 \\
&\leq b(m-1) \sum_{i=1}^{m-1} \sum_{j=1}^{bi} \|\boldsymbol{y}_*^j\|_2^2 \\
&= b(m-1) \sum_{i=1}^{m-1} \sum_{j=b(i-1)+1}^{bi} (m-i)\|\boldsymbol{y}_*^j\|_2^2 \\
&\leq b(m-1)^2 \sum_{i=1}^{(m-1)b} \|\boldsymbol{y}_*^i\|_2^2.
\end{aligned}
$$

Further noticing that $\sum_{i=1}^{(m-1)b} \|\boldsymbol{y}_*^i\|_2^2 \leq \sum_{i=1}^{n} \|\boldsymbol{y}_*^i\|^2 = n\sigma_*^2$, we have

$$
\frac{\eta_k^3 \tilde{L}_0^g}{nb} \sum_{i=1}^{m} \|\boldsymbol{E}^\top \boldsymbol{I}_{bd(i-1)\uparrow}\boldsymbol{y}_*\|_2^2 \leq \frac{\eta_k^3 \tilde{L}_0^g}{nb} b(m-1)^2 n\sigma_*^2 = \frac{\eta_k^3 \tilde{L}_0^g (n-b)^2}{b^2} \sigma_*^2.
$$

The same bound also captures the case $b = n$ and leads to

$$\frac{\eta_k^3 \tilde{L}_0^g}{nb} \sum_{i=1}^m \|\boldsymbol{E}^\top \boldsymbol{I}_{bd(i-1)\uparrow} \boldsymbol{y}_*\|_2^2 \leq \min \left\{ \frac{\eta_k^3 n}{b^2} \hat{L}_0^g \tilde{L}_0^g \|\boldsymbol{y}_*\|_{\boldsymbol{\Lambda}^{-1}}^2, \frac{\eta_k^3 (n-b)^2}{b^2} \tilde{L}_0^g \sigma_*^2 \right\}. \tag{13}$$

Hence, combining Eq. (11)–(13), we obtain

$$\begin{aligned}
\mathcal{I}_2 &\leq \frac{\eta_k^3 n}{b^2} \hat{L}_0^g \tilde{L}_0^g \|\boldsymbol{y}_k - \boldsymbol{y}_*\|_{\boldsymbol{\Lambda}^{-1}}^2 + \frac{\eta_k}{2n} \|\boldsymbol{v} - \boldsymbol{y}_k\|_{\boldsymbol{\Lambda}^{-1}}^2 \\
&\quad + \min \left\{ \frac{\eta_k^3 n}{b^2} \hat{L}_0^g \tilde{L}_0^g \|\boldsymbol{y}_*\|_{\boldsymbol{\Lambda}^{-1}}^2, \frac{\eta_k^3 (n-b)^2}{b^2} \tilde{L}_0^g \sigma_*^2 \right\},
\end{aligned}$$

which finishes the proof. $\qquad \square$

We are now ready to state our convergence results for IGD in the following theorem, with its proof provided for completeness.

**Theorem 5.** *Under Assumptions 1 and 2, if $\eta_k \leq \frac{b}{n\sqrt{2\hat{L}_0^g \tilde{L}_0^g}}$ and $H_K = \sum_{k=1}^K \eta_k$, the output $\hat{\boldsymbol{x}}_K$ of Algorithm 1 with a fixed permutation satisfies*

$$H_K \big( f(\hat{\boldsymbol{x}}_K) - f(\boldsymbol{x}_*) \big) \leq \frac{b}{2n} \|\boldsymbol{x}_0 - \boldsymbol{x}_*\|_2^2 + \sum_{k=1}^K \min \left\{ \frac{\eta_k^3 n}{b^2} \hat{L}_0^g \tilde{L}_0^g \|\boldsymbol{y}_*\|_{\boldsymbol{\Lambda}^{-1}}^2, \frac{\eta_k^3 (n-b)^2}{b^2} \tilde{L}_0^g \sigma_*^2 \right\}.$$

*As a consequence, for any $\epsilon > 0$, there exists a choice of a constant step size $\eta_k = \eta$ such that $f(\hat{\boldsymbol{x}}_K) - f(\boldsymbol{x}_*) \leq \epsilon$ after $\mathcal{O}\left( \frac{n\sqrt{\hat{L}_0^g \tilde{L}_0^g} \|\boldsymbol{x}_0 - \boldsymbol{x}_*\|_2^2}{\epsilon} + \frac{\min\left\{ \sqrt{n\hat{L}_0^g \tilde{L}_0^g} \|\boldsymbol{y}_*\|_{\boldsymbol{\Lambda}^{-1}}, (n-b)\sqrt{\tilde{L}_0^g} \sigma_* \right\} \|\boldsymbol{x}_0 - \boldsymbol{x}_*\|_2^2}{\epsilon^{3/2}} \right)$ gradient queries.*

*Proof.* Combining the bounds in Lemma 10 and 8 and plugging them into Eq. (5) in Lemma 6 without random permutations, we have

$$\mathcal{E}_k \leq \left( \frac{\eta_k^3 n \hat{L}_0^g \tilde{L}_0^g}{b^2} - \frac{\eta_k}{2n} \right) \|\boldsymbol{y}_k - \boldsymbol{y}_*\|_{\boldsymbol{\Lambda}^{-1}}^2 + \min \left\{ \frac{\eta_k^3 n}{b^2} \hat{L}_0^g \tilde{L}_0^g \|\boldsymbol{y}_*\|_{\boldsymbol{\Lambda}^{-1}}^2, \frac{\eta_k^3 (n-b)^2}{b^2} \tilde{L}_0^g \sigma_*^2 \right\}.$$

If $\eta_k \leq \frac{b}{n\sqrt{2\hat{L}_0^g \tilde{L}_0^g}}$, we have $\frac{\eta_k^3 n \hat{L}_0^g \tilde{L}_0^g}{b^2} - \frac{\eta_k}{2n} \leq 0$, thus

$$\mathcal{E}_k \leq \min \left\{ \frac{\eta_k^3 n}{b^2} \hat{L}_0^g \tilde{L}_0^g \|\boldsymbol{y}_*\|_{\boldsymbol{\Lambda}^{-1}}^2, \frac{\eta_k^3 (n-b)^2}{b^2} \tilde{L}_0^g \sigma_*^2 \right\}.$$

Using the definition of $\mathcal{E}_k$ and telescoping from $k = 1$ to $K$, we obtain

$$\begin{aligned}
\sum_{k=1}^K \eta_k \mathrm{Gap}^{\boldsymbol{v}}(\boldsymbol{x}_k, \boldsymbol{y}_*) &\leq \frac{b}{2n} \|\boldsymbol{x}_* - \boldsymbol{x}_0\|_2^2 - \frac{b}{2n} \|\boldsymbol{x}_* - \boldsymbol{x}_K\|_2^2 \\
&\quad + \sum_{k=1}^K \min \left\{ \frac{\eta_k^3 n}{b^2} \hat{L}_0^g \tilde{L}_0^g \|\boldsymbol{y}_*\|_{\boldsymbol{\Lambda}^{-1}}^2, \frac{\eta_k^3 (n-b)^2}{b^2} \tilde{L}_0^g \sigma_*^2 \right\}.
\end{aligned}$$

Noticing that $\mathcal{L}(\boldsymbol{x}, \boldsymbol{v})$ is convex w.r.t. $\boldsymbol{x}$, we have $\mathrm{Gap}^{\boldsymbol{v}}(\hat{\boldsymbol{x}}_K, \boldsymbol{y}_*) \leq \sum_{k=1}^K \eta_k \mathrm{Gap}^{\boldsymbol{v}}(\boldsymbol{x}_k, \boldsymbol{y}_*)/H_K$, where $\hat{\boldsymbol{x}}_K = \sum_{k=1}^K \eta_k \boldsymbol{x}_k / H_K$ and $H_K = \sum_{k=1}^K \eta_k$, so we obtain

$$H_K \mathrm{Gap}^{\boldsymbol{v}}(\hat{\boldsymbol{x}}_K, \boldsymbol{y}_*) \leq \frac{b}{2n} \|\boldsymbol{x}_0 - \boldsymbol{x}_*\|_2^2 + \sum_{k=1}^K \min \left\{ \frac{\eta_k^3 n}{b^2} \hat{L}_0^g \tilde{L}_0^g \|\boldsymbol{y}_*\|_{\boldsymbol{\Lambda}^{-1}}^2, \frac{\eta_k^3 (n-b)^2}{b^2} \tilde{L}_0^g \sigma_*^2 \right\},$$

Further choosing $\boldsymbol{v} = \boldsymbol{y}_{\hat{\boldsymbol{x}}_K}$, we obtain

$$H_K \big( f(\hat{\boldsymbol{x}}_K) - f(\boldsymbol{x}_*) \big) \leq \frac{b}{2n} \|\boldsymbol{x}_0 - \boldsymbol{x}_*\|_2^2 + \sum_{k=1}^K \min \left\{ \frac{\eta_k^3 n}{b^2} \hat{L}_0^g \tilde{L}_0^g \|\boldsymbol{y}_*\|_{\boldsymbol{\Lambda}^{-1}}^2, \frac{\eta_k^3 (n-b)^2}{b^2} \tilde{L}_0^g \sigma_*^2 \right\}. \tag{14}$$

To analyze the individual gradient oracle complexity, we choose constant stepsizes $\eta \leq \frac{b}{n\sqrt{2\hat{L}_0^g \tilde{L}_0^g}}$ and assume $b < n$ without loss of generality, then Eq. (14) becomes

$$f(\hat{\boldsymbol{x}}_K) - f(\boldsymbol{x}_*) \leq \frac{b}{2n\eta K}\|\boldsymbol{x}_0 - \boldsymbol{x}_*\|_2^2 + \min\left\{\frac{\eta^2 n}{b^2}\hat{L}_0^g \tilde{L}_0^g \|\boldsymbol{y}_*\|_{\boldsymbol{\Lambda}^{-1}}^2, \frac{\eta^2(n-b)^2}{b^2}\tilde{L}_0^g \sigma_*^2\right\}.$$

When $\hat{L}_0^g \|\boldsymbol{y}_*\|_{\boldsymbol{\Lambda}^{-1}}^2 \leq \frac{(n-b)^2}{n}\sigma_*^2$, we set $\eta = \min\left\{\frac{b}{n\sqrt{2\hat{L}_0^g \tilde{L}_0^g}}, \left(\frac{b^3 \|\boldsymbol{x}_0-\boldsymbol{x}_*\|_2^2}{2n^2 \hat{L}_0^g \tilde{L}_0^g K \|\boldsymbol{y}_*\|_{\boldsymbol{\Lambda}^{-1}}^2}\right)^{1/3}\right\}$ and consider the following two possible cases:

- "Small $K$" case: if $\eta = \frac{b}{n\sqrt{2\hat{L}_0^g \tilde{L}_0^g}} \leq \left(\frac{b^3 \|\boldsymbol{x}_0-\boldsymbol{x}_*\|_2^2}{2n^2 \hat{L}_0^g \tilde{L}_0^g K \|\boldsymbol{y}_*\|_{\boldsymbol{\Lambda}^{-1}}^2}\right)^{1/3}$, we have

$$f(\hat{\boldsymbol{x}}_K) - f(\boldsymbol{x}_*) \leq \frac{b}{2n\eta K}\|\boldsymbol{x}_0 - \boldsymbol{x}_*\|_2^2 + \frac{\eta^2 n}{b^2}\hat{L}_0^g \tilde{L}_0^g \|\boldsymbol{y}_*\|_{\boldsymbol{\Lambda}^{-1}}^2$$

$$\leq \frac{\sqrt{\hat{L}_0^g \tilde{L}_0^g}}{\sqrt{2}K}\|\boldsymbol{x}_0 - \boldsymbol{x}_*\|_2^2 + \frac{\left(\hat{L}_0^g \tilde{L}_0^g\right)^{1/3}\|\boldsymbol{y}_*\|_{\boldsymbol{\Lambda}^{-1}}^{2/3}\|\boldsymbol{x}_0 - \boldsymbol{x}_*\|_2^{4/3}}{2^{2/3}n^{1/3}K^{2/3}}.$$

- "Large $K$" case: if $\eta = \left(\frac{b^3 \|\boldsymbol{x}_0-\boldsymbol{x}_*\|_2^2}{2n^2 \hat{L}_0^g \tilde{L}_0^g K \|\boldsymbol{y}_*\|_{\boldsymbol{\Lambda}^{-1}}^2}\right)^{1/3} \leq \frac{b}{\sqrt{2\hat{L}_0^g \tilde{L}_0^g}}$, we have

$$f(\hat{\boldsymbol{x}}_K) - f(\boldsymbol{x}_*) \leq \frac{b}{2n\eta K}\|\boldsymbol{x}_0 - \boldsymbol{x}_*\|_2^2 + \frac{\eta^2 n}{b^2}\hat{L}_0^g \tilde{L}_0^g \|\boldsymbol{y}_*\|_{\boldsymbol{\Lambda}^{-1}}^2$$

$$\leq \frac{2^{1/3}\left(\hat{L}_0^g \tilde{L}_0^g\right)^{1/3}\|\boldsymbol{y}_*\|_{\boldsymbol{\Lambda}^{-1}}^{2/3}\|\boldsymbol{x}_0 - \boldsymbol{x}_*\|_2^{4/3}}{n^{1/3}K^{2/3}}.$$

Combining these two cases, we have

$$f(\hat{\boldsymbol{x}}_K) - f(\boldsymbol{x}_*) \leq \frac{\sqrt{\hat{L}_0^g \tilde{L}_0^g}}{\sqrt{2}K}\|\boldsymbol{x}_0 - \boldsymbol{x}_*\|_2^2 + \frac{2^{1/3}\left(\hat{L}_0^g \tilde{L}_0^g\right)^{1/3}\|\boldsymbol{y}_*\|_{\boldsymbol{\Lambda}^{-1}}^{2/3}\|\boldsymbol{x}_0 - \boldsymbol{x}_*\|_2^{4/3}}{n^{1/3}K^{2/3}}.$$

Hence, to guarantee $\mathbb{E}[f(\hat{\boldsymbol{x}}_K) - f(\boldsymbol{x}_*)] \leq \epsilon$ for $\epsilon > 0$, the total number of required individual gradient evaluations will be

$$nK \geq \max\left\{\frac{n\sqrt{2\hat{L}_0^g \tilde{L}_0^g}\|\boldsymbol{x}_0 - \boldsymbol{x}_*\|_2^2}{\epsilon}, \frac{4n^{1/2}\left(\hat{L}_0^g \tilde{L}_0^g\right)^{1/2}\|\boldsymbol{y}_*\|_{\boldsymbol{\Lambda}^{-1}}\|\boldsymbol{x}_0 - \boldsymbol{x}_*\|_2^2}{\epsilon^{3/2}}\right\}. \quad (15)$$

When $\frac{(n-b)^2}{n}\sigma_*^2 \leq \hat{L}_0^g \|\boldsymbol{y}_*\|_{\boldsymbol{\Lambda}^{-1}}^2$, we set $\eta = \min\left\{\frac{b}{n\sqrt{2\hat{L}_0^g \tilde{L}_0^g}}, \left(\frac{b^3 \|\boldsymbol{x}_0-\boldsymbol{x}_*\|_2^2}{2n(n-b)^2 \tilde{L}_0^g K \sigma_*^2}\right)^{1/3}\right\}$ and consider the two cases as below:

- "Small $K$" case: if $\eta = \frac{b}{n\sqrt{2\hat{L}_0^g \tilde{L}_0^g}} \leq \left(\frac{b^3 \|\boldsymbol{x}_0-\boldsymbol{x}_*\|_2^2}{2n(n-b)^2 \tilde{L}_0^g K \sigma_*^2}\right)^{1/3}$, we have

$$f(\hat{\boldsymbol{x}}_K) - f(\boldsymbol{x}_*) \leq \frac{b}{2n\eta K}\|\boldsymbol{x}_0 - \boldsymbol{x}_*\|_2^2 + \frac{\eta^2(n-b)^2}{b^2}\tilde{L}_0^g \sigma_*^2$$

$$\leq \frac{\sqrt{\hat{L}_0^g \tilde{L}_0^g}}{\sqrt{2}K}\|\boldsymbol{x}_0 - \boldsymbol{x}_*\|_2^2 + \frac{(n-b)^{2/3}(\tilde{L}_0^g)^{1/3}\sigma_*^{2/3}\|\boldsymbol{x}_0 - \boldsymbol{x}_*\|_2^{4/3}}{2^{2/3}n^{2/3}K^{2/3}}.$$

- "Large $K$" case: if $\eta = \left(\frac{b^3 \|\boldsymbol{x}_0-\boldsymbol{x}_*\|_2^2}{2n(n-b)^2 \tilde{L}_0^g K \sigma_*^2}\right)^{1/3} \leq \frac{b}{n\sqrt{2\hat{L}_0^g \tilde{L}_0^g}}$, we have

$$f(\hat{\boldsymbol{x}}_K) - f(\boldsymbol{x}_*) \leq \frac{b}{2n\eta K}\|\boldsymbol{x}_0 - \boldsymbol{x}_*\|_2^2 + \frac{\eta^2(n-b)^2}{b^2}\tilde{L}_0^g \sigma_*^2$$

$$\leq \frac{2^{1/3}(n-b)^{2/3}(\tilde{L}_0^g)^{1/3}\sigma_*^{2/3}\|\boldsymbol{x}_0 - \boldsymbol{x}_*\|_2^{4/3}}{n^{2/3}K^{2/3}}.$$

Combining these two cases, we obtain

$$f(\hat{\boldsymbol{x}}_K) - f(\boldsymbol{x}_*) \le \frac{\sqrt{\hat{L}_0^g \tilde{L}_0^g}}{\sqrt{2}K}\|\boldsymbol{x}_0 - \boldsymbol{x}_*\|_2^2 + \frac{2^{1/3}(n-b)^{2/3}(\tilde{L}^g)^{1/3}\sigma_*^{2/3}\|\boldsymbol{x}_0 - \boldsymbol{x}_*\|_2^{4/3}}{n^{2/3}K^{2/3}}.$$

To guarantee $\mathbb{E}[f(\hat{\boldsymbol{x}}_K) - f(\boldsymbol{x}_*)] \le \epsilon$ for $\epsilon > 0$, the total number of required individual gradient evaluations will be

$$nK \ge \max\left\{\frac{n\sqrt{2\hat{L}_0^g \tilde{L}_0^g}\|\boldsymbol{x}_0 - \boldsymbol{x}_*\|_2^2}{\epsilon}, \frac{4(n-b)(\tilde{L}_0^g)^{1/2}\sigma_*\|\boldsymbol{x}_0 - \boldsymbol{x}_*\|_2^2}{\epsilon^{3/2}}\right\}. \tag{16}$$

Combining Eq. (15) and Eq. (16), we finally have

$$nK \ge \frac{n\sqrt{2\hat{L}_0^g \tilde{L}_0^g}\|\boldsymbol{x}_0 - \boldsymbol{x}_*\|_2^2}{\epsilon}$$
$$+ \min\left\{\frac{4n^{1/2}(\hat{L}_0^g \tilde{L}_0^g)^{1/2}\|\boldsymbol{y}_*\|_{\boldsymbol{\Lambda}^{-1}}\|\boldsymbol{x}_0 - \boldsymbol{x}_*\|_2^2}{\epsilon^{3/2}}, \frac{4(n-b)(\tilde{L}_0^g)^{1/2}\sigma_*\|\boldsymbol{x}_0 - \boldsymbol{x}_*\|_2^2}{\epsilon^{3/2}}\right\},$$

thus finishing the proof. □

## C   Omitted Proofs for Smooth Convex Settting From Section 3

Before proceeding to the omitted proofs for the smooth convex settings in finite-sum with linear predictors, we first recall its primal-dual reformulation, then state the specialized version of a primal-dual shuffled SGD algorithm in Algorithm 2. Recall that (PL) admits an explicit reformulation using convex conjugates of $\ell_i$:

$$\min_{\boldsymbol{x}\in\mathbb{R}^d}\max_{\boldsymbol{y}\in\mathbb{R}^n}\left\{\mathcal{L}(\boldsymbol{x},\boldsymbol{y}) := \frac{1}{n}\langle\boldsymbol{A}\boldsymbol{x},\boldsymbol{y}\rangle - \frac{1}{n}\sum_{i=1}^n \ell_i^*(\boldsymbol{y}^i) = \frac{1}{n}\sum_{i=1}^n\left(\boldsymbol{a}_i^\top \boldsymbol{x}\boldsymbol{y}^i - \ell_i^*(\boldsymbol{y}^i)\right)\right\} \tag{PL-PD}$$

where $\boldsymbol{y}_{\boldsymbol{x}}^i = \arg\max_{\boldsymbol{y}^i\in\mathbb{R}}\{\boldsymbol{y}^i\boldsymbol{a}_i^\top\boldsymbol{x} - \ell_i^*(\boldsymbol{y}^i)\}$ (different from the general smooth convex finite-sum settings in Section 2 and Appendix B). Further, for notational convenience, we assume that the partition is ordered, in the sense that for $1 \le j < j' \le m$, $\max_{i\in\mathcal{S}^j} i < \min_{i'\in\mathcal{S}^{j'}} i'.$[2] We denote by $\boldsymbol{y}^{(j)}$ the subvector of $\boldsymbol{y}\in\mathbb{R}^n$ indexed by the elements of $\mathcal{S}^j$, and by $\boldsymbol{A}^{(j)}$ the submatrix obtained from $\boldsymbol{A}\in\mathbb{R}^{n\times d}$ by selecting the rows indexed by $\mathcal{S}^j$.

Based on the formulation (PL-PD), we view shuffled SGD as a primal-dual method with block coordinate updates on the dual side, as summarized in Algorithm 2, for completeness. To see the equivalence, in $i$-th inner iteration of $k$-th epoch, we first update the $i$-th block $\boldsymbol{y}_k^{(i)}\in\mathbb{R}^b$ of the dual vector $\boldsymbol{y}_{k-1}\in\mathbb{R}^n$ based on $\boldsymbol{x}_{k-1,i}$ as in Line 6. Since the dual update has a decomposable structure, this maximization step corresponds to computing the (sub)gradients $\{\ell'_{\pi_j^{(k)}}(\boldsymbol{a}_{\pi_j^{(k)}}^\top \boldsymbol{x}_{k-1,i})\}_{j=b(i-1)+1}^{bi}$ at $\boldsymbol{x}_{k-1,i}$ for the batch of individual losses indexed by $\{\pi_j^{(k)}\}_{j=b(i-1)+1}^{bi}$. Then in Line 7, we perform a minimization step using $\boldsymbol{y}_k^{(i)}$ to compute $\boldsymbol{x}_{k-1,i+1}$ on the primal side. Combining these two steps, we have $\boldsymbol{x}_{k-1,i+1} = \boldsymbol{x}_{k-1,i} - \frac{\eta_k}{b}\sum_{j=b(i-1)+1}^{bi}\ell'_{\pi_j^{(k)}}(\boldsymbol{a}_{\pi_j^{(k)}}^\top \boldsymbol{x}_{k-1,i})\boldsymbol{a}_{\pi_j^{(k)}}$, which is exactly the *original primal shuffled SGD updating scheme*.

---

[2]This is without loss of generality, as it can be achieved by reordering the rows in the data matrix.

**Algorithm 2** Shuffled SGD (Primal-Dual View)

---

1: **Input:** Initial point $\boldsymbol{x}_0 \in \mathbb{R}^d$, batch size $b > 0$, step size $\{\eta_k\} > 0$, number of epochs $K > 0$
2: **for** $k = 1$ to $K$ **do**
3:  Generate any permutation $\pi^{(k)}$ of $[n]$ (either deterministic or random)
4:  $\boldsymbol{x}_{k-1,1} = \boldsymbol{x}_{k-1}$
5:  **for** $i = 1$ to $m$ **do**
6:    $\boldsymbol{y}_k^{(i)} = \arg\max_{\boldsymbol{y} \in \mathbb{R}^b} \big\{ \boldsymbol{y}^\top \boldsymbol{A}_k^{(i)} \boldsymbol{x}_{k-1,i} - \sum_{j=1}^b \ell^*_{\pi^{(k)}_{b(i-1)+j}} (\boldsymbol{y}^j) \big\}$
7:    $\boldsymbol{x}_{k-1,i+1} = \arg\min_{\boldsymbol{x} \in \mathbb{R}^d} \big\{ \boldsymbol{y}_k^{(i)\top} \boldsymbol{A}_k^{(i)} \boldsymbol{x} + \frac{b}{2\eta_k} \|\boldsymbol{x} - \boldsymbol{x}_{k-1,i}\|^2 \big\}$
8:  **end for**
9:  $\boldsymbol{x}_k = \boldsymbol{x}_{k-1,m+1},\ \boldsymbol{y}_k = \big( \boldsymbol{y}_k^{(1)}, \boldsymbol{y}_k^{(2)}, \ldots, \boldsymbol{y}_k^{(m)} \big)^\top$
10: **end for**
11: **Return:** $\hat{\boldsymbol{x}}_K = \sum_{k=1}^K \eta_k \boldsymbol{x}_k / \sum_{k=1}^K \eta_k$

---

### C.1 Omitted Proofs for the Random Reshuffling/Shuffle-Once Schemes

**Lemma 9.** *Given $\{\boldsymbol{y}_k^{(i)}\}_{i=1}^m$ and $\{\boldsymbol{x}_{k-1,i}\}_{i=1}^{m+1}$ generated by Algorithm 2 for $k \in [K]$, let $\mathcal{E}_k := \eta_k \mathrm{Gap}^{\boldsymbol{v}}(\boldsymbol{x}_k, \boldsymbol{y}_*) + \frac{b}{2n}\|\boldsymbol{x}_* - \boldsymbol{x}_k\|_2^2 - \frac{b}{2n}\|\boldsymbol{x}_* - \boldsymbol{x}_{k-1}\|_2^2$. If Assumption 3 holds, then*

$$
\begin{aligned}
\mathcal{E}_k &\le \frac{\eta_k}{n} \sum_{i=1}^m \boldsymbol{y}_k^{(i)\top} \boldsymbol{A}_k^{(i)} (\boldsymbol{x}_k - \boldsymbol{x}_{k-1,i+1}) \\
&\quad + \frac{\eta_k}{n} \sum_{i=1}^m \big( \boldsymbol{v}_k^{(i)} - \boldsymbol{y}_k^{(i)} \big)^\top \boldsymbol{A}_k^{(i)} (\boldsymbol{x}_k - \boldsymbol{x}_{k-1,i}) \\
&\quad - \frac{\eta_k}{2n} \|\boldsymbol{y}_k - \boldsymbol{v}_k\|_{\boldsymbol{\Lambda}_k^{-1}}^2 - \frac{\eta_k}{2n} \|\boldsymbol{y}_k - \boldsymbol{y}_{*,k}\|_{\boldsymbol{\Lambda}_k^{-1}}^2 \\
&\quad - \frac{b}{2n} \sum_{i=1}^m \|\boldsymbol{x}_{k-1,i} - \boldsymbol{x}_{k-1,i+1}\|_2^2,
\end{aligned}
\tag{17}
$$

*Proof.* By Line 6 in Alg. 2, we have $\boldsymbol{y}_k^{(i)} = \arg\max_{\boldsymbol{y} \in \mathbb{R}^b} \big\{ \boldsymbol{y}^\top \boldsymbol{A}_k^{(i)} \boldsymbol{x}_{k-1,i} - \sum_{j=1}^b \ell^*_{\pi^{(k)}_{b(i-1)+j}} (\boldsymbol{y}^j) \big\}$ for $i \in [m]$. Notice that since

$$
\boldsymbol{y}^\top \boldsymbol{A}_k^{(i)} \boldsymbol{x}_{k-1,i} - \sum_{j=1}^b \ell^*_{\pi^{(k)}_{b(i-1)+j}} (\boldsymbol{y}^j) = \sum_{j=1}^b \Big( \boldsymbol{y}^j \boldsymbol{a}_{\pi^{(k)}_{b(i-1)+j}}^\top \boldsymbol{x}_{k-1,i} - \ell^*_{\pi^{(k)}_{b(i-1)+j}} (\boldsymbol{y}^j) \Big)
$$

is separable, we have $\boldsymbol{y}_k^j = \arg\max_{y \in \mathbb{R}} \{ y \boldsymbol{a}_{\pi_j^{(k)}}^\top \boldsymbol{x}_{k-1,i} - \ell^*_{\pi_j^{(k)}}(y) \}$ for $b(i-1) + 1 \le j \le bi$, thus $\boldsymbol{a}_{\pi_j^{(k)}}^\top \boldsymbol{x}_{k-1,i} \in \partial \ell^*_{\pi_j^{(k)}}(\boldsymbol{y}_k^j)$. Since $\ell_i^*$ is $\frac{1}{L_i}$-strongly convex by Assumption 3, then by Lemma 4 we obtain for $b(i-1) + 1 \le j \le bi$

$$
\ell^*_{\pi_j^{(k)}} \big( \boldsymbol{v}_k^j \big) \ge \ell^*_{\pi_j^{(k)}} (\boldsymbol{y}_k^j) + \boldsymbol{a}_{\pi_j^{(k)}}^\top \boldsymbol{x}_{k-1,i} \big( \boldsymbol{v}_k^j - \boldsymbol{y}_k^j \big) + \frac{1}{2 L_{\pi_j^{(k)}}} \big( \boldsymbol{v}_k^j - \boldsymbol{y}_k^j \big)^2,
$$

which leads to

$$
\begin{aligned}
\mathcal{L}(\boldsymbol{x}_k, \boldsymbol{v}) &= \frac{1}{n} \sum_{i=1}^m \Big( \boldsymbol{v}_k^{(i)\top} \boldsymbol{A}_k^{(i)} \boldsymbol{x}_{k-1,i} - \sum_{j=b(i-1)+1}^{bi} \ell^*_{\pi_j^{(k)}} (\boldsymbol{v}_k^j) \Big) + \frac{1}{n} \sum_{i=1}^m \boldsymbol{v}_k^{(i)\top} \boldsymbol{A}_k^{(i)} (\boldsymbol{x}_k - \boldsymbol{x}_{k-1,i}) \\
&\le \frac{1}{n} \sum_{i=1}^m \Big( \boldsymbol{y}_k^{(i)\top} \boldsymbol{A}_k^{(i)} \boldsymbol{x}_{k-1,i} - \sum_{j=b(i-1)+1}^{bi} \ell^*_{\pi_j^{(k)}} (\boldsymbol{y}_k^j) \Big) + \frac{1}{n} \sum_{i=1}^m \boldsymbol{v}_k^{(i)\top} \boldsymbol{A}_k^{(i)} (\boldsymbol{x}_k - \boldsymbol{x}_{k-1,i}) \\
&\quad - \frac{1}{2n} \|\boldsymbol{y}_k - \boldsymbol{v}_k\|_{\boldsymbol{\Lambda}_k^{-1}}^2.
\end{aligned}
\tag{18}
$$

Using the same argument for $\mathcal{L}(\boldsymbol{x}_*, \boldsymbol{y}_*)$ as $\boldsymbol{a}_j^\top \boldsymbol{x}_* \in \partial \ell_j^*(\boldsymbol{y}_*^j)$ for $j \in [n]$, we have

$$\mathcal{L}(\boldsymbol{x}_*, \boldsymbol{y}_*) = \frac{1}{n} \sum_{i=1}^m \left( \boldsymbol{y}_{*,k}^{(i)\top} \boldsymbol{A}_k^{(i)} \boldsymbol{x}_* - \sum_{j=b(i-1)+1}^{bi} \ell_{\pi_j^{(k)}}^* (\boldsymbol{y}_{*,k}^j) \right)$$

$$\geq \frac{1}{n} \sum_{i=1}^m \left( \boldsymbol{y}_k^{(i)\top} \boldsymbol{A}_k^{(i)} \boldsymbol{x}_* - \sum_{j=b(i-1)+1}^{bi} \ell_{\pi_j^{(k)}}^* (\boldsymbol{y}_k^j) \right) + \frac{1}{2n} \|\boldsymbol{y}_k - \boldsymbol{y}_{*,k}\|_{\boldsymbol{\Lambda}_k^{-1}}^2. \qquad (19)$$

Adding and substracting the term $\frac{b}{2n\eta_k} \sum_{i=1}^m \|\boldsymbol{x}_* - \boldsymbol{x}_{k-1,i}\|_2^2$ on the R.H.S. of Eq. (19), we obtain

$$\mathcal{L}(\boldsymbol{x}_*, \boldsymbol{y}_*) \geq \frac{1}{n} \sum_{i=1}^m \left( \boldsymbol{y}_k^{(i)\top} \boldsymbol{A}_k^{(i)} \boldsymbol{x}_* + \frac{b}{2\eta_k} \|\boldsymbol{x}_* - \boldsymbol{x}_{k-1,i}\|_2^2 - \sum_{j=b(i-1)+1}^{bi} \ell_{\pi_j^{(k)}}^* (\boldsymbol{y}_k^j) \right)$$

$$- \frac{b}{2n\eta_k} \sum_{i=1}^m \|\boldsymbol{x}_* - \boldsymbol{x}_{k-1,i}\|_2^2 + \frac{1}{2n} \|\boldsymbol{y}_k - \boldsymbol{y}_{*,k}\|_{\boldsymbol{\Lambda}_k^{-1}}^2.$$

By Line 7 of Alg. 2, we have $\boldsymbol{x}_{k-1,i+1} = \arg\min_{\boldsymbol{x} \in \mathbb{R}^d} \left\{ \boldsymbol{y}_k^{(i)\top} \boldsymbol{A}_k^{(i)} \boldsymbol{x} + \frac{b}{2\eta_k} \|\boldsymbol{x} - \boldsymbol{x}_{k-1,i}\|_2^2 \right\}$. Further noticing that $\phi_k^{(i)}(\boldsymbol{x}) := \boldsymbol{y}_k^{(i)\top} \boldsymbol{A}_k^{(i)} \boldsymbol{x} + \frac{b}{2\eta_k} \|\boldsymbol{x} - \boldsymbol{x}_{k-1,i}\|_2^2$ is $\frac{b}{\eta_k}$-strongly convex w.r.t. $\boldsymbol{x}$ and $\nabla \phi_k^{(i)}(\boldsymbol{x}_{k-1,i+1}) = \boldsymbol{0}$, we have

$$\phi_k^{(i)}(\boldsymbol{x}_*) \geq \phi_k^{(i)}(\boldsymbol{x}_{k-1,i+1}) + \frac{b}{2\eta_k} \|\boldsymbol{x}_* - \boldsymbol{x}_{k-1,i+1}\|_2^2,$$

which leads to

$$\mathcal{L}(\boldsymbol{x}_*, \boldsymbol{y}_*) \geq \frac{1}{n} \sum_{i=1}^m \left( \boldsymbol{y}_k^{(i)\top} \boldsymbol{A}_k^{(i)} \boldsymbol{x}_{k-1,i+1} + \frac{b}{2\eta_k} \|\boldsymbol{x}_{k-1,i+1} - \boldsymbol{x}_{k-1,i}\|_2^2 - \sum_{j=b(i-1)+1}^{bi} \ell_{\pi_j^{(k)}}^* (\boldsymbol{y}_k^j) \right)$$

$$+ \frac{b}{2n\eta_k} \sum_{i=1}^m \left( \|\boldsymbol{x}_* - \boldsymbol{x}_{k-1,i+1}\|_2^2 - \|\boldsymbol{x}_* - \boldsymbol{x}_{k-1,i}\|_2^2 \right) + \frac{1}{2n} \|\boldsymbol{y}_k - \boldsymbol{y}_{*,k}\|_{\boldsymbol{\Lambda}_k^{-1}}^2$$

$$\overset{(i)}{=} \frac{1}{n} \sum_{i=1}^m \left( \boldsymbol{y}_k^{(i)\top} \boldsymbol{A}_k^{(i)} \boldsymbol{x}_{k-1,i+1} + \frac{b}{2\eta_k} \|\boldsymbol{x}_{k-1,i+1} - \boldsymbol{x}_{k-1,i}\|_2^2 - \sum_{j=b(i-1)+1}^{bi} \ell_{\pi_j^{(k)}}^* (\boldsymbol{y}_k^j) \right)$$

$$+ \frac{b}{2n\eta_k} \|\boldsymbol{x}_k - \boldsymbol{x}_*\|_2^2 - \frac{b}{2n\eta_k} \|\boldsymbol{x}_{k-1} - \boldsymbol{x}_*\|_2^2 + \frac{1}{2n} \|\boldsymbol{y}_k - \boldsymbol{y}_{*,k}\|_{\boldsymbol{\Lambda}_k^{-1}}^2, \qquad (20)$$

where we telescope from $i = 1$ to $m$ for the term $\sum_{i=1}^m \left( \|\boldsymbol{x}_* - \boldsymbol{x}_{k-1,i+1}\|_2^2 - \|\boldsymbol{x}_* - \boldsymbol{x}_{k-1,i}\|_2^2 \right)$, and use the definitions that $\boldsymbol{x}_k = \boldsymbol{x}_{k-1,m+1}$ and $\boldsymbol{x}_{k-1} = \boldsymbol{x}_{k-1,1}$ for $(i)$.

Combining the bounds from Eq. (18) and Eq. (20) and denoting

$$\mathcal{E}_k := \eta_k \big( \mathcal{L}(\boldsymbol{x}_k, \boldsymbol{v}) - \mathcal{L}(\boldsymbol{x}_*, \boldsymbol{y}_*) \big) + \frac{b}{2n} \|\boldsymbol{x}_* - \boldsymbol{x}_k\|_2^2 - \frac{b}{2n} \|\boldsymbol{x}_* - \boldsymbol{x}_{k-1}\|_2^2,$$

we obtain

$$\mathcal{E}_k \leq \frac{\eta_k}{n} \sum_{i=1}^m \boldsymbol{y}_k^{(i)\top} \boldsymbol{A}_k^{(i)} (\boldsymbol{x}_{k-1,i} - \boldsymbol{x}_{k-1,i+1}) + \frac{\eta_k}{n} \sum_{i=1}^m \boldsymbol{v}_k^{(i)\top} \boldsymbol{A}_k^{(i)} (\boldsymbol{x}_k - \boldsymbol{x}_{k-1,i})$$

$$- \frac{\eta_k}{2n} \|\boldsymbol{y}_k - \boldsymbol{v}_k\|_{\boldsymbol{\Lambda}_k^{-1}}^2 - \frac{\eta_k}{2n} \|\boldsymbol{y}_k - \boldsymbol{y}_{*,k}\|_{\boldsymbol{\Lambda}_k^{-1}}^2 - \frac{b}{2n} \sum_{i=1}^m \|\boldsymbol{x}_{k-1,i} - \boldsymbol{x}_{k-1,i+1}\|_2^2$$

$$= \frac{\eta_k}{n} \sum_{i=1}^m \boldsymbol{y}_k^{(i)\top} \boldsymbol{A}_k^{(i)} (\boldsymbol{x}_k - \boldsymbol{x}_{k-1,i+1}) + \frac{\eta_k}{n} \sum_{i=1}^m (\boldsymbol{v}_k^{(i)} - \boldsymbol{y}_k^{(i)})^\top \boldsymbol{A}_k^{(i)} (\boldsymbol{x}_k - \boldsymbol{x}_{k-1,i})$$

$$- \frac{\eta_k}{2n} \|\boldsymbol{y}_k - \boldsymbol{v}_k\|_{\boldsymbol{\Lambda}_k^{-1}}^2 - \frac{\eta_k}{2n} \|\boldsymbol{y}_k - \boldsymbol{y}_{*,k}\|_{\boldsymbol{\Lambda}_k^{-1}}^2 - \frac{b}{2n} \sum_{i=1}^m \|\boldsymbol{x}_{k-1,i} - \boldsymbol{x}_{k-1,i+1}\|_2^2,$$

thus completing the proof. $\qquad\qquad\qquad\square$

**Lemma 10.** *For any $k \in [K]$, the iterates $\{\boldsymbol{y}_k^{(i)}\}_{i=1}^m$ and $\{\boldsymbol{x}_{k-1,i}\}_{i=1}^{m+1}$ in Algorithm 2 satisfy*

$$\mathcal{T}_1 = \frac{b}{2n} \sum_{i=1}^m \|\boldsymbol{x}_{k-1,i} - \boldsymbol{x}_{k-1,i+1}\|_2^2 - \frac{b}{2n} \|\boldsymbol{x}_{k-1} - \boldsymbol{x}_k\|_2^2.$$

*Proof.* By Line 7 in Alg. 2, we have $\boldsymbol{A}_k^{(i)\top} \boldsymbol{y}_k^{(i)} = \frac{b}{\eta_k}(\boldsymbol{x}_{k-1,i} - \boldsymbol{x}_{k-1,i+1})$. Further noticing that $\boldsymbol{x}_k - \boldsymbol{x}_{k-1,i+1} = -\sum_{j=i+1}^m (\boldsymbol{x}_{k-1,j} - \boldsymbol{x}_{k-1,j+1})$, we obtain

$$
\begin{aligned}
\mathcal{T}_1 &:= \frac{\eta_k}{n} \sum_{i=1}^m \boldsymbol{y}_k^{(i)\top} \boldsymbol{A}_k^{(i)}(\boldsymbol{x}_k - \boldsymbol{x}_{k-1,i+1}) \\
&= -\frac{b}{n} \sum_{i=1}^{m-1} \sum_{j=i+1}^m \langle \boldsymbol{x}_{k-1,i} - \boldsymbol{x}_{k-1,i+1}, \boldsymbol{x}_{k-1,j} - \boldsymbol{x}_{k-1,j+1} \rangle \\
&= \frac{b}{2n} \sum_{i=1}^m \|\boldsymbol{x}_{k-1,i} - \boldsymbol{x}_{k-1,i+1}\|^2 - \frac{b}{2n} \left\| \sum_{i=1}^m (\boldsymbol{x}_{k-1,i} - \boldsymbol{x}_{k-1,i+1}) \right\|^2,
\end{aligned}
$$

thus completing the proof. $\square$

**Lemma 11.** *Under Assumption 4, for any $k \in [K]$, the iterates $\{\boldsymbol{y}_k^{(i)}\}_{i=1}^m$ and $\{\boldsymbol{x}_{k-1,i}\}_{i=1}^{m+1}$ generated by Algorithm 2 with uniformly random shuffling satisfy*

$$\mathbb{E}[\mathcal{T}_2] \leq \mathbb{E}\left[ \frac{\eta_k^3 n \hat{L}_{\pi^{(k)}} \tilde{L}_{\pi^{(k)}}}{b^2} \|\boldsymbol{y}_k - \boldsymbol{y}_{*,k}\|_{\boldsymbol{\Lambda}_k^{-1}}^2 + \frac{\eta_k}{2n} \|\boldsymbol{v}_k - \boldsymbol{y}_k\|_{\boldsymbol{\Lambda}_k^{-1}}^2 \right] + \frac{\eta_k^3 \tilde{L}(n-b)(n+b)}{6b^2(n-1)} \sigma_*^2.$$

*Proof.* By Line 7 in Alg. 2, we have $\boldsymbol{x}_{k-1,i} - \boldsymbol{x}_{k-1,i+1} = \frac{\eta_k}{b} \boldsymbol{A}_k^{(i)\top} \boldsymbol{y}_k^{(i)}$. Using the definition of $\boldsymbol{I}_{j\uparrow}$ for $0 \leq j \leq n-1$ as in Section 1, we obtain

$$\boldsymbol{x}_k - \boldsymbol{x}_{k-1,i} = -\sum_{j=i}^m (\boldsymbol{x}_{k-1,j} - \boldsymbol{x}_{k-1,j+1}) = -\frac{\eta_k}{b} \sum_{j=i}^m \boldsymbol{A}_k^{(j)\top} \boldsymbol{y}_k^{(j)} = -\frac{\eta_k}{b} \boldsymbol{A}_k \boldsymbol{I}_{b(i-1)\uparrow} \boldsymbol{y}_k.$$

Also, we have $\boldsymbol{A}_k^{(i)\top}(\boldsymbol{v}_k^{(i)} - \boldsymbol{y}_k^{(i)}) = \boldsymbol{A}_k \boldsymbol{I}_{(i)}(\boldsymbol{v}_k - \boldsymbol{y}_k)$ by the definition of $\boldsymbol{I}_{(i)}$ in Section 3. Combining these two observations, we have

$$
\begin{aligned}
\mathcal{T}_2 &:= \frac{\eta_k}{n} \sum_{i=1}^m \left( \boldsymbol{v}_k^{(i)} - \boldsymbol{y}_k^{(i)} \right)^\top \boldsymbol{A}_k^{(i)}(\boldsymbol{x}_k - \boldsymbol{x}_{k-1,i}) \\
&= -\frac{\eta_k^2}{bn} \sum_{i=1}^m \left\langle \boldsymbol{A}_k^\top \boldsymbol{I}_{b(i-1)\uparrow} \boldsymbol{y}_k, \boldsymbol{A}_k^\top \boldsymbol{I}_{(i)}(\boldsymbol{v}_k - \boldsymbol{y}_k) \right\rangle \\
&\overset{(i)}{=} -\frac{\eta_k^2}{bn} \sum_{i=1}^m \left\langle \boldsymbol{A}_k^\top \boldsymbol{I}_{b(i-1)\uparrow}(\boldsymbol{y}_k - \boldsymbol{y}_{*,k}), \boldsymbol{A}_k^\top \boldsymbol{I}_{(i)}(\boldsymbol{v}_k - \boldsymbol{y}_k) \right\rangle && (21) \\
&\quad -\frac{\eta_k^2}{bn} \sum_{i=1}^m \left\langle \boldsymbol{A}_k^\top \boldsymbol{I}_{b(i-1)\uparrow} \boldsymbol{y}_{*,k}, \boldsymbol{A}_k^\top \boldsymbol{I}_{(i)}(\boldsymbol{v}_k - \boldsymbol{y}_k) \right\rangle, && (22)
\end{aligned}
$$

where we make a decomposition w.r.t. $\boldsymbol{y}_{*,k}$ in $(i)$. For the first term in Eq. (21), we use Young's inequality for $\alpha > 0$ and have

$$
\begin{aligned}
&-\frac{\eta_k^2}{bn} \sum_{i=1}^m \left\langle \boldsymbol{A}_k^\top \boldsymbol{I}_{b(i-1)\uparrow}(\boldsymbol{y}_k - \boldsymbol{y}_{*,k}), \boldsymbol{A}_k^\top \boldsymbol{I}_{(i)}(\boldsymbol{v}_k - \boldsymbol{y}_k) \right\rangle \\
&\leq \frac{\eta_k^2 \alpha}{2bn} \sum_{i=1}^m \|\boldsymbol{A}_k^\top \boldsymbol{I}_{b(i-1)\uparrow}(\boldsymbol{y}_k - \boldsymbol{y}_{*,k})\|_2^2 + \frac{\eta_k^2}{2bn\alpha} \sum_{i=1}^m \|\boldsymbol{A}_k^\top \boldsymbol{I}_{(i)}(\boldsymbol{v}_k - \boldsymbol{y}_k)\|_2^2.
\end{aligned}
\quad (23)
$$

Expanding the squares and rearranging the terms in Eq. (23), we have

$$\frac{\eta_k^2 \alpha}{2bn} \sum_{i=1}^{m} \| \boldsymbol{A}_k^\top \boldsymbol{I}_{b(i-1)\uparrow}(\boldsymbol{y}_k - \boldsymbol{y}_{*,k}) \|_2^2$$

$$= \frac{\eta_k^2 \alpha}{2bn} \sum_{i=1}^{m} (\boldsymbol{y}_k - \boldsymbol{y}_{*,k})^\top \boldsymbol{I}_{b(i-1)\uparrow} \boldsymbol{A}_k \boldsymbol{A}_k^\top \boldsymbol{I}_{b(i-1)\uparrow}(\boldsymbol{y}_k - \boldsymbol{y}_{*,k})$$

$$= \frac{\eta_k^2 \alpha}{2bn} (\boldsymbol{y}_k - \boldsymbol{y}_{*,k})^\top \Big( \sum_{i=1}^{m} \boldsymbol{I}_{b(i-1)\uparrow} \boldsymbol{A}_k \boldsymbol{A}_k^\top \boldsymbol{I}_{b(i-1)\uparrow} \Big)(\boldsymbol{y}_k - \boldsymbol{y}_{*,k}) \qquad (24)$$

$$= \frac{\eta_k^2 \alpha}{2bn} (\boldsymbol{y}_k - \boldsymbol{y}_{*,k})^\top \boldsymbol{\Lambda}_k^{-1/2} \boldsymbol{\Lambda}_k^{1/2} \Big( \sum_{i=1}^{m} \boldsymbol{I}_{b(i-1)\uparrow} \boldsymbol{A}_k \boldsymbol{A}_k^\top \boldsymbol{I}_{b(i-1)\uparrow} \Big) \boldsymbol{\Lambda}_k^{1/2} \boldsymbol{\Lambda}_k^{-1/2} (\boldsymbol{y}_k - \boldsymbol{y}_{*,k})$$

$$\overset{(i)}{\leq} \frac{\eta_k^2 \alpha}{2bn} \Big\| \boldsymbol{\Lambda}_k^{1/2} \Big( \sum_{i=1}^{m} \boldsymbol{I}_{b(i-1)\uparrow} \boldsymbol{A}_k \boldsymbol{A}_k^\top \boldsymbol{I}_{b(i-1)\uparrow} \Big) \boldsymbol{\Lambda}_k^{1/2} \Big\|_2 \| \boldsymbol{y}_k - \boldsymbol{y}_{*,k} \|_{\boldsymbol{\Lambda}_k^{-1}}^2,$$

where we use Cauchy-Schwarz inequality for $(i)$. Using a similar argument, we also have

$$\frac{\eta_k^2}{2bn\alpha} \sum_{i=1}^{m} \| \boldsymbol{A}_k^\top \boldsymbol{I}_{(i)}(\boldsymbol{v}_k - \boldsymbol{y}_k) \|_2^2 \leq \frac{\eta_k^2}{2bn\alpha} \Big\| \boldsymbol{\Lambda}_k^{1/2} \Big( \sum_{i=1}^{m} \boldsymbol{I}_{(i)} \boldsymbol{A}_k \boldsymbol{A}_k^\top \boldsymbol{I}_{(i)} \Big) \boldsymbol{\Lambda}_k^{1/2} \Big\|_2 \| \boldsymbol{v}_k - \boldsymbol{y}_k \|_{\boldsymbol{\Lambda}_k^{-1}}^2.$$

By the definitions of $\hat{L}_{\pi^{(k)}}$ and $\tilde{L}_{\pi^{(k)}}$, and choosing $\alpha = 2\eta_k \tilde{L}_{\pi^{(k)}}$ in Eq. (23), we obtain

$$-\frac{\eta_k^2}{bn} \sum_{i=1}^{m} \big\langle \boldsymbol{A}_k^\top \boldsymbol{I}_{b(i-1)\uparrow}(\boldsymbol{y}_k - \boldsymbol{y}_{*,k}), \boldsymbol{A}_k^\top \boldsymbol{I}_{(i)}(\boldsymbol{v}_k - \boldsymbol{y}_k) \big\rangle$$

$$\leq \frac{\eta_k^3 n \hat{L}_{\pi^{(k)}} \tilde{L}_{\pi^{(k)}}}{b^2} \| \boldsymbol{y}_k - \boldsymbol{y}_{*,k} \|_{\boldsymbol{\Lambda}_k^{-1}}^2 + \frac{\eta_k}{4n} \| \boldsymbol{v}_k - \boldsymbol{y}_k \|_{\boldsymbol{\Lambda}_k^{-1}}^2. \qquad (25)$$

For the second term in Eq. (22), we apply Young's inequality with $\beta > 0$ and proceed as above:

$$-\frac{\eta_k^2}{bn} \sum_{i=1}^{m} \big\langle \boldsymbol{A}_k^\top \boldsymbol{I}_{b(i-1)\uparrow} \boldsymbol{y}_{*,k}, \boldsymbol{A}_k^\top \boldsymbol{I}_{(i)}(\boldsymbol{v}_k - \boldsymbol{y}_k) \big\rangle$$

$$\leq \frac{\eta_k^2 \beta}{2bn} \sum_{i=1}^{m} \| \boldsymbol{A}_k^\top \boldsymbol{I}_{b(i-1)\uparrow} \boldsymbol{y}_{*,k} \|_2^2 + \frac{\eta_k^2}{2bn\beta} \sum_{i=1}^{m} \| \boldsymbol{A}_k^\top \boldsymbol{I}_{(i)}(\boldsymbol{v}_k - \boldsymbol{y}_k) \|_2^2$$

$$\leq \frac{\eta_k^2 \beta}{2bn} \sum_{i=1}^{m} \| \boldsymbol{A}_k^\top \boldsymbol{I}_{b(i-1)\uparrow} \boldsymbol{y}_{*,k} \|_2^2 + \frac{\eta_k^2}{2n\beta} \tilde{L}_{\pi^{(k)}} \| \boldsymbol{v}_k - \boldsymbol{y}_k \|_{\boldsymbol{\Lambda}_k^{-1}}^2.$$

Noticing that $\tilde{L}_{\pi^{(k)}} \leq \tilde{L}$, we choose $\beta = 2\eta_k \tilde{L}$ and obtain

$$-\frac{\eta_k^2}{bn} \sum_{i=1}^{m} \big\langle \boldsymbol{A}_k^\top \boldsymbol{I}_{b(i-1)\uparrow} \boldsymbol{y}_{*,k}, \boldsymbol{A}_k^\top \boldsymbol{I}_{(i)}(\boldsymbol{v}_k - \boldsymbol{y}_k) \big\rangle$$

$$\leq \frac{\eta_k^3 \tilde{L}}{nb} \sum_{i=1}^{m} \| \boldsymbol{A}_k^\top \boldsymbol{I}_{b(i-1)\uparrow} \boldsymbol{y}_{*,k} \|_2^2 + \frac{\eta_k}{4n} \| \boldsymbol{v}_k - \boldsymbol{y}_k \|_{\boldsymbol{\Lambda}_k^{-1}}^2. \qquad (26)$$

Combining Eq. (25) and Eq. (26), we have

$$\mathcal{T}_2 \leq \frac{\eta_k^3 \tilde{L}}{nb} \sum_{i=1}^{m} \| \boldsymbol{A}_k^\top \boldsymbol{I}_{b(i-1)\uparrow} \boldsymbol{y}_{*,k} \|_2^2 + \frac{\eta_k^3 n \hat{L}_{\pi^{(k)}} \tilde{L}_{\pi^{(k)}}}{b^2} \| \boldsymbol{y}_k - \boldsymbol{y}_{*,k} \|_{\boldsymbol{\Lambda}_k^{-1}}^2 + \frac{\eta_k}{2n} \| \boldsymbol{v}_k - \boldsymbol{y}_k \|_{\boldsymbol{\Lambda}_k^{-1}}^2. \quad (27)$$

We first assume the RR scheme. Taking conditional expectation w.r.t. the randomness up to but not including $k$-th epoch, we have

$$\mathbb{E}_k[\mathcal{T}_2] \leq \frac{\eta_k^3 \tilde{L}}{nb} \mathbb{E}_k \Big[ \sum_{i=1}^{m} \| \boldsymbol{A}_k^\top \boldsymbol{I}_{b(i-1)\uparrow} \boldsymbol{y}_{*,k} \|_2^2 \Big]$$

$$+ \mathbb{E}_k \Big[ \frac{\eta_k^3 n \hat{L}_{\pi^{(k)}} \tilde{L}_{\pi^{(k)}}}{b^2} \| \boldsymbol{y}_k - \boldsymbol{y}_{*,k} \|_{\boldsymbol{\Lambda}_k^{-1}}^2 + \frac{\eta_k}{2n} \| \boldsymbol{v}_k - \boldsymbol{y}_k \|_{\boldsymbol{\Lambda}_k^{-1}}^2 \Big].$$

For the first term $\frac{\eta_k^3 \tilde{L}}{nb} \mathbb{E}_k \Big[ \sum_{i=1}^m \|\boldsymbol{A}_k^\top \boldsymbol{I}_{b(i-1)\uparrow} \boldsymbol{y}_{*,k}\|_2^2 \Big]$, the only randomness is from the random permutation $\pi^{(k)}$. In this case, each term $\boldsymbol{A}_k^\top \boldsymbol{I}_{b(i-1)\uparrow} \boldsymbol{y}_{*,k}$ can be considered as a sum of a batch sampled without replacement from $\{\boldsymbol{y}_*^j \boldsymbol{a}_j\}_{j\in[n]}$, while $\sum_{j=1}^n \boldsymbol{y}_*^j \boldsymbol{a}_j = 0$ as $\boldsymbol{x}_*$ is the minimizer, we then can use Lemma 5 and obtain

$$
\begin{aligned}
\frac{\eta_k^3 \tilde{L}}{nb} \mathbb{E}_k \Big[ \sum_{i=1}^m \|\boldsymbol{A}_k^\top \boldsymbol{I}_{b(i-1)\uparrow} \boldsymbol{y}_{*,k}\|_2^2 \Big] &\overset{(i)}{=} \frac{\eta_k^3 \tilde{L}}{nb} \sum_{i=1}^m \mathbb{E}_{\pi^{(k)}} \big[ \|\boldsymbol{A}_k^\top \boldsymbol{I}_{b(i-1)\uparrow} \boldsymbol{y}_{*,k}\|_2^2 \big] \\
&= \frac{\eta_k^3 \tilde{L}}{nb} \sum_{i=1}^m \big( n - b(i-1) \big)^2 \mathbb{E}_{\pi^{(k)}} \Big[ \Big\| \frac{\boldsymbol{A}_k^\top \boldsymbol{I}_{b(i-1)\uparrow} \boldsymbol{y}_{*,k}}{n - b(i-1)} \Big\|_2^2 \Big] \\
&\overset{(ii)}{\leq} \frac{\eta_k^3 \tilde{L}}{nb} \sum_{i=1}^m \big( n - b(i-1) \big)^2 \frac{b(i-1)}{\big( n - b(i-1) \big)(n-1)} \sigma_*^2 \\
&= \frac{\eta_k^3 \tilde{L}(n-b)(n+b)}{6b^2(n-1)} \sigma_*^2,
\end{aligned}
$$

where $(i)$ is due to the linearity of expectation, and we use our definition $\sigma_*^2 = \frac{1}{n} \sum_{j=1}^n (\boldsymbol{y}_*^j)^2 \|\boldsymbol{a}_j\|_2^2 = \mathbb{E}_j \big[ \|\boldsymbol{y}_*^j \boldsymbol{a}_j\|_2^2 \big]$ for $(ii)$. Taking expectation w.r.t. all the randomness on both sides and using the law of total expectation, we obtain

$$
\mathbb{E}[\mathcal{T}_2] \leq \mathbb{E} \Big[ \frac{\eta_k^3 n \hat{L}_{\pi^{(k)}} \tilde{L}_{\pi^{(k)}}}{b^2} \|\boldsymbol{y}_k - \boldsymbol{y}_{*,k}\|_{\boldsymbol{\Lambda}_k^{-1}}^2 + \frac{\eta_k}{2n} \|\boldsymbol{v}_k - \boldsymbol{y}_k\|_{\boldsymbol{\Lambda}_k^{-1}}^2 \Big] + \frac{\eta_k^3 \tilde{L}(n-b)(n+b)}{6b^2(n-1)} \sigma_*^2.
$$

For the SO scheme, since there is only one random permutation generated at the very beginning, we can take expectation w.r.t. all the randomness on both sides of (27), and the randomness for the term $\frac{\eta_k^3 \tilde{L}}{nb} \mathbb{E} \Big[ \sum_{i=1}^m \|\boldsymbol{A}_k^\top \boldsymbol{I}_{b(i-1)\uparrow} \boldsymbol{y}_{*,k}\|_2^2 \Big]$ is only from the initial random permutation. So the above argument still applies to this case, and we complete the proof. $\qquad\square$

**Theorem 2.** *Under Assumptions 3 and 4, if $\eta_k \leq \frac{b}{n \sqrt{2 \hat{L}_{\pi^{(k)}} \tilde{L}_{\pi^{(k)}}}}$ and $H_K = \sum_{k=1}^K \eta_k$, then the output $\hat{\boldsymbol{x}}_K$ of Alg. 1 with uniformly random (RR/SO) shuffling satisfies*

$$
\mathbb{E}[H_K(f(\hat{\boldsymbol{x}}_K) - f(\boldsymbol{x}_*))] \leq \frac{b}{2n} \|\boldsymbol{x}_0 - \boldsymbol{x}_*\|_2^2 + \sum_{k=1}^K \frac{\eta_k^3 \tilde{L}(n-b)(n+b)}{6b^2(n-1)} \sigma_*^2.
$$

*As a result, given $\epsilon > 0$, there exists a constant step size $\eta_k = \eta$ such that $\mathbb{E}[f(\hat{\boldsymbol{x}}_K) - f(\boldsymbol{x}_*)] \leq \epsilon$ after $\mathcal{O}\big( \frac{n\sqrt{\hat{L}\tilde{L}} \|\boldsymbol{x}_0 - \boldsymbol{x}_*\|_2^2}{\epsilon} + \sqrt{\frac{(n-b)(n+b)}{n(n-1)}} \frac{\sqrt{n\tilde{L}} \sigma_* \|\boldsymbol{x}_0 - \boldsymbol{x}_*\|_2^2}{\epsilon^{3/2}} \big)$ individual gradient queries.*

*Proof.* Combining the bounds in Lemma 10 and 11 and plugging them into Eq. (17), we obtain

$$
\mathbb{E}[\mathcal{E}_k] \leq \mathbb{E} \Big[ \Big( \frac{\eta_k^3 n \hat{L}_{\pi^{(k)}} \tilde{L}_{\pi^{(k)}}}{b^2} - \frac{\eta_k}{2n} \Big) \|\boldsymbol{y}_k - \boldsymbol{y}_{*,k}\|_{\boldsymbol{\Lambda}_k^{-1}}^2 \Big] + \frac{\eta_k^3 \tilde{L}(n-b)(n+b)}{6b^2(n-1)} \sigma_*^2.
$$

For the stepsize $\eta_k$ such that $\eta_k \leq \frac{b}{n \sqrt{2 \hat{L}_{\pi^{(k)}} \tilde{L}_{\pi^{(k)}}}}$, we have $\frac{\eta_k^3 n \hat{L}_{\pi^{(k)}} \tilde{L}_{\pi^{(k)}}}{b^2} - \frac{\eta_k}{2n} \leq 0$, thus

$$
\mathbb{E}[\mathcal{E}_k] \leq \frac{\eta_k^3 \tilde{L}(n-b)(n+b)}{6b^2(n-1)} \sigma_*^2.
$$

Noticing that $\mathcal{E}_k = \eta_k \text{Gap}^{\boldsymbol{v}}(\boldsymbol{x}_k, \boldsymbol{y}_*) + \frac{b}{2n} \|\boldsymbol{x}_* - \boldsymbol{x}_k\|_2^2 - \frac{b}{2n} \|\boldsymbol{x}_* - \boldsymbol{x}_{k-1}\|_2^2$ and telescoping from $k = 1$ to $K$, we have

$$
\mathbb{E} \Big[ \sum_{k=1}^K \eta_k \text{Gap}^{\boldsymbol{v}}(\boldsymbol{x}_k, \boldsymbol{y}_*) \Big] \leq \frac{b}{2n} \|\boldsymbol{x}_* - \boldsymbol{x}_0\|_2^2 - \frac{b}{2n} \mathbb{E}[\|\boldsymbol{x}_* - \boldsymbol{x}_K\|_2^2] + \sum_{k=1}^K \frac{\eta_k^3 \tilde{L}(n-b)(n+b)}{6b^2(n-1)} \sigma_*^2.
$$

Noticing that $\mathcal{L}(\boldsymbol{x}, \boldsymbol{v})$ is convex w.r.t. $\boldsymbol{x}$, we have $\mathrm{Gap}^{\boldsymbol{v}}(\hat{\boldsymbol{x}}_K, \boldsymbol{y}_*) \leq \sum_{k=1}^{K} \eta_k \mathrm{Gap}^{\boldsymbol{v}}(\boldsymbol{x}_k, \boldsymbol{y}_*) / H_K$, where $\hat{\boldsymbol{x}}_K = \sum_{k=1}^{K} \eta_k \boldsymbol{x}_k / H_K$ and $H_K = \sum_{k=1}^{K} \eta_k$, which leads to

$$\mathbb{E}\Big[ H_K \mathrm{Gap}^{\boldsymbol{v}}(\hat{\boldsymbol{x}}_K, \boldsymbol{y}_*) \Big] \leq \frac{b}{2n} \|\boldsymbol{x}_0 - \boldsymbol{x}_*\|_2^2 + \sum_{k=1}^{K} \frac{\eta_k^3 \tilde{L}(n-b)(n+b)}{6b^2(n-1)} \sigma_*^2.$$

Further choosing $\boldsymbol{v} = \boldsymbol{y}_{\hat{\boldsymbol{x}}_K}$, we obtain

$$\mathbb{E}[H_K\big(f(\hat{\boldsymbol{x}}_K) - f(\boldsymbol{x}_*)\big)] \leq \frac{b}{2n} \|\boldsymbol{x}_0 - \boldsymbol{x}_*\|_2^2 + \sum_{k=1}^{K} \frac{\eta_k^3 \tilde{L}(n-b)(n+b)}{6b^2(n-1)} \sigma_*^2. \tag{28}$$

To analyze the individual gradient oracle complexity, we choose constant stepsizes $\eta \leq \frac{b}{n\sqrt{2\hat{L}\tilde{L}}}$, then Eq. (28) will become

$$\mathbb{E}[f(\hat{\boldsymbol{x}}_K) - f(\boldsymbol{x}_*)] \leq \frac{b}{2n\eta K} \|\boldsymbol{x}_0 - \boldsymbol{x}_*\|_2^2 + \frac{\eta^2 \tilde{L}(n-b)(n+b)}{6b^2(n-1)} \sigma_*^2.$$

Without loss of generality, we assume that $b \neq n$, otherwise the method and its analysis reduce to (full) gradient descent. We consider the following two cases:

- "Small $K$" case: if $\eta = \frac{b}{n\sqrt{2\hat{L}\tilde{L}}} \leq \Big( \frac{3b^3(n-1)\|\boldsymbol{x}_0 - \boldsymbol{x}_*\|_2^2}{n(n-b)(n+b)\tilde{L}K\sigma_*^2} \Big)^{1/3}$, we have

$$\begin{aligned}
\mathbb{E}[f(\hat{\boldsymbol{x}}_K) - f(\boldsymbol{x}_*)] &\leq \frac{b}{2n\eta K} \|\boldsymbol{x}_0 - \boldsymbol{x}_*\|_2^2 + \frac{\eta^2 \tilde{L}(n-b)(n+b)}{6b^2(n-1)} \sigma_*^2 \\
&\leq \frac{\sqrt{\hat{L}\tilde{L}}}{\sqrt{2}K} \|\boldsymbol{x}_0 - \boldsymbol{x}_*\|_2^2 + \frac{1}{2} \Big( \frac{(n-b)(n+b)}{n^2(n-1)} \Big)^{1/3} \frac{\tilde{L}^{1/3}\sigma_*^{2/3}\|\boldsymbol{x}_0 - \boldsymbol{x}_*\|_2^{4/3}}{3^{1/3}K^{2/3}}.
\end{aligned}$$

- "Large $K$" case: if $\eta = \Big( \frac{3b^3(n-1)\|\boldsymbol{x}_0 - \boldsymbol{x}_*\|_2^2}{n(n-b)(n+b)\tilde{L}K\sigma_*^2} \Big)^{1/3} \leq \frac{b}{n\sqrt{2\hat{L}\tilde{L}}}$, we have

$$\begin{aligned}
\mathbb{E}[f(\hat{\boldsymbol{x}}_K) - f(\boldsymbol{x}_*)] &\leq \frac{b}{2n\eta K} \|\boldsymbol{x}_0 - \boldsymbol{x}_*\|_2^2 + \frac{\eta^2 \tilde{L}(n-b)(n+b)}{6b^2(n-1)} \sigma_*^2 \\
&\leq \Big( \frac{(n-b)(n+b)}{n^2(n-1)} \Big)^{1/3} \frac{\tilde{L}^{1/3}\sigma_*^{2/3}\|\boldsymbol{x}_0 - \boldsymbol{x}_*\|_2^{4/3}}{3^{1/3}K^{2/3}}.
\end{aligned}$$

Combining these two cases by setting $\eta = \min \Big\{ \frac{b}{n\sqrt{2\hat{L}\tilde{L}}}, \Big( \frac{3b^3(n-1)\|\boldsymbol{x}_0 - \boldsymbol{x}_*\|_2^2}{n(n-b)(n+b)\tilde{L}K\sigma_*^2} \Big)^{1/3} \Big\}$, we obtain

$$\mathbb{E}[f(\hat{\boldsymbol{x}}_K) - f(\boldsymbol{x}_*)] \leq \frac{\sqrt{\hat{L}\tilde{L}}}{\sqrt{2}K} \|\boldsymbol{x}_0 - \boldsymbol{x}_*\|_2^2 + \Big( \frac{(n-b)(n+b)}{n^2(n-1)} \Big)^{1/3} \frac{\tilde{L}^{1/3}\sigma_*^{2/3}\|\boldsymbol{x}_0 - \boldsymbol{x}_*\|_2^{4/3}}{3^{1/3}K^{2/3}}.$$

Hence, to guarantee $\mathbb{E}[f(\hat{\boldsymbol{x}}_K) - f(\boldsymbol{x}_*)] \leq \epsilon$ for $\epsilon > 0$, the total number of individual gradient evaluations will be

$$nK \geq \max \Big\{ \frac{n\sqrt{2\hat{L}\tilde{L}}\|\boldsymbol{x}_0 - \boldsymbol{x}_*\|_2^2}{\epsilon}, \Big( \frac{(n-b)(n+b)}{n-1} \Big)^{1/2} \frac{2^{3/2}\tilde{L}^{1/2}\sigma_*\|\boldsymbol{x}_0 - \boldsymbol{x}_*\|_2^2}{3^{1/2}\epsilon^{3/2}} \Big\},$$

as claimed. $\qquad\square$

## C.2 Omitted Proofs for Incremental Gradient Descenet

We now provide the proof for convergence of IGD in the smooth convex settings. We first prove the following technical lemma, which bounds the inner product term $\mathcal{T}_2 := \frac{\eta_k}{n} \sum_{i=1}^{m} \big( \boldsymbol{v}^{(i)} - \boldsymbol{y}_k^{(i)} \big)^\top \boldsymbol{A}^{(i)} (\boldsymbol{x}_k - \boldsymbol{x}_{k-1,i})$ without random permutations involved.

**Lemma 12.** *For any $k \in [K]$, the iterates $\{\boldsymbol{y}_k^{(i)}\}_{i=1}^m$ and $\{\boldsymbol{x}_{k-1,i}\}_{i=1}^{m+1}$ generated by Algorithm 2 with fixed data ordering satisfy*

$$
\begin{aligned}
\mathcal{T}_2 \leq{} & \frac{\eta_k^3 n}{b^2} \hat{L}_0 \tilde{L}_0 \|\boldsymbol{y}_k - \boldsymbol{y}_*\|_{\boldsymbol{\Lambda}^{-1}}^2 + \frac{\eta_k}{2n} \|\boldsymbol{v} - \boldsymbol{y}_k\|_{\boldsymbol{\Lambda}^{-1}}^2 \\
& + \min\left\{ \frac{\eta_k^3 n}{b^2} \hat{L}_0 \tilde{L}_0 \|\boldsymbol{y}_*\|_{\boldsymbol{\Lambda}^{-1}}^2, \frac{\eta_k^3 (n-b)^2}{b^2} \tilde{L}_0 \sigma_*^2 \right\}.
\end{aligned}
\tag{29}
$$

*Proof.* Proceeding as in Lemma 11, we have

$$
\begin{aligned}
\mathcal{T}_2 :={} & \frac{\eta_k}{n} \sum_{i=1}^m \left( \boldsymbol{v}^{(i)} - \boldsymbol{y}_k^{(i)} \right)^\top \boldsymbol{A}^{(i)} (\boldsymbol{x}_k - \boldsymbol{x}_{k-1,i}) \\
={} & -\frac{\eta_k^2}{bn} \sum_{i=1}^m \left\langle \boldsymbol{A}^\top \boldsymbol{I}_{b(i-1)\uparrow} \boldsymbol{y}_k, \boldsymbol{A}^\top \boldsymbol{I}_{(i)} (\boldsymbol{v} - \boldsymbol{y}_k) \right\rangle \\
={} & -\frac{\eta_k^2}{bn} \sum_{i=1}^m \left\langle \boldsymbol{A}^\top \boldsymbol{I}_{b(i-1)\uparrow} (\boldsymbol{y}_k - \boldsymbol{y}_*), \boldsymbol{A}^\top \boldsymbol{I}_{(i)} (\boldsymbol{v} - \boldsymbol{y}_k) \right\rangle \tag{30} \\
& -\frac{\eta_k^2}{bn} \sum_{i=1}^m \left\langle \boldsymbol{A}^\top \boldsymbol{I}_{b(i-1)\uparrow} \boldsymbol{y}_*, \boldsymbol{A}^\top \boldsymbol{I}_{(i)} (\boldsymbol{v} - \boldsymbol{y}_k) \right\rangle, \tag{31}
\end{aligned}
$$

For both terms in Eq. (30) and Eq. (31), we use Young's inequality for $\alpha = 2\eta_k \tilde{L}_0 > 0$ and proceed as in Eq. (24) to obtain

$$
\begin{aligned}
& -\frac{\eta_k^2}{bn} \sum_{i=1}^m \left\langle \boldsymbol{A}^\top \boldsymbol{I}_{b(i-1)\uparrow} (\boldsymbol{y}_k - \boldsymbol{y}_*), \boldsymbol{A}^\top \boldsymbol{I}_{(i)} (\boldsymbol{v} - \boldsymbol{y}_k) \right\rangle \\
\leq{} & \frac{\eta_k^2 \alpha}{2bn} \sum_{i=1}^m \|\boldsymbol{A}^\top \boldsymbol{I}_{b(i-1)\uparrow} (\boldsymbol{y}_k - \boldsymbol{y}_*)\|_2^2 + \frac{\eta_k^2}{2bn\alpha} \sum_{i=1}^m \|\boldsymbol{A}^\top \boldsymbol{I}_{(i)} (\boldsymbol{v} - \boldsymbol{y}_k)\|_2^2 \\
\leq{} & \frac{\eta_k^2 n\alpha}{2b^2} \hat{L}_0 \|\boldsymbol{y}_k - \boldsymbol{y}_*\|_{\boldsymbol{\Lambda}^{-1}}^2 + \frac{\eta_k^2}{2n\alpha} \tilde{L}_0 \|\boldsymbol{v} - \boldsymbol{y}_k\|_{\boldsymbol{\Lambda}^{-1}}^2 \\
={} & \frac{\eta_k^3 n}{b^2} \hat{L}_0 \tilde{L}_0 \|\boldsymbol{y}_k - \boldsymbol{y}_*\|_{\boldsymbol{\Lambda}^{-1}}^2 + \frac{\eta_k}{4n} \|\boldsymbol{v} - \boldsymbol{y}_k\|_{\boldsymbol{\Lambda}^{-1}}^2 \tag{32}
\end{aligned}
$$

and

$$
\begin{aligned}
& -\frac{\eta_k^2}{bn} \sum_{i=1}^m \left\langle \boldsymbol{A}^\top \boldsymbol{I}_{b(i-1)\uparrow} \boldsymbol{y}_*, \boldsymbol{A}^\top \boldsymbol{I}_{(i)} (\boldsymbol{v} - \boldsymbol{y}_k) \right\rangle \\
\leq{} & \frac{\eta_k^2 \alpha}{2bn} \sum_{i=1}^m \|\boldsymbol{A}^\top \boldsymbol{I}_{b(i-1)\uparrow} \boldsymbol{y}_*\|_2^2 + \frac{\eta_k^2}{2bn\alpha} \sum_{i=1}^m \|\boldsymbol{A}^\top \boldsymbol{I}_{(i)} (\boldsymbol{v} - \boldsymbol{y}_k)\|_2^2 \\
\leq{} & \frac{\eta_k^2 \alpha}{2bn} \sum_{i=1}^m \|\boldsymbol{A}^\top \boldsymbol{I}_{b(i-1)\uparrow} \boldsymbol{y}_*\|_2^2 + \frac{\eta_k^2}{2n\alpha} \tilde{L}_0 \|\boldsymbol{v} - \boldsymbol{y}_k\|_{\boldsymbol{\Lambda}^{-1}}^2 \\
={} & \frac{\eta_k^3 \tilde{L}_0}{nb} \sum_{i=1}^m \|\boldsymbol{A}^\top \boldsymbol{I}_{b(i-1)\uparrow} \boldsymbol{y}_*\|_2^2 + \frac{\eta_k}{4n} \|\boldsymbol{v} - \boldsymbol{y}_k\|_{\boldsymbol{\Lambda}^{-1}}^2, \tag{33}
\end{aligned}
$$

where again we used $\alpha = 2\eta_k \tilde{L}_0$. We then prove the term $\frac{\eta_k^3 \tilde{L}_0}{nb} \sum_{i=1}^m \|\boldsymbol{A}^\top \boldsymbol{I}_{b(i-1)\uparrow} \boldsymbol{y}_*\|_2^2$ in Eq. (33) is no larger than the minimum of $\frac{\eta_k^3 n}{b^2} \hat{L}_0 \tilde{L}_0 \|\boldsymbol{y}_*\|_{\boldsymbol{\Lambda}^{-1}}^2$ and $\frac{\eta_k^3 (n-b)^2}{b^2} \tilde{L}_0 \sigma_*^2$. Note that when $b = n$, we have $\boldsymbol{A}^\top \boldsymbol{I}_{(0)\uparrow} \boldsymbol{y}_* = 0$, so this term disappears. When $b < n$, the former one can be derived as in Eq.(24), which gives

$$
\begin{aligned}
\sum_{i=1}^m \|\boldsymbol{A}^\top \boldsymbol{I}_{b(i-1)\uparrow} \boldsymbol{y}_*\|_2^2 \leq{} & \left\| \boldsymbol{\Lambda}^{1/2} \left( \sum_{i=1}^m \boldsymbol{I}_{b(i-1)\uparrow} \boldsymbol{A}\boldsymbol{A}^\top \boldsymbol{I}_{b(i-1)\uparrow} \right) \boldsymbol{\Lambda}^{1/2} \right\|_2 \|\boldsymbol{y}_*\|_{\boldsymbol{\Lambda}^{-1}}^2 = mn\hat{L}_0 \|\boldsymbol{y}_*\|_{\boldsymbol{\Lambda}^{-1}}^2 \\
={} & \frac{n^2}{b} \hat{L}_0 \|\boldsymbol{y}_*\|_{\boldsymbol{\Lambda}^{-1}}^2.
\end{aligned}
$$

For the latter one, we notice that

$$\sum_{i=1}^{m} \|\boldsymbol{A}^\top \boldsymbol{I}_{b(i-1)\uparrow} \boldsymbol{y}_*\|_2^2 = \sum_{i=1}^{m} \Big\| \sum_{j=b(i-1)+1}^{n} \boldsymbol{y}_*^j \boldsymbol{a}_j \Big\|_2^2$$

$$= \sum_{i=0}^{m-1} \Big\| \sum_{j=bi+1}^{n} \boldsymbol{y}_*^j \boldsymbol{a}_j \Big\|_2^2$$

$$= \sum_{i=1}^{m-1} \Big\| \sum_{j=bi+1}^{n} \boldsymbol{y}_*^j \boldsymbol{a}_j \Big\|_2^2 = \sum_{i=1}^{m-1} \Big\| \sum_{j=1}^{bi} \boldsymbol{y}_*^j \boldsymbol{a}_j \Big\|_2^2,$$

by using the fact that $\sum_{j=1}^{n} \boldsymbol{y}_*^j \boldsymbol{a}_j = 0$. Using Young's inequality, we have

$$\sum_{i=1}^{m-1} \Big\| \sum_{j=1}^{bi} \boldsymbol{y}_*^j \boldsymbol{a}_j \Big\|_2^2 \le \sum_{i=1}^{m-1} bi \sum_{j=1}^{bi} \|\boldsymbol{y}_*^j \boldsymbol{a}_j\|_2^2$$

$$\le b(m-1) \sum_{i=1}^{m-1} \sum_{j=1}^{bi} \|\boldsymbol{y}_*^j \boldsymbol{a}_j\|_2^2$$

$$= b(m-1) \sum_{i=1}^{m-1} \sum_{j=b(i-1)+1}^{bi} (m-i) \|\boldsymbol{y}_*^j \boldsymbol{a}_j\|_2^2$$

$$\le b(m-1)^2 \sum_{i=1}^{(m-1)b} \|\boldsymbol{y}_*^j \boldsymbol{a}_j\|_2^2.$$

By the definition that $\sigma_*^2 = \frac{1}{n} \sum_{j=1}^{n} \|\boldsymbol{y}_*^j \boldsymbol{a}_j\|_2^2$ and $\sum_{i=i}^{(m-1)b} \|\boldsymbol{y}_*^j \boldsymbol{a}_j\|_2^2 \le \sum_{j=1}^{n} \|\boldsymbol{y}_*^j \boldsymbol{a}_j\|_2^2 = n\sigma_*^2$, we obtain

$$\frac{\eta_k^3 \tilde{L}_0}{nb} \sum_{i=1}^{m} \|\boldsymbol{A}^\top \boldsymbol{I}_{b(i-1)\uparrow} \boldsymbol{y}_*\|_2^2 \le \frac{\eta_k^3 \tilde{L}_0}{b} b(m-1)^2 \sigma_*^2 = \frac{\eta_k^3 (n-b)^2}{b^2} \tilde{L}_0 \sigma_*^2. \tag{34}$$

Note that the bound in Eq. (34) equals to zero when $b = n$, which recovers the case of full gradient descent, so we have

$$\frac{\eta_k^3 \tilde{L}_0}{nb} \sum_{i=1}^{m} \|\boldsymbol{A}^\top \boldsymbol{I}_{b(i-1)\uparrow} \boldsymbol{y}_*\|_2^2 \le \min\Big\{ \frac{\eta_k^3 n}{b^2} \hat{L}_0 \tilde{L}_0 \|\boldsymbol{y}_*\|_{\boldsymbol{\Lambda}^{-1}}^2, \frac{\eta_k^3 (n-b)^2}{b^2} \tilde{L}_0 \sigma_*^2 \Big\}. \tag{35}$$

Combining Eq. (32)–(35), we obtain

$$\mathcal{T}_2 \le \frac{\eta_k^3 n}{b^2} \hat{L}_0 \tilde{L}_0 \|\boldsymbol{y}_k - \boldsymbol{y}_*\|_{\boldsymbol{\Lambda}^{-1}}^2 + \frac{\eta_k}{2n} \|\boldsymbol{v} - \boldsymbol{y}_k\|_{\boldsymbol{\Lambda}^{-1}}^2 + \min\Big\{ \frac{\eta_k^3 n}{b^2} \hat{L}_0 \tilde{L}_0 \|\boldsymbol{y}_*\|_{\boldsymbol{\Lambda}^{-1}}^2, \frac{\eta_k^3 (n-b)^2}{b^2} \tilde{L}_0 \sigma_*^2 \Big\},$$

thus finishing the proof. $\qquad \square$

**Theorem 6.** *Under Assumptions 3 and 4, if $\eta_k \le \frac{b}{n\sqrt{2\hat{L}_0 \tilde{L}_0}}$ and $H_K = \sum_{k=1}^{K} \eta_k$, the output $\hat{\boldsymbol{x}}_K$ of Alg. 2 with a fixed permutation satisfies*

$$H_K \big( f(\hat{\boldsymbol{x}}_K) - f(\boldsymbol{x}_*) \big) \le \frac{b}{2n} \|\boldsymbol{x}_0 - \boldsymbol{x}_*\|_2^2 + \sum_{k=1}^{K} \min\Big\{ \frac{\eta_k^3 n}{b^2} \hat{L}_0 \tilde{L}_0 \|\boldsymbol{y}_*\|_{\boldsymbol{\Lambda}^{-1}}^2, \frac{\eta_k^3 (n-b)^2}{b^2} \tilde{L}_0 \sigma_*^2 \Big\}.$$

*As a consequence, given $\epsilon > 0$, there exists a constant step size $\eta_k = \eta$ such that $f(\hat{\boldsymbol{x}}_K) - f(\boldsymbol{x}_*) \le \epsilon$ after the number of gradient queries bounded by $\mathcal{O}\Big( \frac{n\sqrt{\hat{L}_0 \tilde{L}_0}\|\boldsymbol{x}_0 - \boldsymbol{x}_*\|_2^2}{\epsilon} + \frac{\min\big\{ \sqrt{n\hat{L}_0 \tilde{L}_0}\|\boldsymbol{y}_*\|_{\boldsymbol{\Lambda}^{-1}}, (n-b)\sqrt{\tilde{L}_0}\sigma_* \big\} \|\boldsymbol{x}_0 - \boldsymbol{x}_*\|_2^2}{\epsilon^{3/2}} \Big)$.*

*Proof.* Proceeding as in Lemmas 9 and 10, but without random permutations, we have

$$\mathcal{E}_k \leq \frac{\eta_k}{n}\sum_{i=1}^{m} \boldsymbol{y}_k^{(i)\top} \boldsymbol{A}^{(i)}(\boldsymbol{x}_k - \boldsymbol{x}_{k-1,i+1}) + \frac{\eta_k}{n}\sum_{i=1}^{m}\big(\boldsymbol{v}^{(i)} - \boldsymbol{y}_k^{(i)}\big)^\top \boldsymbol{A}^{(i)}(\boldsymbol{x}_k - \boldsymbol{x}_{k-1,i})$$

$$- \frac{\eta_k}{2n}\|\boldsymbol{y}_k - \boldsymbol{v}\|_{\boldsymbol{\Lambda}^{-1}}^2 - \frac{\eta_k}{2n}\|\boldsymbol{y}_k - \boldsymbol{y}_*\|_{\boldsymbol{\Lambda}^{-1}}^2 - \frac{b}{2n}\sum_{i=1}^{m}\|\boldsymbol{x}_{k-1,i} - \boldsymbol{x}_{k-1,i+1}\|_2^2$$

$$\leq \frac{\eta_k}{n}\sum_{i=1}^{m}\big(\boldsymbol{v}^{(i)} - \boldsymbol{y}_k^{(i)}\big)^\top \boldsymbol{A}_k^{(i)}(\boldsymbol{x}_k - \boldsymbol{x}_{k-1,i}) - \frac{\eta_k}{2n}\|\boldsymbol{y}_k - \boldsymbol{v}\|_{\boldsymbol{\Lambda}^{-1}}^2 - \frac{\eta_k}{2n}\|\boldsymbol{y}_k - \boldsymbol{y}_*\|_{\boldsymbol{\Lambda}^{-1}}^2. \quad (36)$$

Using the bound in Lemma 12 and applying Eq. (29) into Eq. (36), we obtain

$$\mathcal{E}_k \leq \Big(\frac{\eta_k^3 n \hat{L}_0 \tilde{L}_0}{b^2} - \frac{\eta_k}{2n}\Big)\|\boldsymbol{y}_k - \boldsymbol{y}_*\|_{\boldsymbol{\Lambda}^{-1}}^2 + \min\Big\{\frac{\eta_k^3 n}{b^2}\hat{L}_0 \tilde{L}_0 \|\boldsymbol{y}_*\|_{\boldsymbol{\Lambda}^{-1}}^2, \frac{\eta_k^3 (n-b)^2}{b^2}\tilde{L}_0 \sigma_*^2\Big\}.$$

If $\eta_k \leq \frac{b}{n\sqrt{2\hat{L}_0 \tilde{L}_0}}$, we have $\frac{\eta_k^3 n \hat{L}_0 \tilde{L}_0}{b^2} - \frac{\eta_k}{2n} \leq 0$, thus

$$\mathcal{E}_k \leq \min\Big\{\frac{\eta_k^3 n}{b^2}\hat{L}_0 \tilde{L}_0 \|\boldsymbol{y}_*\|_{\boldsymbol{\Lambda}^{-1}}^2, \frac{\eta_k^3 (n-b)^2}{b^2}\tilde{L}_0 \sigma_*^2\Big\}.$$

Noticing that $\mathcal{E}_k = \eta_k \text{Gap}^{\boldsymbol{v}}(\boldsymbol{x}_k, \boldsymbol{y}_*) + \frac{b}{2n}\|\boldsymbol{x}_* - \boldsymbol{x}_k\|_2^2 - \frac{b}{2n}\|\boldsymbol{x}_* - \boldsymbol{x}_{k-1}\|_2^2$ and telescoping from $k = 1$ to $K$, we have

$$\sum_{k=1}^{K}\eta_k \text{Gap}^{\boldsymbol{v}}(\boldsymbol{x}_k, \boldsymbol{y}_*)$$

$$\leq \frac{b}{2n}\|\boldsymbol{x}_* - \boldsymbol{x}_0\|_2^2 - \frac{b}{2n}\|\boldsymbol{x}_* - \boldsymbol{x}_K\|_2^2 + \sum_{k=1}^{K}\min\Big\{\frac{\eta_k^3 n}{b^2}\hat{L}_0 \tilde{L}_0 \|\boldsymbol{y}_*\|_{\boldsymbol{\Lambda}^{-1}}^2, \frac{\eta_k^3 (n-b)^2}{b^2}\tilde{L}_0 \sigma_*^2\Big\}.$$

Noticing that $\mathcal{L}(\boldsymbol{x}, \boldsymbol{v})$ is convex w.r.t. $\boldsymbol{x}$, we have $\text{Gap}^{\boldsymbol{v}}(\hat{\boldsymbol{x}}_K, \boldsymbol{y}_*) \leq \sum_{k=1}^{K}\eta_k \text{Gap}^{\boldsymbol{v}}(\boldsymbol{x}_k, \boldsymbol{y}_*)/H_K$, where $\hat{\boldsymbol{x}}_K = \sum_{k=1}^{K}\eta_k \boldsymbol{x}_k/H_K$ and $H_K = \sum_{k=1}^{K}\eta_k$, so we obtain

$$H_K \text{Gap}^{\boldsymbol{v}}(\hat{\boldsymbol{x}}_K, \boldsymbol{y}_*) \leq \frac{b}{2n}\|\boldsymbol{x}_0 - \boldsymbol{x}_*\|_2^2 + \sum_{k=1}^{K}\min\Big\{\frac{\eta_k^3 n}{b^2}\hat{L}_0 \tilde{L}_0 \|\boldsymbol{y}_*\|_{\boldsymbol{\Lambda}^{-1}}^2, \frac{\eta_k^3 (n-b)^2}{b^2}\tilde{L}_0 \sigma_*^2\Big\},$$

Further choosing $\boldsymbol{v} = \boldsymbol{y}_{\hat{\boldsymbol{x}}_K}$, we obtain

$$H_K\big(f(\hat{\boldsymbol{x}}_K) - f(\boldsymbol{x}_*)\big) \leq \frac{b}{2n}\|\boldsymbol{x}_0 - \boldsymbol{x}_*\|_2^2 + \sum_{k=1}^{K}\min\Big\{\frac{\eta_k^3 n}{b^2}\hat{L}_0 \tilde{L}_0 \|\boldsymbol{y}_*\|_{\boldsymbol{\Lambda}^{-1}}^2, \frac{\eta_k^3 (n-b)^2}{b^2}\tilde{L}_0 \sigma_*^2\Big\}.$$

(37)

To analyze the individual gradient oracle complexity, we choose constant stepsizes $\eta \leq \frac{b}{n\sqrt{2\hat{L}_0 \tilde{L}_0}}$ and assume $b < n$ without loss of generality, then Eq. (37) becomes

$$f(\hat{\boldsymbol{x}}_K) - f(\boldsymbol{x}_*) \leq \frac{b}{2n\eta K}\|\boldsymbol{x}_0 - \boldsymbol{x}_*\|_2^2 + \min\Big\{\frac{\eta^2 n}{b^2}\hat{L}_0 \tilde{L}_0 \|\boldsymbol{y}_*\|_{\boldsymbol{\Lambda}^{-1}}^2, \frac{\eta^2 (n-b)^2}{b^2}\tilde{L}_0 \sigma_*^2\Big\}.$$

When $\hat{L}_0\|\boldsymbol{y}_*\|_{\boldsymbol{\Lambda}^{-1}}^2 \leq \frac{(n-b)^2}{n}\sigma_*^2$, we set $\eta = \min\Big\{\frac{b}{n\sqrt{2\hat{L}_0 \tilde{L}_0}}, \Big(\frac{b^3\|\boldsymbol{x}_0 - \boldsymbol{x}_*\|_2^2}{2n^2 \hat{L}_0 \tilde{L}_0 K \|\boldsymbol{y}_*\|_{\boldsymbol{\Lambda}^{-1}}^2}\Big)^{1/3}\Big\}$ and consider the following two possible cases:

- "Small $K$" case: if $\eta = \frac{b}{n\sqrt{2\hat{L}_0 \tilde{L}_0}} \leq \Big(\frac{b^3\|\boldsymbol{x}_0 - \boldsymbol{x}_*\|_2^2}{2n^2 \hat{L}_0 \tilde{L}_0 K \|\boldsymbol{y}_*\|_{\boldsymbol{\Lambda}^{-1}}^2}\Big)^{1/3}$, we have

$$f(\hat{\boldsymbol{x}}_K) - f(\boldsymbol{x}_*) \leq \frac{b}{2n\eta K}\|\boldsymbol{x}_0 - \boldsymbol{x}_*\|_2^2 + \frac{\eta^2 n}{b^2}\hat{L}_0 \tilde{L}_0 \|\boldsymbol{y}_*\|_{\boldsymbol{\Lambda}^{-1}}^2$$

$$\leq \frac{\sqrt{\hat{L}_0 \tilde{L}_0}}{\sqrt{2}K}\|\boldsymbol{x}_0 - \boldsymbol{x}_*\|_2^2 + \frac{\hat{L}_0^{1/3}\tilde{L}_0^{1/3}\|\boldsymbol{y}_*\|_{\boldsymbol{\Lambda}^{-1}}^{2/3}\|\boldsymbol{x}_0 - \boldsymbol{x}_*\|_2^{4/3}}{2^{2/3}n^{1/3}K^{2/3}}.$$

- "Large $K$" case: if $\eta = \left( \frac{b^3 \|\boldsymbol{x}_0 - \boldsymbol{x}_*\|_2^2}{2n^2 \hat{L}_0 \tilde{L}_0 K \|\boldsymbol{y}_*\|_{\boldsymbol{\Lambda}^{-1}}^2} \right)^{1/3} \leq \frac{b}{\sqrt{2\hat{L}_0 \tilde{L}_0}}$, we have

$$f(\hat{\boldsymbol{x}}_K) - f(\boldsymbol{x}_*) \leq \frac{b}{2n\eta K} \|\boldsymbol{x}_0 - \boldsymbol{x}_*\|_2^2 + \frac{\eta^2 n}{b^2} \hat{L}_0 \tilde{L}_0 \|\boldsymbol{y}_*\|_{\boldsymbol{\Lambda}^{-1}}^2$$
$$\leq \frac{2^{1/3} \hat{L}_0^{1/3} \tilde{L}_0^{1/3} \|\boldsymbol{y}_*\|_{\boldsymbol{\Lambda}^{-1}}^{2/3} \|\boldsymbol{x}_0 - \boldsymbol{x}_*\|_2^{4/3}}{n^{1/3} K^{2/3}}.$$

Combining these two cases, we have

$$f(\hat{\boldsymbol{x}}_K) - f(\boldsymbol{x}_*) \leq \frac{\sqrt{\hat{L}_0 \tilde{L}_0}}{\sqrt{2}K} \|\boldsymbol{x}_0 - \boldsymbol{x}_*\|_2^2 + \frac{2^{1/3} \hat{L}_0^{1/3} \tilde{L}_0^{1/3} \|\boldsymbol{y}_*\|_{\boldsymbol{\Lambda}^{-1}}^{2/3} \|\boldsymbol{x}_0 - \boldsymbol{x}_*\|_2^{4/3}}{n^{1/3} K^{2/3}}.$$

Hence, to guarantee $\mathbb{E}[f(\hat{\boldsymbol{x}}_K) - f(\boldsymbol{x}_*)] \leq \epsilon$ for $\epsilon > 0$, the total number of individual gradient evaluations will be

$$nK \geq \max \left\{ \frac{n\sqrt{2\hat{L}_0 \tilde{L}_0} \|\boldsymbol{x}_0 - \boldsymbol{x}_*\|_2^2}{\epsilon}, \frac{4n^{1/2} \hat{L}_0^{1/2} \tilde{L}_0^{1/2} \|\boldsymbol{y}_*\|_{\boldsymbol{\Lambda}^{-1}} \|\boldsymbol{x}_0 - \boldsymbol{x}_*\|_2^2}{\epsilon^{3/2}} \right\}. \tag{38}$$

When $\frac{(n-b)^2}{n} \sigma_*^2 \leq \hat{L}_0 \|\boldsymbol{y}_*\|_{\boldsymbol{\Lambda}^{-1}}^2$, we set $\eta = \min \left\{ \frac{b}{n\sqrt{2\hat{L}_0 \tilde{L}_0}}, \left( \frac{b^3 \|\boldsymbol{x}_0 - \boldsymbol{x}_*\|_2^2}{2n(n-b)^2 \tilde{L}_0 K \sigma_*^2} \right)^{1/3} \right\}$ and consider the two cases as below:

- "Small $K$" case: if $\eta = \frac{b}{n\sqrt{2\hat{L}_0 \tilde{L}_0}} \leq \left( \frac{b^3 \|\boldsymbol{x}_0 - \boldsymbol{x}_*\|_2^2}{2n(n-b)^2 \tilde{L}_0 K \sigma_*^2} \right)^{1/3}$, we have

$$f(\hat{\boldsymbol{x}}_K) - f(\boldsymbol{x}_*) \leq \frac{b}{2n\eta K} \|\boldsymbol{x}_0 - \boldsymbol{x}_*\|_2^2 + \frac{\eta^2 (n-b)^2}{b^2} \tilde{L}_0 \sigma_*^2$$
$$\leq \frac{\sqrt{\hat{L}_0 \tilde{L}_0}}{\sqrt{2}K} \|\boldsymbol{x}_0 - \boldsymbol{x}_*\|_2^2 + \frac{(n-b)^{2/3} \tilde{L}_0^{1/3} \sigma_*^{2/3} \|\boldsymbol{x}_0 - \boldsymbol{x}_*\|_2^{4/3}}{2^{2/3} n^{2/3} K^{2/3}}.$$

- "Large $K$" case: if $\eta = \left( \frac{b^3 \|\boldsymbol{x}_0 - \boldsymbol{x}_*\|_2^2}{2n(n-b)^2 \tilde{L}_0 K \sigma_*^2} \right)^{1/3} \leq \frac{b}{n\sqrt{2\hat{L}_0 \tilde{L}_0}}$, we have

$$f(\hat{\boldsymbol{x}}_K) - f(\boldsymbol{x}_*) \leq \frac{b}{2n\eta K} \|\boldsymbol{x}_0 - \boldsymbol{x}_*\|_2^2 + \frac{\eta^2 (n-b)^2}{b^2} \tilde{L}_0 \sigma_*^2$$
$$\leq \frac{2^{1/3} (n-b)^{2/3} \tilde{L}_0^{1/3} \sigma_*^{2/3} \|\boldsymbol{x}_0 - \boldsymbol{x}_*\|_2^{4/3}}{n^{2/3} K^{2/3}}.$$

Combining these two cases, we obtain

$$f(\hat{\boldsymbol{x}}_K) - f(\boldsymbol{x}_*) \leq \frac{\sqrt{\hat{L}_0 \tilde{L}_0}}{\sqrt{2}K} \|\boldsymbol{x}_0 - \boldsymbol{x}_*\|_2^2 + \frac{2^{1/3} (n-b)^{2/3} \tilde{L}_0^{1/3} \sigma_*^{2/3} \|\boldsymbol{x}_0 - \boldsymbol{x}_*\|_2^{4/3}}{n^{2/3} K^{2/3}}.$$

To guarantee $\mathbb{E}[f(\hat{\boldsymbol{x}}_K) - f(\boldsymbol{x}_*)] \leq \epsilon$ for $\epsilon > 0$, the total number of individual gradient evaluations will be

$$nK \geq \max \left\{ \frac{n\sqrt{2\hat{L}_0 \tilde{L}_0} \|\boldsymbol{x}_0 - \boldsymbol{x}_*\|_2^2}{\epsilon}, \frac{4(n-b) \tilde{L}_0^{1/2} \sigma_* \|\boldsymbol{x}_0 - \boldsymbol{x}_*\|_2^2}{\epsilon^{3/2}} \right\}. \tag{39}$$

Combining Eq. (38) and Eq. (39), we finally have

$$nK \geq \frac{n\sqrt{2\hat{L}_0 \tilde{L}_0} \|\boldsymbol{x}_0 - \boldsymbol{x}_*\|_2^2}{\epsilon}$$
$$+ \min \left\{ \frac{4n^{1/2} \hat{L}_0^{1/2} \tilde{L}_0^{1/2} \|\boldsymbol{y}_*\|_{\boldsymbol{\Lambda}^{-1}} \|\boldsymbol{x}_0 - \boldsymbol{x}_*\|_2^2}{\epsilon^{3/2}}, \frac{4(n-b) \tilde{L}_0^{1/2} \sigma_* \|\boldsymbol{x}_0 - \boldsymbol{x}_*\|_2^2}{\epsilon^{3/2}} \right\},$$

thus finishing the proof. $\qquad\square$

# D Omitted Proofs for Non-Smooth Convex Setting From Section 3

Before we prove Theorem 3 in convex Lipschitz settings, for completeness, we first recall the following standard first-order characterization of convexity.

**Lemma 13.** *Let $f : \mathbb{R}^d \to \mathbb{R}$ be a continuous convex function. Then, for any $\boldsymbol{x}, \boldsymbol{y} \in \mathbb{R}^d$:*

$$f(\boldsymbol{y}) \geq f(\boldsymbol{x}) + \langle \boldsymbol{g_x}, \boldsymbol{y} - \boldsymbol{x} \rangle,$$

*where $\boldsymbol{g_x} \in \partial f(\boldsymbol{x})$, and $\partial f(\boldsymbol{x})$ is the subdifferential of $f$ at $\boldsymbol{x}$.*

The following technical lemma provides a primal-dual gap bound in convex nonsmooth settings.

**Lemma 14.** *For any $k \in [K]$, the iterates $\{\boldsymbol{y}_k^{(i)}\}_{i=1}^m$ and $\{\boldsymbol{x}_{k-1,i}\}_{i=1}^{m+1}$ generated by Algorithm 2 satisfy*

$$
\mathcal{E}_k \leq \frac{\eta_k}{n} \sum_{i=1}^m \left( \boldsymbol{y}_k^{(i)\top} \boldsymbol{A}_k^{(i)} (\boldsymbol{x}_k - \boldsymbol{x}_{k-1,i+1}) + (\boldsymbol{v}_k^{(i)} - \boldsymbol{y}_k^{(i)})^\top \boldsymbol{A}_k^{(i)} (\boldsymbol{x}_k - \boldsymbol{x}_{k-1,i}) \right)
$$
$$
\tag{40}
$$
$$
- \frac{b}{2n} \sum_{i=1}^m \| \boldsymbol{x}_{k-1,i} - \boldsymbol{x}_{k-1,i+1} \|_2^2,
$$

*where $\mathcal{E}_k := \eta_k \big( \mathcal{L}(\boldsymbol{x}_k, \boldsymbol{v}) - \mathcal{L}(\boldsymbol{x}_*, \boldsymbol{y}_*) \big) + \frac{b}{2n} \| \boldsymbol{x}_* - \boldsymbol{x}_k \|_2^2 - \frac{b}{2n} \| \boldsymbol{x}_* - \boldsymbol{x}_{k-1} \|_2^2$.*

*Proof.* By the same argument as in the proof for Lemma 9, we know that $\boldsymbol{a}_{\pi_j^{(k)}}^\top \boldsymbol{x}_{k-1,i} \in \partial \ell_{\pi_j^{(k)}}^* (\boldsymbol{y}_k^j)$ for $b(i-1) + 1 \leq j \leq bi$, then by Lemma 13 we have

$$\ell_{\pi_j^{(k)}}^* (\boldsymbol{v}_k^j) \geq \ell_{\pi_j^{(k)}}^* (\boldsymbol{y}_k^j) + \boldsymbol{a}_{\pi_j^{(k)}}^\top \boldsymbol{x}_{k-1,i} (\boldsymbol{v}_k^j - \boldsymbol{y}_k^j),$$

which leads to

$$
\mathcal{L}(\boldsymbol{x}_k, \boldsymbol{v}) = \frac{1}{n} \sum_{i=1}^m \left( \boldsymbol{v}_k^{(i)\top} \boldsymbol{A}_k^{(i)} \boldsymbol{x}_{k-1,i} - \sum_{j=b(i-1)+1}^{bi} \ell_{\pi_j^{(k)}}^* (\boldsymbol{v}_k^j) \right) + \frac{1}{n} \sum_{i=1}^m \boldsymbol{v}_k^{(i)\top} \boldsymbol{A}_k^{(i)} (\boldsymbol{x}_k - \boldsymbol{x}_{k-1,i})
$$
$$
\leq \frac{1}{n} \sum_{i=1}^m \left( \boldsymbol{y}_k^{(i)\top} \boldsymbol{A}_k^{(i)} \boldsymbol{x}_{k-1,i} - \sum_{j=b(i-1)+1}^{bi} \ell_{\pi_j^{(k)}}^* (\boldsymbol{y}_k^j) \right) + \frac{1}{n} \sum_{i=1}^m \boldsymbol{v}_k^{(i)\top} \boldsymbol{A}_k^{(i)} (\boldsymbol{x}_k - \boldsymbol{x}_{k-1,i}).
$$
$$
\tag{41}
$$

Using the same argument for $\mathcal{L}(\boldsymbol{x}_*, \boldsymbol{y}_*)$ as $\boldsymbol{a}_j^\top \boldsymbol{x}_* \in \partial \ell_j^* (\boldsymbol{y}_*^j)$ for $j \in [n]$, we have

$$
\mathcal{L}(\boldsymbol{x}_*, \boldsymbol{y}_*) = \frac{1}{n} \sum_{i=1}^m \left( \boldsymbol{y}_{*,k}^{(i)\top} \boldsymbol{A}_k^{(i)} \boldsymbol{x}_* - \sum_{j=b(i-1)+1}^{bi} \ell_{\pi_j^{(k)}}^* (\boldsymbol{y}_{*,k}^j) \right)
$$
$$
\geq \frac{1}{n} \sum_{i=1}^m \left( \boldsymbol{y}_k^{(i)\top} \boldsymbol{A}_k^{(i)} \boldsymbol{x}_* - \sum_{j=b(i-1)+1}^{bi} \ell_{\pi_j^{(k)}}^* (\boldsymbol{y}_k^j) \right). \tag{42}
$$

Adding and substracting the term $\frac{b}{2n\eta_k} \sum_{i=1}^m \| \boldsymbol{x}_* - \boldsymbol{x}_{k-1,i} \|_2^2$ on the R.H.S. of Eq. (42), we obtain

$$
\mathcal{L}(\boldsymbol{x}_*, \boldsymbol{y}_*) \geq \frac{1}{n} \sum_{i=1}^m \left( \boldsymbol{y}_k^{(i)\top} \boldsymbol{A}_k^{(i)} \boldsymbol{x}_* + \frac{b}{2\eta_k} \| \boldsymbol{x}_* - \boldsymbol{x}_{k-1,i} \|_2^2 - \sum_{j=b(i-1)+1}^{bi} \ell_{\pi_j^{(k)}}^* (\boldsymbol{y}_k^j) \right)
$$
$$
- \frac{b}{2n\eta_k} \sum_{i=1}^m \| \boldsymbol{x}_* - \boldsymbol{x}_{k-1,i} \|_2^2.
$$

Denote $\phi_k^{(i)}(\boldsymbol{x}) := \boldsymbol{y}_k^{(i)\top} \boldsymbol{A}_k^{(i)} \boldsymbol{x} + \frac{b}{2\eta_k} \| \boldsymbol{x} - \boldsymbol{x}_{k-1,i} \|_2^2$, which is $\frac{b}{\eta_k}$-strongly convex w.r.t. $\boldsymbol{x}$. Noticing that $\boldsymbol{x}_{k-1,i+1} = \arg\min_{\boldsymbol{x} \in \mathbb{R}^d} \left\{ \boldsymbol{y}_k^{(i)\top} \boldsymbol{A}_k^{(i)} \boldsymbol{x} + \frac{b}{2\eta_k} \| \boldsymbol{x} - \boldsymbol{x}_{k-1,i} \|^2 \right\}$ by Line 7 of Alg. 2, we have $\nabla \phi_k^{(i)}(\boldsymbol{x}_{k-1,i+1}) = \boldsymbol{0}$, which leads to

$$\phi_k^{(i)}(\boldsymbol{x}_*) \geq \phi_k^{(i)}(\boldsymbol{x}_{k-1,i+1}) + \frac{b}{2\eta_k} \| \boldsymbol{x}_* - \boldsymbol{x}_{k-1,i+1} \|_2^2.$$

Thus, we obtain

$$\mathcal{L}(\boldsymbol{x}_*, \boldsymbol{y}_*) \geq \frac{1}{n} \sum_{i=1}^{m} \left( \boldsymbol{y}_k^{(i)\top} \boldsymbol{A}_k^{(i)} \boldsymbol{x}_{k-1,i+1} + \frac{b}{2\eta_k} \|\boldsymbol{x}_{k-1,i+1} - \boldsymbol{x}_{k-1,i}\|_2^2 - \sum_{j=b(i-1)+1}^{bi} \ell_{\pi_j^{(k)}}^* (\boldsymbol{y}_k^j) \right)$$

$$+ \frac{b}{2n\eta_k} \sum_{i=1}^{m} \left( \|\boldsymbol{x}_* - \boldsymbol{x}_{k-1,i+1}\|_2^2 - \|\boldsymbol{x}_* - \boldsymbol{x}_{k-1,i}\|_2^2 \right)$$

$$\stackrel{(i)}{=} \frac{1}{n} \sum_{i=1}^{m} \left( \boldsymbol{y}_k^{(i)\top} \boldsymbol{A}_k^{(i)} \boldsymbol{x}_{k-1,i+1} + \frac{b}{2\eta_k} \|\boldsymbol{x}_{k-1,i+1} - \boldsymbol{x}_{k-1,i}\|_2^2 - \sum_{j=b(i-1)+1}^{bi} \ell_{\pi_j^{(k)}}^* (\boldsymbol{y}_k^j) \right)$$

$$+ \frac{b}{2n\eta_k} \|\boldsymbol{x}_k - \boldsymbol{x}_*\|_2^2 - \frac{b}{2n\eta_k} \|\boldsymbol{x}_{k-1} - \boldsymbol{x}_*\|_2^2, \tag{43}$$

where $(i)$ is by telescoping $\sum_{i=1}^{m} \left( \|\boldsymbol{x}_* - \boldsymbol{x}_{k-1,i+1}\|_2^2 - \|\boldsymbol{x}_* - \boldsymbol{x}_{k-1,i}\|_2^2 \right)$ and using $\boldsymbol{x}_k = \boldsymbol{x}_{k-1,m+1}$ and $\boldsymbol{x}_{k-1} = \boldsymbol{x}_{k-1,1}$, which both hold by definition.

Combining the bounds from Eq. (41) and Eq. (43), and denoting

$$\mathcal{E}_k := \eta_k \big( \mathcal{L}(\boldsymbol{x}_k, \boldsymbol{v}) - \mathcal{L}(\boldsymbol{x}_*, \boldsymbol{y}_*) \big) + \frac{b}{2n} \|\boldsymbol{x}_* - \boldsymbol{x}_k\|_2^2 - \frac{b}{2n} \|\boldsymbol{x}_* - \boldsymbol{x}_{k-1}\|_2^2,$$

we finally obtain

$$\mathcal{E}_k \leq \frac{\eta_k}{n} \sum_{i=1}^{m} \boldsymbol{y}_k^{(i)\top} \boldsymbol{A}_k^{(i)} (\boldsymbol{x}_{k-1,i} - \boldsymbol{x}_{k-1,i+1}) + \frac{\eta_k}{n} \sum_{i=1}^{m} \boldsymbol{v}_k^{(i)\top} \boldsymbol{A}_k^{(i)} (\boldsymbol{x}_k - \boldsymbol{x}_{k-1,i})$$

$$- \frac{b}{2n} \sum_{i=1}^{m} \|\boldsymbol{x}_{k-1,i} - \boldsymbol{x}_{k-1,i+1}\|_2^2$$

$$= \frac{\eta_k}{n} \sum_{i=1}^{m} \boldsymbol{y}_k^{(i)\top} \boldsymbol{A}_k^{(i)} (\boldsymbol{x}_k - \boldsymbol{x}_{k-1,i+1}) + \frac{\eta_k}{n} \sum_{i=1}^{m} (\boldsymbol{v}_k^{(i)} - \boldsymbol{y}_k^{(i)})^\top \boldsymbol{A}_k^{(i)} (\boldsymbol{x}_k - \boldsymbol{x}_{k-1,i})$$

$$- \frac{b}{2n} \sum_{i=1}^{m} \|\boldsymbol{x}_{k-1,i} - \boldsymbol{x}_{k-1,i+1}\|_2^2,$$

thus completing the proof. $\qquad \square$

Note that we can still use Lemma 10 to bound the first inner product term in Eq. (40), as we are studying the same algorithm. The following lemma provides a bound on the second inner product term $\mathcal{T}_2 := \frac{\eta_k}{n} \sum_{i=1}^{m} \left( \boldsymbol{v}_k^{(i)} - \boldsymbol{y}_k^{(i)} \right)^\top \boldsymbol{A}_k^{(i)} (\boldsymbol{x}_k - \boldsymbol{x}_{k-1,i})$ in Eq. (40).

**Lemma 15.** *Under Assumption 5, for any $k \in [K]$, the iterates $\{\boldsymbol{y}_k^{(i)}\}_{i=1}^{m}$ and $\{\boldsymbol{x}_{k-1,i}\}_{i=1}^{m+1}$ generated by Algorithm 2 satisfy*

$$\mathcal{T}_2 \leq \frac{\eta_k^2 \sqrt{\hat{G}_{\pi^{(k)}} \tilde{G}_{\pi^{(k)}}}}{b} \|\boldsymbol{y}_k\|_{\boldsymbol{\Gamma}_k^{-1}}^2 + \frac{\eta_k^2 \sqrt{\hat{G}_{\pi^{(k)}} \tilde{G}_{\pi^{(k)}}}}{4b} \|\boldsymbol{v}_k - \boldsymbol{y}_k\|_{\boldsymbol{\Gamma}_k^{-1}}^2. \tag{44}$$

*Proof.* Proceeding as in Lemma 11, we have

$$\mathcal{T}_2 := \frac{\eta_k}{n} \sum_{i=1}^{m} \left( \boldsymbol{v}_k^{(i)} - \boldsymbol{y}_k^{(i)} \right)^\top \boldsymbol{A}_k^{(i)} (\boldsymbol{x}_k - \boldsymbol{x}_{k-1,i}) = -\frac{\eta_k^2}{bn} \sum_{i=1}^{m} \left\langle \boldsymbol{A}_k^\top \boldsymbol{I}_{b(i-1)\uparrow} \boldsymbol{y}_k, \boldsymbol{A}_k^\top \boldsymbol{I}_{(i)} (\boldsymbol{v}_k - \boldsymbol{y}_k) \right\rangle.$$

Using Young's inequality for some $\alpha > 0$ and proceeding as in Eq. (24), we obtain

$$\mathcal{T}_2 \leq \frac{\eta_k^2 \alpha}{2bn} \sum_{i=1}^{m} \|\boldsymbol{A}_k^\top \boldsymbol{I}_{b(i-1)\uparrow} \boldsymbol{y}_k\|_2^2 + \frac{\eta_k^2}{2bn\alpha} \sum_{i=1}^{m} \|\boldsymbol{A}_k^\top \boldsymbol{I}_{(i)} (\boldsymbol{v}_k - \boldsymbol{y}_k)\|_2^2$$

$$\leq \frac{\eta_k^2 n\alpha}{2b^2} \hat{G}_{\pi^{(k)}} \|\boldsymbol{y}_k\|_{\boldsymbol{\Gamma}_k^{-1}}^2 + \frac{\eta_k^2}{2n\alpha} \tilde{G}_{\pi^{(k)}} \|\boldsymbol{v}_k - \boldsymbol{y}_k\|_{\boldsymbol{\Gamma}_k^{-1}}^2,$$

where we use our definitions that $\hat{G}_{\pi^{(k)}} := \frac{1}{mn}\big\|\mathbf{\Gamma}_k^{1/2}\big(\sum_{j=1}^m \mathbf{I}_{b(j-1)\uparrow}\mathbf{A}_k\mathbf{A}_k^\top \mathbf{I}_{b(j-1)\uparrow}\big)\mathbf{\Gamma}_k^{1/2}\big\|_2$ and $\tilde{G}_{\pi^{(k)}} := \frac{1}{b}\big\|\mathbf{\Gamma}_k^{1/2}\big(\sum_{j=1}^m \mathbf{I}_{(j)}\mathbf{A}_k\mathbf{A}_k^\top \mathbf{I}_{(j)}\big)\mathbf{\Gamma}_k^{1/2}\big\|_2$. It remains to choose $\alpha = \frac{2b}{n}\sqrt{\frac{\tilde{G}_k}{\hat{G}_k}}$ to finish the proof. $\qquad\square$

We are now ready to prove Theorem 3 for the convergence of shuffled SGD in the convex nonsmooth Lipschitz settings.

**Theorem 3.** *Under Assumption 5, if $H_K = \sum_{k=1}^K \eta_k$ and $\bar{G} = \mathbb{E}_\pi[\sqrt{\hat{G}_\pi \tilde{G}_\pi}]$, the output $\hat{\boldsymbol{x}}_K$ of Alg. 1 with possible uniformly random shuffling satisfies*

$$\mathbb{E}[H_K(f(\hat{\boldsymbol{x}}_K) - f(\boldsymbol{x}_*))] \le \frac{1}{2n}\|\boldsymbol{x}_0 - \boldsymbol{x}_*\|_2^2 + \sum_{k=1}^K 2\eta_k^2 n\bar{G},$$

*As a result, for any $\epsilon > 0$, there exists a step size $\eta_k = \eta$ such that $\mathbb{E}[f(\hat{\boldsymbol{x}}_K) - f(\boldsymbol{x}_*)] \le \epsilon$ after $\mathcal{O}\big(\frac{n\bar{G}\|\boldsymbol{x}_0 - \boldsymbol{x}_*\|_2^2}{\epsilon^2}\big)$ individual gradient queries.*

*Proof.* To simplify the presentation of our analysis, we first assume $\|\boldsymbol{v}\|_{\mathbf{\Gamma}^{-1}}^2 \le n$, which will be later verified by our choice of $\boldsymbol{v} = \boldsymbol{y}_{\hat{\boldsymbol{x}}_K}$ and Assumption 5.

Combining the bounds in Lemma 10 and 15 and plugging them into Eq. (40), we have

$$
\begin{aligned}
\mathcal{E}_k &\le \frac{\eta_k^2\sqrt{\hat{G}_{\pi^{(k)}}\tilde{G}_{\pi^{(k)}}}}{b}\|\boldsymbol{y}_k\|_{\mathbf{\Gamma}_k^{-1}}^2 + \frac{\eta_k^2\sqrt{\hat{G}_{\pi^{(k)}}\tilde{G}_{\pi^{(k)}}}}{4b}\|\boldsymbol{v}_k - \boldsymbol{y}_k\|_{\mathbf{\Gamma}_k^{-1}}^2 \\
&\overset{(i)}{\le} \frac{\eta_k^2\sqrt{\hat{G}_{\pi^{(k)}}\tilde{G}_{\pi^{(k)}}}}{b}\|\boldsymbol{y}_k\|_{\mathbf{\Gamma}_k^{-1}}^2 + \frac{\eta_k^2\sqrt{\hat{G}_{\pi^{(k)}}\tilde{G}_{\pi^{(k)}}}}{2b}(\|\boldsymbol{v}\|_{\mathbf{\Gamma}^{-1}}^2 + \|\boldsymbol{y}_k\|_{\mathbf{\Gamma}_k^{-1}}^2) \\
&\overset{(ii)}{\le} \frac{2\eta_k^2 n\sqrt{\hat{G}_{\pi^{(k)}}\tilde{G}_{\pi^{(k)}}}}{b},
\end{aligned}
\tag{45}
$$

where we use Young's inequality for $\|\boldsymbol{v}_k - \boldsymbol{y}_k\|_{\mathbf{\Gamma}_k^{-1}}^2$ and $\|\boldsymbol{v}_k\|_{\mathbf{\Gamma}_k^{-1}}^2 = \|\boldsymbol{v}\|_{\mathbf{\Gamma}^{-1}}^2$ as $\boldsymbol{v}$ is a fixed vector for $(i)$, and $(ii)$ is due to $\|\boldsymbol{y}_k\|_{\mathbf{\Gamma}_k^{-1}}^2 \le n$ by Assumption 5 and assuming that $\|\boldsymbol{v}\|_{\mathbf{\Gamma}^{-1}}^2 \le n$. Proceeding as the proof for Theorem 2, we first assume the RR scheme and take conditional expectation w.r.t. the randomness up to but not including $k$-th epoch, then we obtain

$$\mathbb{E}_k[\mathcal{E}_k] \le \frac{2\eta_k^2 n\mathbb{E}_k\big[\sqrt{\hat{G}_{\pi^{(k)}}\tilde{G}_{\pi^{(k)}}}\big]}{b}.$$

Since the randomness only comes from the random permutation $\pi^{(k)}$, we have

$$\mathbb{E}_k[\mathcal{E}_k] \le \frac{2\eta_k^2 n\mathbb{E}_\pi[\sqrt{\hat{G}_\pi \tilde{G}_\pi}]}{b}.$$

For notational convenience, we denote $\bar{G} = \mathbb{E}_\pi[\sqrt{\hat{G}_\pi \tilde{G}_\pi}]$, and further take expectation w.r.t. all the randomness on both sides and use the law of total expectation to obtain

$$\mathbb{E}[\mathcal{E}_k] \le \frac{2\eta_k^2 n\bar{G}}{b}.\tag{46}$$

For the SO scheme, there is one random permutation $\pi$ generated at the very beginning such that $\pi^{(k)} = \pi$ for all $k \in [K]$. So we can directly take expectation w.r.t. all the randomness on both sides of Eq. (45), with the randomness only from $\pi$, which leads to the same bound as Eq. (46) with $\mathbb{E}\big[\sqrt{\hat{G}_{\pi^{(k)}}\tilde{G}_{\pi^{(k)}}}\big] = \mathbb{E}_\pi\big[\sqrt{\hat{G}_\pi \tilde{G}_\pi}\big]$. Note that for incremental gradient (IG) descent, we can let $\bar{G} = \sqrt{\hat{G}_0\tilde{G}_0}$ without randomness involved, where $\hat{G}_0 = \hat{G}_{\pi^{(0)}}$ and $\tilde{G}_0 = \tilde{G}_{\pi^{(0)}}$ w.r.t. the initial, fixed permutation $\pi^{(0)}$ of the data matrix $\boldsymbol{A}$.

Noticing that $\mathcal{E}_k = \eta_k \text{Gap}^{\boldsymbol{v}}(\boldsymbol{x}_k, \boldsymbol{y}_*) + \frac{b}{2n}\|\boldsymbol{x}_* - \boldsymbol{x}_k\|_2^2 - \frac{b}{2n}\|\boldsymbol{x}_* - \boldsymbol{x}_{k-1}\|_2^2$ and telescoping from $k = 1$ to $K$, we have

$$\mathbb{E}\Big[\sum_{k=1}^{K} \eta_k \text{Gap}^{\boldsymbol{v}}(\boldsymbol{x}_k, \boldsymbol{y}_*)\Big] \leq \frac{b}{2n}\|\boldsymbol{x}_* - \boldsymbol{x}_0\|_2^2 - \frac{b}{2n}\mathbb{E}[\|\boldsymbol{x}_* - \boldsymbol{x}_K\|_2^2] + \sum_{k=1}^{K} \frac{2\eta_k^2 n\bar{G}}{b}.$$

Noticing that $\mathcal{L}(\boldsymbol{x}, \boldsymbol{v})$ is convex wrt $\boldsymbol{x}$, we have $\text{Gap}^{\boldsymbol{v}}(\hat{\boldsymbol{x}}_K, \boldsymbol{y}_*) \leq \sum_{k=1}^{K} \eta_k \text{Gap}^{\boldsymbol{v}}(\boldsymbol{x}_k, \boldsymbol{y}_*)/H_K$, where $\hat{\boldsymbol{x}}_K = \sum_{k=1}^{K} \eta_k \boldsymbol{x}_k / H_K$ and $H_K = \sum_{k=1}^{K} \eta_k$, so we obtain

$$\mathbb{E}\Big[H_K \text{Gap}^{\boldsymbol{v}}(\hat{\boldsymbol{x}}_K, \boldsymbol{y}_*)\Big] \leq \frac{b}{2n}\|\boldsymbol{x}_0 - \boldsymbol{x}_*\|_2^2 + \sum_{k=1}^{K} \frac{2\eta_k^2 n\bar{G}}{b}.$$

Further choosing $\boldsymbol{v} = \boldsymbol{y}_{\hat{\boldsymbol{x}}_K}$, which also verifies $\|\boldsymbol{v}\|_{\boldsymbol{\Gamma}^{-1}}^2 = \|\boldsymbol{y}_{\hat{\boldsymbol{x}}_K}\|_{\boldsymbol{\Gamma}^{-1}}^2 \leq n$ by Assumption 5, we obtain

$$\mathbb{E}[H_K(f(\hat{\boldsymbol{x}}_K) - f(\boldsymbol{x}_*))] \leq \frac{b}{2n}\|\boldsymbol{x}_0 - \boldsymbol{x}_*\|_2^2 + \sum_{k=1}^{K} \frac{2\eta_k^2 n\bar{G}}{b}.$$

To analyze the individual gradient oracle complexity, we choose constant stepsize $\eta$. Then, the above bound becomes

$$\mathbb{E}[f(\hat{\boldsymbol{x}}_K) - f(\boldsymbol{x}_*)] \leq \frac{b}{2n\eta K}\|\boldsymbol{x}_0 - \boldsymbol{x}_*\|_2^2 + \frac{2n\eta\bar{G}}{b}.$$

Choosing $\eta = \frac{b\|\boldsymbol{x}_0 - \boldsymbol{x}_*\|_2}{2n\sqrt{K\bar{G}}}$, we have

$$\mathbb{E}[f(\hat{\boldsymbol{x}}_K) - f(\boldsymbol{x}_*)] \leq \frac{2\sqrt{\bar{G}}\|\boldsymbol{x}_0 - \boldsymbol{x}_*\|_2}{\sqrt{K}}.$$

Hence, given $\epsilon > 0$, to ensure $\mathbb{E}[f(\hat{\boldsymbol{x}}_K) - f(\boldsymbol{x}_*)] \leq \epsilon$, the total number of individual gradient evaluations will be

$$nK \geq \frac{4n\bar{G}\|\boldsymbol{x}_0 - \boldsymbol{x}_*\|_2^2}{\epsilon^2},$$

thus completing the proof. $\qquad\square$

We now briefly discuss this result. The total number of individual gradient queries is $\mathcal{O}\big(\frac{n\bar{G}\|\boldsymbol{x}_0 - \boldsymbol{x}_*\|_2^2}{\epsilon^2}\big)$, which appears independent of the batch size, but this is actually not the case, as the parameter $\bar{G} = \mathbb{E}_\pi[\sqrt{\hat{G}_\pi \tilde{G}_\pi}]$ depends on the block partitioning, due to Eq. (4). When $b = n$, as a sanity check, we recover the standard guarantee of (full) subgradient descent, which is expected, as in this case shuffled SGD reduces to subgradient descent. When $b = 1$, however, the bound is worse than the corresponding bound for standard SGD, by a factor $\mathcal{O}(n\bar{G}/G^2)$. By a similar sequence of inequalities as in Eq. (4), this factor is never worse than $n$, but it is typically much smaller, taking values as small as 1. We note that it is not known whether a better bound is possible for shuffled SGD in this setting, as the only seemingly tighter upper bound from [42] applies only for constant $K$, when $n = \Omega(\frac{1}{\epsilon^2})$, and under an additional boundedness assumption for the algorithm iterates.

## E    Experiment Details

We implement the computation of $\hat{L}$ and $L_{\max}$ in Julia, a high-performance scientific computation programming language, and compute matrix operator norms using the default settings in the Julia `Arpack` Package. However, limited by computational memory and time constraint, our selection of datasets is focused on moderately large-scale datasets of $n$ in the order of $O(10^5)$. We also include comparisons of small datasets such as `a1a` and `sonar`.

### E.1    Evaluations of $L_{\max}/\tilde{L}_\pi$ on Synthetic Gaussian Datasets

We first study the gap between $\tilde{L}_\pi$ and $L_{\max}$ for different batch sizes $b$, as shown in Figure 2. As in Section 4.1, we focus on their dependence on the data matrix, and assume that the loss functions $\ell_i$ all have the same smoothness constant. In this case, the ratio $L_{\max}/\tilde{L}_\pi$ that characterizes the gap between

$\tilde{L}_\pi$ and $L_{\max}$ will become $L_{\max}/\tilde{L}_\pi = (\max_{1 \le i \le n}\{\|\boldsymbol{a}_i\|_2^2\})/(\frac{1}{b}\|\sum_{j=1}^{m} \boldsymbol{I}_{(j)} \boldsymbol{A}_\pi \boldsymbol{A}_\pi^\top \boldsymbol{I}_{(j)}\|_2)$. In particular, we run experiments on standard Gaussian data of size $(n, d)$. We fix the dimension $d = 500$, and vary the number of samples with $n = 100, 500, 1000, 2000$. In Figure 2, we plot the ratio $L_{\max}/\tilde{L}_\pi$ versus the batch size $b$ for 100 different random permutations $\pi$, where the dotted lines represent the mean values and the filled regions indicate the standard deviation of permutations. We observe that the ratio $L_{\max}/\tilde{L}_\pi$ is concentrated around its empirical mean and exhibits $b^\alpha$ ($\alpha \in [0.74, 0.87]$) growth as the batch size $b$ increases. In particular, if we choose $b = \sqrt{n}$, the ratio can be $\mathcal{O}(n^{0.4})$.

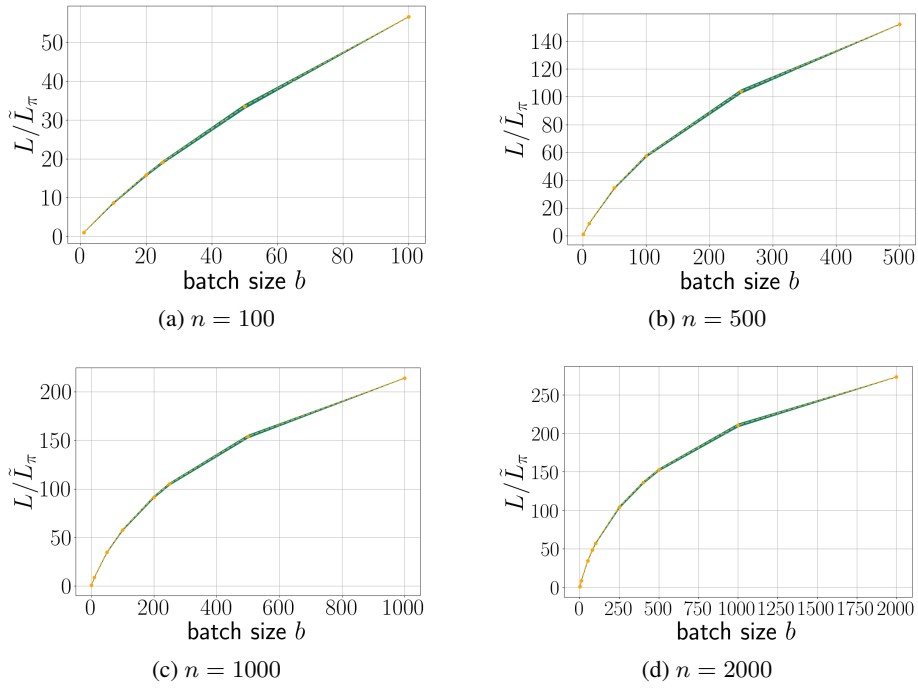

Figure 2: Illustrations of $L_{\max}/\tilde{L}_\pi$ for different batch size $b$ on synthetic Gaussian data of size $(n, d)$.

## E.2 Distributions of $L_{\max}/\hat{L}_\pi$

In this subsection, we include histograms in Figure 3 to illustrate the spread of $L_{\max}/\hat{L}_\pi$ with respect to random permutations, for completeness. We observe that in all the examples $L_{\max}/\hat{L}_\pi$ is concentrated around its empirical mean. The following plots are normalized, with y-axis representing the empirical probability density. The x-axis represents $L_{\max}/\hat{L}_\pi$.

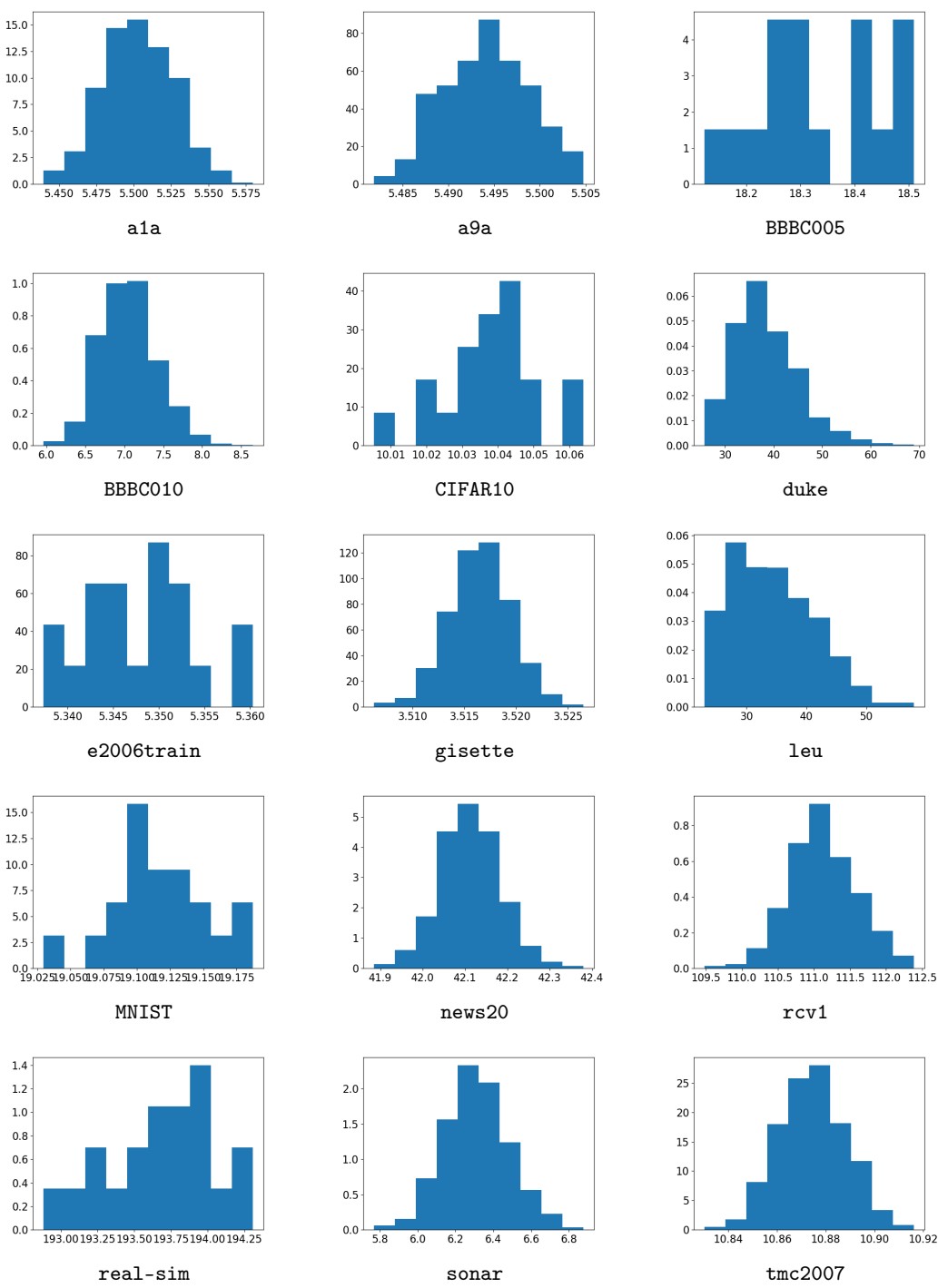

Figure 3: Visualization of the empirical distributions of $L/\hat{L}$ for 15 large-scale datasets.

