# OpenReview forum: "Tighter Convergence Bounds for Shuffled SGD via Primal-Dual Perspective"
_NeurIPS.cc/2024/Conference — NeurIPS 2024 poster_

### Official Review · Reviewer_S4vW · 2024-07-03

**Soundness:** 3
**Presentation:** 3
**Contribution:** 3
**Rating:** 7
**Confidence:** 4

**Summary:**

This paper aims at improving existing bounds on random reshuffling. While lower bounds are tight in the worst case, refined smoothness definitions allow to take a larger step-size in favorable cases, which in turns allows faster convergence. These smoothness definitions involve a supremum over permutation-dependent partial averaging of individual smoothness constants.

The dependence on these refined smoothness constants is obtained through a primal-dual interpretation of SGD, though the convergence analysis does not consist in applying existing primal-dual analyses to this formulation. As a side note, results include mini-batching, which is nice to have.

For generalized linear models, individual covariances can be averaged instead of individual smoothness constants, thus allowing to gain even more especially in high-dimensions. A non-smooth extension is given, performing similar averagings over individual Lipschitz constants.

Typos:
Line 27: why does SGD rely on a standard basis vector?? This looks like a mix between SGD and coordinate descent.
L45: we choose take
Algo 1: shouldn't this be a $\nabla f_i$ on line 7?

**Strengths:**

- Nice use of the finite-sum assuption, which allows to use a primal-dual reformulation.
- Improves over existing bounds for shuffled SGD, mimicking cyclic coordinate descent results.

**Weaknesses:**

- $\tilde{L}$, the $L_{max}$-like constant, is the one that appears in Theorems 1 and 2. In particular, the more "average smoothness" constant (from which most of the improvement seems to come) only affect higher-order terms by using larger step-sizes to be used, but does not change anything for small step-sizes.
- While results are strong for generalized linear models (taking into account the actual spectrum of each Hessian), general results are quite looser (though they still improve on existing ones).

**Questions:**

1) In the end, everything seems to depend on primal quantities. What makes an actual difference in the analysis that could not have been done in the primal (or why was it more intuitive to do it in the primal)? Why can't the full analyses be done by replacing the $y^i_k with $\nabla f_i(x_{k,i})$?

In particular, what technical challenges do you expect moving to the non-convex setting? In the end all quantities can, but the primal-dual interpretation of SGD does not hold anymore since biconjugacy is used at some point if I'm not mistaken).

2) What are the connections with existing (primal-dual) CD analyses? I see that the primal-dual reformulation allows to study a setting that is close (although different since CD would not get any residual variance). It is said in Appendix A that results are technically disjoints, so then why is this inspiration helpful still?

3) It seems that the discussion from Section 4 only applies to the generalized linear models case, because it does not seem possible to benefit from structure in the general case. Do you have any intuition for why such assumptions cannot be made using Hessians, and thus benefit from similar improvements? Additionally, how would Table 2 look with these bounds instead of the generalized linear models ones?

4) Do you actually expect to depend on the smoothness for the average permutation (instead of worst-case)? Is this complicated to analyze directly because of the dependence between iterates within a single epoch?

**Limitations:**

See weaknesses

---

> ### Author Rebuttal · Authors · 2024-08-06
>
> **We thank the reviewer for the positive evaluation of our paper. We hope that the clarifications provided below address the reviewer’s questions. We are happy to answer further questions in the discussion phase.**
>
> ---
>
> ### Summary
> > Typos
>
> Thanks for pointing out. It should be $E_{i_t} [ \nabla f_{i_t} (x_t) / p_{i_t}] = \nabla f (x_t)$ in Line 27. We will correct those typos in our revision.
>
> ---
>
> ### Questions
> > Q1
>
> Correct. In our case, the primal-dual view allowed us to separate the linear transformation ($a_i^\top x$) from nonlinearity (loss $\ell_i$), as is done in splitting primal-dual methods like [1]. The separation of linear vs nonlinear portion was useful for handling cancellations coming from the cyclic updates (see e.g., the proof of Lemma 9) and obtaining the data dependent bounds (see e.g., Lines 692–696 in the proof of Lemma 11). That said, given that in the end the algorithm is equivalent to standard shuffled SGD (i.e., the algorithm is primal-only and the primal-dual perspective is only there for the analysis), we expect that there is a way to write the analysis as primal-only. We did not pursue this direction as we did not think it would improve the readability and interpretations of the proofs.
>
> > Q2
>
> We first note that in nonconvex settings one would need guarantees in terms of stationarity, which is different from the optimality gap in convex settings. Also, all the existing work on shuffled SGD for nonconvex objectives requires additional assumptions about the problem, such as bounded variance at all points [2] or bounded sequence correlation [3].
>
> However, we believe that our techniques can be generalized to at least some (but still fundamental in the context of ML applications) nonconvex objectives, such as those arising from generalized linear models (GLMs). As a specific example, one could consider minimizing the mean squared loss of a GLM (e.g., ReLU as the most basic example): $$\min_x \frac{1}{2 n} \sum_{i=1}^n (\sigma(x^\top a_i) - b_i)^2$$ where $\sigma(t)$ is an activation function; for example $\sigma(t) = \mathrm{ReLU}(t) = \max\\{0, t\\}.$
>
> One can “dualize” the quadratic portion of this objective to write it in a primal-dual form as: $$\min_x \max_y \frac{1}{n} \sum_{i=1}^n (\sigma(x^\top a_i) - b_i)y_i - (1/2)y_i^2.$$ In this case, we get a nonlinear (but structured) coupling of the primal and the dual, which we believe can be handled with a separate analysis. One relevant observation is that in this case $a_i y_i$ (which for convex ERM functions in our paper was the gradient of the component function) now corresponds to the gradient of a *surrogate* function for the nonconvex GLM problem. SGD-style methods applied to the surrogate function with the same gradient field $a_i y_i$ as here have been widely studied in the learning theory literature and used to argue about the learnability of these problems (under distributional assumptions); see e.g., [4] and references therein. We leave the investigation of nonconvex problems based on GLMs as an interesting topic for future study.
>
>
> > Q3
>
> Thanks for raising this point. We draw inspiration from primal-dual methods to separate the linear and nonlinear components of the objectives and follow the methodology of gap construction in primal-dual methods. However, details of the analysis are crucially different, as discussed in Appendix A. First, most CD analyses cannot provide tight bounds for the setting with random permutation of blocks, while the only works handling random permutations are specialized to convex quadratics [5,6]. Further, for those works providing fine-grained CD analyses [7–9] (most CD works only provide analyses with the worst-case dependence on the number of blocks), they either rely on a certain block-wise descent lemma [7] ( there is no smoothness or descent on the dual side for shuffled SGD), or use extrapolation steps which are incompatible with the shuffled SGD algorithm [8, 9].
>
>
> > Q4
>
> To use Hessians, we would need to assume the function is twice (continuously) differentiable, which are the assumptions we are not using right now. It is however a good insight that possibly similar fine-grained bounds can be obtained by working with the Hessian. An alternative approach would be to assume that functions are smooth with respect to Mahalanobis (semi)norms, similar to what was done in [7–9], and then obtain bounds dependent on the matrices defining those norms. This is something that seems doable and we can consider adding if the reviewer thinks it would be useful to include.
>
>
> > Q5
>
> We first note the dependence on the worst-case permutation mainly appears in our final complexity and comes from the need to select a constant, deterministic step size, as discussed in Lines 716-723 in the proof. In principle, one can replace it with a certain high-probability bound on $\hat L_\pi$ and $\tilde L_\pi$. We would like to further point out that for the 15 datasets we looked at, the smoothness parameter over permutations seems to concentrate around the mean, as illustrated in Appendix E. Thus in practice, the difference between these two quantities appears minor.
>
> On the other hand, for the dependence on $\tilde L$ in Lemma 2 and Theorem 2, the reviewer is correct on the technical difficulty. We refer to our proof in Lines 695-705, as an example. From the inequality in Lines 695–696, we need to bound
> $\tilde L_{\pi^{(k)}}\\|A_k^\top I_{(i - 1)\uparrow}y_{\ast, k}\\|^2$ with permutation randomness coupled together. Such a term is nontrivial to bound, so we first relax $\tilde L_{\pi^{(k)}}$ to $\tilde L$ to reduce the randomness on $\\|A_k^\top I_{(i - 1)\uparrow}y_{\ast,k}\\|^2$ only, and then use Lemma 8 to bound that term as in Lines 703–704. However, we conjecture that the dependence on the average permutation could be achieved by a much more complicated analysis combining our fine-grained analysis with the techniques of conditional expectations, which we leave for future research.

---

> > ### Comment · Reviewer_S4vW · 2024-08-12
> >
> > Thank you for carefully answering my questions.
> >
> > I believe the Mahalanobis semi-norm extension (which as you point out allows to bypass the second order differentiability requirement, though might be less tight for non-GLMs) would be nice to have, but I don't consider it compulsory.

---

> > > ### Author Response · Authors · 2024-08-12
> > >
> > > Thank you for acknowledging our response and for the clarification regarding the Mahalanobis semi-norm extension.

---

> ### Author Response · Authors · 2024-08-06
> **References for the rebuttal**
>
> ### References
>
> [1] Chambolle, A. and Pock, T. A first-order primal-dual algorithm for convex problems with applications to imaging. Journal of Mathematical Imaging and Vision, 40(1):120–145, 2011.
>
> [2] Mishchenko, K., Khaled, A., and Richtarik, P. Random reshuffling: Simple analysis with vast improvements. In Proc. NeurIPS’20, 2020.
>
> [3] Koloskova, A., Doikov, N., Stich, S. U., & Jaggi, M. On Convergence of Incremental Gradient for Non-Convex Smooth Functions. In Proc. ICML'24, 2024.
>
> [4] Wang P, Zarifis N, Diakonikolas I, Diakonikolas J. Robustly learning a single neuron via sharpness. In Proc. ICML’23, 2023.
>
> [5] Lee, C.-P. and Wright, S. J. Random permutations fix a worst case for cyclic coordinate descent. IMA Journal of Numerical Analysis, 39(3):1246–1275, 2019.
>
> [6] Wright, S. and Lee, C.-P. Analyzing random permutations for cyclic coordinate descent. Mathematics of Computation, 89(325):2217–2248, 2020.
>
> [7] Xufeng Cai, Chaobing Song, Stephen J Wright, and Jelena Diakonikolas. Cyclic block coordinate descent with variance reduction for composite nonconvex optimization. In Proc. ICML'23, 2023.
>
> [8] Chaobing Song and Jelena Diakonikolas. Cyclic Coordinate Dual Averaging with Extrapolation for Generalized Variational Inequalities. SIAM Journal on Optimization, 2023.
>
> [9] Cheuk Yin Lin, Chaobing Song, and Jelena Diakonikolas. Accelerated cyclic coordinate dual averaging with extrapolation for composite convex optimization. In Proc. ICML'23, 2023.

---

### Official Review · Reviewer_EeyJ · 2024-07-10

**Soundness:** 3
**Presentation:** 3
**Contribution:** 3
**Rating:** 6
**Confidence:** 2

**Summary:**

This paper studied the convergence of random shuffled SGD through the lens of dual coordinate descent. By leveraging the analysis in coordinate descent, the author(s) derived a rate that is $O(\sqrt{n})$ faster than existing rate.

**Strengths:**

Pros:
-  The paper is well written and easy to follow, contributions are clearly stated (Table 1).
-  The rate derived by the author(s) is better than previous state-of-the-art.

**Weaknesses:**

- The primal-dual relationship between SGD and coordinate descent has a quite long history, e.g. [1]. The author(s) should mention these works in the related work section or section 2.

[1] Stochastic Dual Coordinate Ascent Methods for Regularized Loss Minimization, Shai Shalev-Shwartz and Tong Zhang, 2013.

**Questions:**

The convergence analysis of coordinate descent usually have to assume the objective is smooth, which means the primal objective is strongly convex. But the strong convexity assumption was not used in this paper, I wonder how does the author(s) circumvent this assumption?

**Limitations:**

Limitations are not well-addressed. I do not see any potential negative impact of this paper.

---

> ### Author Rebuttal · Authors · 2024-08-06
>
> **We thank the reviewer for their valuable feedback. We hope that the answers provided below address the reviewer’s concerns and that the reviewer would consider reevaluating our work. We appreciate the opportunity to answer further questions in the discussion phase.**
>
> ---
>
> ### Weaknesses
> > The primal-dual relationship between SGD and coordinate descent has a quite long history, e.g. [1]. The author(s) should mention these works in the related work section or section 2.
>
> Thank you for pointing out this line of work. We will cite this paper and discuss related work in our revision. In particular, [1] provides theoretical results only for SDCA, which chooses the dual coordinate to optimize *uniformly at random*. The cyclic variant SDCA-Perm (related to shuffled SGD) that samples the dual coordinate without replacement is only presented as an empirical example and studied through numerical experiments.
>
> ---
>
> ### Questions
> > The convergence analysis of coordinate descent usually have to assume the objective is smooth, which means the primal objective is strongly convex. But the strong convexity assumption was not used in this paper, I wonder how does the author(s) circumvent this assumption?
>
> The reviewer is correct, and that is exactly why the existing analyses for coordinate methods cannot be applied to shuffled SGD methods. Note that there is no smoothness or descent on the dual side in our setting, and we are also taking the "best response" steps on the dual, which is strongly convex (due to primal smoothness). It is also important to note here that cyclic methods would be making specific gradient or proximal steps and use the properties of such steps in the analysis, while here we can only allow the dual update to be “best response” so that we maintain the equivalence with the standard (primal-only) shuffled SGD. To handle these issues, we follow a primal-dual approach, which is different from traditional analysis of coordinate methods.
>
> Strong convexity is mainly used to bound the gap by introducing negative terms $-\\|y_k - y_*\\|^2_{\Lambda^{-1}}$ on the dual side; see Lines 669-670 in the proof as an example. As a side note, our analysis can also deduce convergence results on the dual variables $\\|y_k - y_*\\|^2_{\Lambda^{-1}}$.
>
> The major part in which we build connections with coordinate methods is on how to characterize the difference between the intermediate iterate $x_{k- 1, i}$ and the iterate $x_k$ after one full cycle, as mentioned in Lines 73-79. To improve over previous worst-case analysis and obtain a tighter fine-grained bound, one needs to avoid using global smoothness with a triangle inequality, which is a prevalent approach in existing analyses of shuffled SGD [2–4]. Instead, we derive the fine-grained bounds on the partial sum of intermediate dual variables $y_k^i$ by tracking the progress of the cyclic update on the dual side, in the aggregate; see e.g., the proof of Lemma 11 to bound $\mathcal{T}_2$. Our technique mirrors the very recent advance on the fine-grained bounds for coordinate methods [5–7]. However, our setting and algorithm are technically disjoint with these works, as discussed in Lines 455-462 in Appendix A.
>
> ---
>
> ### References
>
> [1] Shalev-Shwartz, S., & Zhang, T. Stochastic dual coordinate ascent methods for regularized loss minimization. Journal of Machine Learning Research, 14(1), 2013.
>
> [2] Jaeyoung Cha, Jaewook Lee, and Chulhee Yun. Tighter lower bounds for shuffling SGD: Random permutations and beyond. In Proc. ICML'23, 2023.
>
> [3] Konstantin Mishchenko, Ahmed Khaled, and Peter Richtárik. Random reshuffling: Simple analysis with vast improvements. In Proc. NeurIPS’20, 2020.
>
> [4] Lam M Nguyen, Quoc Tran-Dinh, Dzung T Phan, Phuong Ha Nguyen, and Marten Van Dijk. A unified convergence analysis for shuffling-type gradient methods. The Journal of Machine Learning Research, 22(1):9397–9440, 2021.
>
> [5] Xufeng Cai, Chaobing Song, Stephen J Wright, and Jelena Diakonikolas. Cyclic block coordinate descent with variance reduction for composite nonconvex optimization. In Proc. ICML'23, 2023.
>
> [6] Chaobing Song and Jelena Diakonikolas. Cyclic Coordinate Dual Averaging with Extrapolation for Generalized Variational Inequalities. SIAM Journal on Optimization, 2023.
>
> [7] Cheuk Yin Lin, Chaobing Song, and Jelena Diakonikolas. Accelerated cyclic coordinate dual averaging with extrapolation for composite convex optimization. In Proc. ICML'23, 2023.

---

> > ### Comment · Reviewer_EeyJ · 2024-08-12
> > **Thanks for your response**
> >
> > I would like to thank the author(s) for the detailed response. My questions are addressed, I have raised my score slightly.

---

> > > ### Author Response · Authors · 2024-08-12
> > > **Thank you**
> > >
> > > Thank you for carefully considering our response and acknowledging that your questions have been addressed. We appreciate your increased support of our work.

---

### Official Review · Reviewer_Ua2v · 2024-07-13

**Soundness:** 2
**Presentation:** 2
**Contribution:** 2
**Rating:** 5
**Confidence:** 4

**Summary:**

This paper considers the primal-dual aspect of SGD using sampling without replacement (shuffled SGD). The authors present results for shuffled SGD on smooth and non-smooth convex settings with tight convergence bounds for several shuffling schemes (IG, SO, and RR). In some specific settings, the convergence rate can be improved with a factor of $\sqrt{n}$. The authors perform experiments to demonstrate that their bound is tight and better than prior work.

**Strengths:**

The theoretical results seem to be solid, although I did not check the proofs. The authors choose to investigate a different type of smoothness constant compared to the prior work, thus lead to improvement in the bound. In addition, they study the primal-dual aspect for linear predictors and then extend the results to non-smooth setting.

**Weaknesses:**

The improvement in the convergence is not significant as it is only changing the Lipschitz constant.
The authors should be more transparent about the setting where there is $\sqrt{n}$ improvement as it seems like this only applies to linear predictors (the claim in abstract, line 14-17 is not clear). The authors should include that context in your statements. Another weakness is that the bounds in Section 3 depend on the data.

**Questions:**

Why do you call RR and SO uniformly random shuffling?

How the results in section 3 reduces and compares to general convex finite-sum problems when $a_i$ are all-1 vectors i.e. loss function applies on $x$?

I have read the author responses.

---

> ### Author Rebuttal · Authors · 2024-08-06
>
> **We thank the reviewer for their feedback and kindly request them to consider our responses below when evaluating our work. We appreciate the opportunity to further engage in a discussion, as needed.**
>
> ---
>
> ### Weaknesses
> > The improvement in the convergence is not significant as it is only changing the Lipschitz constant.
>
> We respectfully disagree with this point, as obtaining a dependence on a fine-grained smoothness/Lipschitzness constant directly leads to up to $O(\sqrt{n})$ complexity improvement. Dependence on other problem parameters cannot be improved, by the lower bound results in [1].
>
> > The authors should be more transparent about the setting where there is $\sqrt{n}$ improvement as it seems like this only applies to linear predictors (the claim in abstract, line 14-17 is not clear).
>
> For general convex smooth problems, one can also expect $O(\sqrt{n})$ improvement, as our results improve the dependence from the max to average smoothness.
>
> > Another weakness is that the bounds in Section 3 depend on the data.
>
> We note that such data-dependent bounds are generally appreciated in theory, because they automatically predict faster convergence than the worst-case analysis based on component smoothness.
>
> ---
>
> ### Questions
> > Why do you call RR and SO uniformly random shuffling?
>
> It is because the permutation is chosen uniformly at random over the set of possible permutations. This terminology is standard in the shuffled SGD literature [1–3].
>
> > How the results in section 3 reduces and compares to general convex finite-sum problems when $a_i$ are all-$1$ vectors i.e. loss function applies on $x$?
>
> We first note that having all $1$s in $a_i$ does not reduce to the general convex finite sum setting, as in that case each $f_i$ would be univariate, dependent on the sum of the entries of $x$.
>
> For the comparison between the results in two sections, we note that our results in Theorem 2 provide a tighter bound than directly applying Theorem 1 to the generalized linear models. This is because Eq. (3) is a tighter estimate than Eq. (2), where the data matrix $A$ and the smoothness constants from the nonlinear part $\Lambda$ are separated in Eq. (3).
>
> In particular, consider the case where $\ell_i$ are all $1$-smooth for simplicity and $f_i = \ell_i(\langle a_i, x\rangle)$ is $\\|a_i\\|^2$-smooth. For brevity, we omit the permutation notation and let the batch size be $1$, then Eq. (3) and Eq. (2) reduce to the operator norms of two $n \times n$-dimensional matrices $M$ and $N$ normalized by $1/n^2$ respectively, where their entries are given by $M_{ij} = \min\\{i, j\\} a_i^\top a_j$ and $N_{ij} = \min\\{i, j\\} \\|a_i\\| \\|a_j\\|$. Note that the operator norm of $M$ is always smaller than the operator norm of $N$, because the absolute value of each element of $M$ is no larger than the corresponding element of $N$, by the Cauchy-Schwarz inequality. Further, for the case where all the row norms are scaled to $1$ (the average and the maximum component smoothness constants would both be equal to $1$), our results would still be tighter by a factor as large as $\sqrt{n}$. This is implied by our Gaussian example discussed in Lines 251-260 in Section 4, using concentration of measure on the sphere.
>
> For general convex cases, we refer to Appendix B for more discussion.
>
> ---
>
> ### References
>
> [1] Jaeyoung Cha, Jaewook Lee, and Chulhee Yun. Tighter lower bounds for shuffling SGD: Random permutations and beyond. In Proc. ICML'23, 2023.
>
> [2] Konstantin Mishchenko, Ahmed Khaled, and Peter Richtárik. Random reshuffling: Simple analysis with vast improvements. In Proc. NeurIPS’20, 2020.
>
> [3] Lam M Nguyen, Quoc Tran-Dinh, Dzung T Phan, Phuong Ha Nguyen, and Marten Van Dijk. A unified convergence analysis for shuffling-type gradient methods. The Journal of Machine Learning Research, 22(1):9397–9440, 2021.

---

### Official Review · Reviewer_JY5g · 2024-07-13

**Soundness:** 3
**Presentation:** 3
**Contribution:** 3
**Rating:** 5
**Confidence:** 4

**Summary:**

This paper focuses on SGD with shuffling for finite-sum minimization problems. While there exists tight convergence upper and lower bounds for SGD assuming a global smoothness constant $L_{\max}$, the proposed results give a more fine-grained analysis in terms of the component-wise smoothness constants $L_i$ to show possibly faster convergence rates using Fenchel dual arguments. By replacing the $L_{\max}$’s with smaller constants $\tilde{L}, \hat{L}$, the new results can improve the iteration complexity by up to a factor of $L_{\max}/\hat{L} = O(n)$ as demonstrated in the linear predictor example. This work also extends to non-smooth, Lipschitz objectives in the case of linear predictors, and provides empirical evidence supporting the claims.

**Strengths:**

- The idea of the ‘primal-dual’ formulation, which essentially allows one to exploit similar-to-coordinate-descent techniques for better dependencies on the $L_i$’s, is new to me and seems to be a potentially useful technique. I agree that using component-wise structures for improved convergence rates could be one of the many meaningful future directions for the shuffling community.
- The paper presents results that consider general types of shuffling-based algorithms (RR, SO, IG) which I appreciate.

**Weaknesses:**

- Focusing on the results, the only big difference with previous work would be the constant $L_{\max}$ being replaced with $\tilde{L}, \hat{L}$, which means that the improvements are directly linked with how small the ratios $\tilde{L}/L_{\max}, \hat{L}/L_{\max}$ could actually be. While Table 2 suggests that $L_{\max}$ could be quite large (for the linear predictor case), it is quite hard to see how ‘small’ the values $\tilde{L}, \hat{L}$ are in first glance due to the relatively complicated definitions. The paper explains the gap for Gaussian data, but I think this example is a bit too ‘specific’.
- Moreover, based on my understanding, the definitions of $\tilde{L}, \hat{L}$ for the general convex case and the linear predictor case seem to be a bit different. It is still unclear whether these fine-grained bounds could imply a significantly *faster* convergence rate, especially for the general convex case.
- Minor Stuff: I might have missed this, but it seems that the definition of $\boldsymbol{A}_{\pi}$ (which I think is $\\boldsymbol{A}\_{\\pi}  = [a\_{\pi_1} \cdots a\_{\pi_n}]^{\top}$) is missing in Section 3.1. Also, the proofs are all written for batch size $b$, might be confusing to readers: maybe it would have been better if the main body also contained statements including $b$.
- TYPO: Line 9 of Algorithm 1, $m$ → $n$
- TYPO: Line 473 mini-bath → mini-batch

**Questions:**

- Is there a way to intuitively understand when or why $\tilde{L}/L_{\max}, \hat{L}/L_{\max}$ could be ‘much’ smaller than $1$ (with possible $n^{\alpha}$-ish dependencies), both for the  Or are there at least any examples apart from the linear predictor that demonstrate significantly small $\tilde{L}/L_{\max}, \hat{L}/L_{\max}$?
- Based on my knowledge, in coordinate descent algorithms it is common to use *different* step sizes for different coordinates, say $\eta_i = 1/L_i$, to improve dependencies on the $L$’s. Would it be possible to do something similar for this case if we take the primal-dual approach?
- Can you elaborate a bit more on why the inequality $(iii)$ of $(4)$ (in Section 4) is loose by a factor of $n$ in most cases? My understanding of the linear predictor case is that the norm-trace gap corresponds to the largest VS sum of singular values, and hence if the singular values are even, then the gap can be as loose as a factor of $n$, while if we have a single large $\lambda_{\max}$ then the inequality can be tight. I wonder if (i) this interpretation is correct, and (ii) whether something similar could hold for the $\tilde{L}^g, \hat{L}^g$’s in the general convex case.
- Minor question: Is it true Algorithm 1 is completely equivalent to the standard SGD with random reshuffling (without the $\boldsymbol{y}$’s), and the dual variables are just there for illustrative purposes?

**Limitations:**

The authors adequately addressed the limitations, and there seems to be no potential negative societal impact.

---

> ### Author Rebuttal · Authors · 2024-08-06
>
> **We appreciate the questions the reviewer raised. We hope that the answers provided below address the reviewer’s concerns and that the reviewer would consider reevaluating our work. We will fix the typos, update statements in the main body to apply to batch size $b$, and add more discussions in the revision, as suggested. Please let us know if there is any additional information that would be helpful.**
>
> ---
>
> ### Weaknesses
> > W1: While Table 2 suggests that $L_{\mathrm{max}}$ could be quite large (for the linear predictor case), it is quite hard to see how ‘small’ the values $\tilde{L}$, $\hat{L}$ are in first glance due to the relatively complicated definitions. The paper explains the gap for Gaussian data, but I think this example is a bit too ‘specific’.
>
> We share the same view that it may be difficult to directly compare $\tilde{L}$, $\hat{L}$ against $L_{\mathrm{max}}$, which is why we conducted numerical computations on 15 popular machine learning datasets to illustrate the gap between $\hat{L}$ and $L_{\mathrm{max}}$. In some cases, we demonstrate that the gap can be as large as $O(n)$ in the linear predictor settings, suggesting the effectiveness of our fine-grained analysis through a primal-dual perspective.
>
> > W2: Moreover, based on my understanding, the definitions of $\tilde{L}$, $\hat{L}$ for the general convex case and the linear predictor case seem to be a bit different. It is still unclear whether these fine-grained bounds could imply a significantly faster convergence rate, especially for the general convex case.
>
> The reviewer is correct on the difference between the two cases, while such a difference is intuitive and desirable. For the general convex case, one can view our fine-grained bounds as improving from maximum (in previous works [1–3]) to average smoothness, which could also lead to $O(\sqrt{n})$ improvement in the final complexity (when component smoothness parameters are highly nonuniform). For linear predictors, the bounds are more informative (and in our opinion, interesting), as they are directly dependent on the data matrix. We will further clarify these points in our revision.
>
> ---
>
> ### Questions
> > Q1: Is there a way to intuitively understand when or why $\tilde{L}/L_{\mathrm{max}}$, $\hat{L}/L_{\mathrm{max}}$ could be ‘much’ smaller than $1$ (with possible $n^{\alpha}$-ish dependencies), both for the Or are there at least any examples apart from the linear predictor that demonstrate significantly small $\tilde{L}/L_{\mathrm{max}}$, $\hat{L}/L_{\mathrm{max}}$?
>
> For linear predictors, the main source of the improvement can be seen as our bounds being dependent on the operator (as opposed to Frobenius) norm of the data matrices, as discussed in Lines 241-250 in Section 4. The operator-to-Frobenius norm relaxation is almost always loose, and often by a factor of $n$ for $n \times n$ matrices, as discussed in Lines 249-250. At an intuitive level, we expect this relaxation to be loose (and our bound to be tighter) when there is weak correlation between the data points, which we take to an extreme in our Gaussian data example (but similar conclusions could be drawn for e.g., sub-Gaussian data). We expect this to be the case in datasets where data is collected from independent sources, which is a standard assumption for even being able to guarantee good statistical properties pertaining to learning. For general convex smooth functions, our dependence on the smoothness also improves from the maximum to the average, leading to $O(n)$ improvement when the component smoothness constants are highly nonuniform.
>
> > Q2: Based on my knowledge, in coordinate descent algorithms it is common to use different step sizes for different coordinates, say $\eta_i = 1/L_i$, to improve dependencies on the $L$’s. Would it be possible to do something similar for this case if we take the primal-dual approach?
>
> This is a good point. We first note that our focus is not on proposing new algorithms but deriving tighter bounds for shuffled SGD, so we stick to the constant step size over all components/inner iterations, which agrees with previous works on shuffled SGD [1–3] and empirical practice. On the other hand, deploying different step sizes for each component gradient update based on the component smoothness would be an interesting approach. We believe it is possible to incorporate different step sizes in our analysis by carrying out the argument using a weighted $\ell_2$ norm, similar to what was done for coordinate methods in nonconvex settings [4]. However, this is out of scope of the present work, and we leave it for future research.
>
>
> > Q3: Can you elaborate a bit more on why the inequality (iii) of (4) (in Section 4) is loose by a factor of $n$ in most cases? My understanding of the linear predictor case is that the norm-trace gap corresponds to the largest VS sum of singular values, and hence if the singular values are even, then the gap can be as loose as a factor of n, while if we have a single large $\lambda_{\mathrm{max}}$ then the inequality can be tight. I wonder if (i) this interpretation is correct, and (ii) whether something similar could hold for the $\tilde{L}^g$, $\hat{L}^g$’s in the general convex case.
>
> The reviewer’s interpretation is correct. This is exactly what happens in our Gaussian example, but one can generalize. For general convex settings, there is no specific structure of data matrices in the optimization objective, so the best one may expect is the improvement from the max (in previous works [1–3]) to the average smoothness constant. We discuss those in Lines 493-501 in Appendix B.
>
> > Q4: Minor question: Is it true Algorithm 1 is completely equivalent to the standard SGD with random reshuffling (without the $\mathbf{y}$’s), and the dual variables are just there for illustrative purposes?
>
> Yes, the primal-dual version is provided for convenience of the primal-dual analysis; there is no difference in the actual algorithm.

---

> ### Author Response · Authors · 2024-08-06
> **References for the rebuttal**
>
> ### References
>
> [1] Jaeyoung Cha, Jaewook Lee, and Chulhee Yun. Tighter lower bounds for shuffling SGD: Random permutations and beyond. In Proc. ICML'23, 2023.
>
> [2] Konstantin Mishchenko, Ahmed Khaled, and Peter Richtárik. Random reshuffling: Simple analysis with vast improvements. In Proc. NeurIPS’20, 2020.
>
> [3] Lam M Nguyen, Quoc Tran-Dinh, Dzung T Phan, Phuong Ha Nguyen, and Marten Van Dijk. A unified convergence analysis for shuffling-type gradient methods. The Journal of Machine Learning Research, 22(1):9397–9440, 2021.
>
> [4] Xufeng Cai, Chaobing Song, Stephen J Wright, and Jelena Diakonikolas. Cyclic block coordinate descent with variance reduction for composite nonconvex optimization. In Proc. ICML'23, 2023.

---

> > ### Comment · Reviewer_JY5g · 2024-08-11
> >
> > Thank you for your detailed and precise answers.

---

> > > ### Author Response · Authors · 2024-08-11
> > > **A small request for clarification**
> > >
> > > Thank you for reading through our rebuttal. We are glad to learn you found our answers detailed and precise.
> > >
> > > We noticed that you kept your score as 'borderline' and wanted to kindly ask if you could provide some additional insight into your reasoning. Understanding your perspective would be valuable for us to address any remaining concerns and improve the work further.
> > >
> > > Thank you in advance,
> > > Authors

---

### Author Rebuttal · Authors · 2024-08-06

We would like to thank all the reviewers for their precious time and valuable feedback. In this top-level rebuttal, we reiterate the contributions and strengths of our work.

---

We first summarize our main contribution by quoting the Review S4vW where they acutely pointed out
> This paper aims at improving existing bounds on random reshuffling. While lower bounds are tight in the worst case, refined smoothness definitions allow to take a larger step-size in favorable cases, which in turns allows faster convergence.

---

Further, we are encouraged by reviewers recognizing the following aspects of our work:
- **Improved complexity**: All reviewers recognize that our fine-grained analysis provides improved complexity over previous worst-case analyses.

- **Novel primal-dual view**:

    * > (Reviewer JY5g) “New to me and seems to be a potentially useful technique, … using component-wise structures for improved convergence rates could be one of the many meaningful future directions for the shuffling community”,

    * > (Reviewer S4vW) “Nice use of the finite-sum assumption, which allows to use a primal-dual reformulation”.

- **Extensive study**:

    * > (Reviewer JY5g) “consider general types of shuffling-based algorithms (RR, SO, IG) which I appreciate”,

    * > (Reviewer Ua2v) “extend the results to non-smooth setting”,

    * > (Reviewer S4vW) “As a side note, results include mini-batching, which is nice to have”.

- **Writeup**: Three reviews rate our soundness 3 and presentation 3 (Review JY5g, Review EeyJ, Review S4vW). It is also particularly mentioned that

    * > (Reviewer EeyJ) “The paper is well written and easy to follow, contributions are clearly stated (Table 1)”.

---

We look forward to interacting with the reviewers and providing any further explanations as needed.

Sincerely,

Authors

---

### Decision · Program_Chairs · 2024-09-25

**Decision:**

Accept (poster)

**Comment:**

The paper studies shuffling-based SGD on convex finite-sum optimization problems. Through a primal-dual analysis on the primal algorithm SGD, the authors develop new convergence theorems for random-reshuffling and shuffle-once SGD with alternative smoothness constants that can improve upon the existing bounds that contain the max smoothness constant. The authors then develop convergence bounds for smooth and non-smooth convex generalized linear model cases, demonstrating similar but more fine-grained bounds that can better reflect the data matrix.

The paper presents nontrivial and novel analysis inspired from the connection of finite-sum SGD with coordinate descent in the dual space. The bounds improve upon existing results in the literature and can offer tighter bounds especially when there is a large gap between the max smoothness constant and the "average" smoothness. The discussion provided in Section 4 is inspiring, in that it suggests that we should take the structure of data into account in the convergence analysis to get tighter results.

The newly proposed constants $\tilde L$, $\hat L$, etc…  are not readily interpretable, as the authors also admitted in the discussion. The authors provide some numerical experiments demonstrating that they can be smaller than $L_\max$, but I recommend the authors to supplement the paper with some theoretical analysis of the proposed constants in some specialized scenarios, e.g., when $L_i$'s decay as $i^{-\alpha}$.

Overall, the reviews were mostly positive and the contributions look solid; hence I recommend acceptance of this paper.